# Failure criteria for porous dome rocks and lavas: a study of Mt. Unzen, Japan

Rebecca Coats[1], Jackie E. Kendrick[1], Paul A. Wallace[1], Takahiro Miwa[2], Adrian J. Hornby[1,3], James D. Ashworth[1], Takeshi Matsushima[4], Yan Lavallée[1]

[1]Department of Earth, Ocean and Ecological Sciences, The University of Liverpool, Liverpool, L69 3GP, UK

[2]National Research Institute for Earth Science and Disaster Prevention, Ibaraki, 305-0006, Japan

[3]Now at Department of Earth and Environmental Sciences, Ludwig-Maximilians-Universität München, Munich, 80333, Germany

[4]Institute of Seismology and Volcanology, Kyushu University, Nagasaki, 855 0843, Japan

*Correspondence to*: Rebecca Coats (r.coats@liverpool.ac.uk)

**Abstract**

The strength and macroscopic deformation mode (brittle vs ductile) of rocks is generally related to the porosity and pressure conditions, with occasional considerations of strain rate. At high temperature, molten rocks abide to Maxwell's viscoelasticity and their deformation mode is generally defined by strain rate or reciprocally, by comparing the relaxation timescale of the material (for a given condition) to the observation timescale – a dimensionless ratio known as the Deborah (*De*) number. Volcanic materials are extremely heterogeneous, with variable concentrations of crystals, glass/ melt and vesicles (of different sizes), and a complete description of the conditions leading to flow or rupture as a function of temperature, stress and strain rate (or timescale of observation) eludes us. Here, we examined the conditions which lead to the macroscopic failure of variably vesicular (0.09-0.35), crystal–rich (~ 75 vol.%), pristine and altered, dome rocks (at ambient temperature) and lavas (at 900 °C) from Mt. Unzen Volcano, Japan. We found that the strength of the dome rocks decreases with porosity and is commonly independent of strain rate; when comparing pristine and altered rocks, we found that the precipitation of secondary mineral phases in the original pore space caused minor strengthening. The strength of the lavas (at 900 °C) also decreases with porosity. Importantly, the results demonstrate that these dome rocks are weaker at ambient temperatures than when heated and deformed at 900 °C (for a given strain rate resulting in brittle behaviour). Thermal stressing (by heating and cooling a rock up to 900 °C at a rate of 4 °C min$^{-1}$, before testing its strength at ambient temperature) was found not to affect the strength of rocks.

In the magmatic state (900 °C), the rheology of the dome lavas is strongly strain rate dependent. Under low experimental strain rate conditions ($\leq 10^{-4}$ s$^{-1}$) ductile deformation dominated (i.e., the material sustained substantial, pervasive deformation) and displayed a non–Newtonian, shear thinning behaviour. In this regime, the apparent viscosities of the dome lavas were found to be essentially equivalent, independent of vesicularity, likely due to the lack of pore pressurisation and efficient pore collapse during shear. At high experimental strain rates ($\geq 10^{-4}$ s$^{-1}$) the lavas displayed an increasingly brittle response (i.e., deformation resulted in failure along localised faults); we observed an increase in strength and a decrease in strain–to–failure as a function of strain rate. To constrain the conditions leading to failure of the lavas, we analysed and compared the critical Deborah number at failure (*De*$_c$) of these lavas to that of pure melt (*De*$_{melt}$=$10^{-3}$–$10^{-2}$; Webb & Dingwell, 1990). We found that the presence of crystals decreases *De*$_c$ to between $6.6\times10^{-4}$–$1\times10^{-4}$. The vesicularity ($\varphi$), which dictates the strength of lavas, further controls *De*$_c$ following a linear trend. We discuss the implications of these findings for the case of magma ascent and lava dome structural stability.

# 多孔質な岩石及び溶岩の破壊基準：雲仙火山溶岩ドームでの研究

マグマ(溶岩)と岩石のレオロジーと強度は，応力の蓄積と散逸を支配し，噴火様式や山体の構造的安定性に影響を与える．火山噴出物は極端に不均質であり，様々な量・サイズの結晶，ガラス(メルト),気泡を含む．そのため，温度・応力・歪速度の関数として，その流れや亀裂形成を引き起こす状態を完全に記載することは難しい．ここで我々は，雲仙火山において溶岩ドームを形成し様々な発泡度(9-35%)を有する高結晶度(〜75%)な岩石(常温)と溶岩(900度)について，その破壊を引き起こす状態を検討した．その結果，我々は岩石の強度は空隙率とともに

減少し，歪速度に依存しないことを発見した： 新鮮な岩石と変質したものでは，後者でわずかに強度が大きい．また，溶岩(900℃)の強度も空隙率とともに減少する．この結果は重要なことに，脆性的振る舞いを起こす歪速度において，常温における岩石の強度は，それを900℃まで加熱し変形させたときの強度よりも弱いことを示している．このとき，熱応力は岩石の強度に影響を与えない．

高温条件(900℃)では，溶岩のレオロジーは歪速度に強く依存する．低歪速度下($<10^{-4}$ s$^{-1}$)では，溶岩は塑性的に振る

舞い(物質が広範な固体変形を持続させる)，非ニュートン流体としてずり粘減の振る舞いを示した．このレジームでは，溶岩の見かけ粘性は，おそらく剪断時の効率的な空隙崩壊のため，発泡度に依存しない．高歪速度下($>10^{-4}$ s$^{-1}$)では，溶岩は益々の脆性的な応答(局所的な断層に沿った破壊による変形)を示す; 歪速度の関数として，強度の増加と破壊へ至るときの歪の減少が観察された．溶岩の破壊を引き起こす状態を制約するため，これら溶岩における破壊時の臨界デボラ数($De_c$, 緩和時間と実験観察時間の比)を解析し，メルトにおけるそれ($De_{melt}$, $=10^{-3}$-

$10^{-2}$; Webb & Dingwell, 1990)と比較した．我々は結晶の存在が$De_c$を$6.6\times10^{-4}$–$1\times10^{-4}$まで減少させることを発見した．またさらに，溶岩の強度に影響する発泡度($\varphi$)も$De_c$を線形傾向のようにコントロールする．我々はこれらの発見が与える，マグマ上昇と溶岩ドームの構造的安定性への示唆を議論する．

# 1 Introduction

## 1.1 Lava dome eruptions

Magma ascends to the Earth's surface and erupts through a wide spectrum of eruptive style (e.g. Siebert et al., 2015), which contributes to the construction of different volcanic edifices (e.g. de Silva and Lindsay, 2015). Amongst this activity, lava domes form when viscous magma accumulates and creates mounds of rocks and lava above the vent (Sparks, 1997; Fink and Anderson, 2000). These dome–building events make up approximately 6 % of volcanic eruptions worldwide (Calder et al. 2015) and their characteristics are governed by the rheology of the erupted magmas (Gonnermann and Manga, 2007; Lavallée et al., 2007). The emplacement of lava domes may be endogenous or exogenous, whether growing through inflation from within, or through the piling up of discrete extrusive bodies (Hale and Wadge, 2008). In some extreme cases the latter can manifest as lava spines that extrude in a near–solid state (Angelo Heilprin, 1903; Stasiuk and Jaupart, 1997; Young et al., 1998; Tanguy, 2004; Scott et al., 2008; Vallance et al., 2008; Kendrick et al., 2012; Cashman and Sparks, 2013). Dome eruptions can produce a range of primary hazards, from ash fall to large–scale pyroclastic density currents, generated by gravitational collapse (e.g. Sparks and Young, 2002). They also have the potential to generate secondary hazards such as lahars (e.g. Nevado del Ruiz, Colombia; Pierson et al. 1990); edifice failure induced by magma intrusions (Voight and Elsworth, 1997; Reid et al., 2010), and lava dome collapse, as the mass cools or redistributes (e.g. Elsworth and Voight, 1996). In seismically active areas, strong tectonic earthquakes can both initiate activity and promote structural instability (e.g. Mayu–yama, Japan; Siebert et al. 1987), even in long–dormant systems (e.g. Merapi, Indonesia; Surono et al. 2012). The eruption, emplacement and stability of lava domes reflects the mechanical properties of their constituent materials; thus, it is essential that the evaluation of monitoring data and development of improved hazard forecasting tools at lava dome volcanoes be based on a description of the mechanical and rheological properties of the materials.

## 1.2 Lava dome rheology

The rheology of silicate melts has been explored extensively (e.g. Dingwell and Webb, 1989, 1990; Webb and Dingwell, 1990; Webb and Knoche, 1996; Fluegel, 2007; Giordano et al., 2008; Cordonnier et al., 2012b). Dingwell and Webb (1989) demonstrated that silicate liquids are viscoelastic bodies, that abide to the glass transition– a temperature–time space that defines their structural relaxation according to the theory of viscoelasticity of Maxwell (1867). Maxwell's work established that the structural relaxation time–scale $\tau$ equals the ratio between the melt viscosity $\eta_m$ (in Pa.s) and its elastic modulus at infinite frequency $G_\infty$ (in Pa) according to:

$$\tau = \eta_m / G_\infty \tag{1}$$

Dingwell and Webb (1989) compiled information for different silicate liquids and showed that $G_\infty$ is essentially invariant and approximately $10^{10\pm0.5}$ Pa in the temperature range of interest for magmatic systems. Thus, the relaxation time–scale of silicate melts can simply be related to their viscosity at a given temperature. Extensive experimental efforts in the community have resulted in the creation of a complete, non–Arrhenian model for silicate melt viscosity, as a function of composition and temperature (e.g. Giordano et al., 2008). The concept of viscoelasticity and relaxation timescale can therefore be applied to a range of volcanic processes.

Viscoelasticity dictates the behaviour of a magma. A rheological description of viscoelastic materials may be cast via the non–dimensional Deborah number, $De$ (e.g. Reiner 1964), which is defined by a ratio between Maxwell's relaxation time–scale, $\tau$ (Eq. 1) and the time–scale of observation, $t_{obs}$:

$$De = \frac{\tau}{t_{obs}} \tag{2}$$

This relationship states that under observation timescales longer than the relaxation timescale (for a given melt viscosity), a melt may flow like a liquid; but at short observation timescales, a melt may behave as a solid (like a glass). In such a kinetic framework, increasing the temperature reduces the viscosity and therefore the time required for structural relaxation. As the relaxation time–scale is inversely proportional to the structural relaxation rate, it can thus be said that the structural relaxation rate defines the transition between the liquid and solid states (commonly referred to as the glass transition, $T_g$). Dingwell and Webb (1990) demonstrated that at $De < 10^{-3}$, a silicate melt can be described as a Newtonian fluid. However, when silicate melts are deformed at higher rates where the observation time–scale is short, $10^{-3} < De < 10^{-2}$, the melt structure accumulates damage upon deformation which results in an apparent non–Newtonian behaviour. At $De > 10^{-2}$, silicate melts undergo the glass transition and ruptures (Dingwell and Webb, 1990; Wadsworth et al., 2017); this is known as the critical Deborah number, $De_c$ – a criteria met in several eruptive scenarios, including fragmentation and explosive eruptions (e.g. Dingwell, 1996).

During transport and eruption, magmas crystallise and volatiles are exsolved (e.g. Cashman, 1992; Martel and Schmidt, 2003), resulting in magmatic suspensions, undergoing significant rheological changes (e.g. Lejeune and Richet, 1995; Barmin et al., 2002). In particular, dome–building eruptions have been observed to produce variably vesicular (generally ≲ 0.40) and crystalline (e.g. 0–100 vol.%) lavas (Castro et al., 2005; Mueller et al., 2005, 2011a; Lavallée et al., 2007; Pallister et al., 2008; Cordonnier et al., 2009; Calder et al., 2015; Heap et al., 2016a). The addition of crystals to a melt increases the effective viscosity (Lejeune and Richet, 1995). At moderate crystal fraction (below ~ 25 vol.%) this can be approximated by the Einstein–Roscoe equation (Einstein, 1911; Roscoe, 1952), and variations thereof (see Mader et al., 2013 and references therein). When particle concentrations reach a critical fraction that promotes interaction (typically ≤ 0.25, depending on crystal morphology (Mader et al., 2013)), the suspension becomes non–Newtonian (Deubelbeiss et al., 2011). Experiments on dome lavas at high temperature have shown that the apparent viscosity of these suspensions decreases with strain rate (Lavallée et al., 2007; Avard and Whittington, 2012) – a shear thinning effect influenced by crystal alignment and interaction (Vona et al., 2011); crystal plasticity (Kendrick et al. 2017), and fracture processes (Lavallée et al., 2008; Kendrick et al., 2013b). The addition of a separate gas phase to a magma adds further rheological complexity (Lejeune et al., 1999), serving to increase or decrease viscosity depending upon the volume fraction of bubbles, pore pressure, the initial viscosity of the melt, the amount of deformation they are subjected to (e.g. Manga et al., 1998; Llewellin and Manga, 2005), and pore connectivity, which may promote outgassing and pore compaction (e.g. Ashwell and Kendrick et al., 2015).. Bubbles will affect the viscosity of the suspension depending on their capillary number, Ca, a dimensionless ratio of the deforming viscous stress over the restoring stress from surface tension. A more spherical bubble will generally have a low Ca, as restoring stresses dominate, and will behave as a barrier which fluid flow will have to deviate around resulting in an increased viscosity of the suspension. On the other hand, an elongate bubble generally has a high Ca, as deforming stresses dominate, and may act as free slip surface causing a decrease in the suspension viscosity (e.g. Manga et al., 1998; Mader et al., 2013). Three–phase models, although less explored than two–phase flows, have been modelled by Truby et al. (2015) by combining two sets of two–phase equations. Despite the aforementioned rheological studies focused on the viscosity of magmatic suspensions, the conditions leading to failure of such magmatic suspensions have received less attention. Following the work of Lavallée et al. (2007), Gottsmann et al. (2009) showed that the presence of crystals may reduce the strain rate required to rupture magma (if one was to consider the melt relaxation rate) to conditions where $De < 10^{-2}$ and Lavallée et al. (2008) and Gottsmann et al. (2009) showed that brittle processes may be active at conditions two orders of magnitude lower than such a purely brittle limit. Cordonnier et al. (2012a) explored the effect of crystallinity on magma rupture, showing that $De$ indeed decreases with crystallinity. However, here we note that when determining the Deborah number for their experimental findings, the relaxation time–scale was calculated using the apparent viscosity of the suspension rather than the viscosity of the interstitial melt, which is the basis for the applicability of viscoelasticity in this scenario (this will be discussed further in section 5.2). Important questions remain as to the contribution of vesicles on the rupture of magmas, as the strength of geomaterials in the brittle field is generally described in terms of porosity (e.g. Paterson and Wong, 2005, and references in section 1.3).

### *1.3 Lava dome mechanics*

Various numerical models have been developed to evaluate the structural stability of lava domes and, with sufficient knowledge of a volcanic edifice and the properties of the materials it holds, collapse events can be modelled effectively (e.g. Elsworth and Voight, 1996). Although elegant and complex, these simulations tend to make non–trivial assumptions regarding vent geometry, dome morphology, and material properties (e.g. Ball et al., 2015). Volcanic domes are composed of materials with a vast spectrum of heterogeneities and degree of coherence (Mueller et al., 2011b; Lavallée et al., 2012, 2018) and although assigning fixed values for the material properties of dome rocks may be computationally beneficial, accounting for the wide range of physical and mechanical properties of dome materials remain a great source of uncertainty. Mechanical testing can be carried out to resolve the behaviour of rocks (see Paterson and Wong, 2005 and references therein) and this has resulted in a recent surge in laboratory testing to advance the understanding of the tensile strength, compressive strength, frictional coefficient and flow behaviour of these heterogeneous dome rocks and magmas as a function of temperature and stresses or strain rates (Smith et al., 2007, 2011; Lavallée et al., 2007; Hess et al., 2008; Kendrick et al., 2012, 2013b, 2013a; Kolzenburg et al., 2012; Heap et al., 2014a; Hornby et al., 2015; Lamb et al., 2017; Lamur et al., 2017 and more.)

The uniaxial compressive strength of volcanic rocks has been found to inversely correlate with porosity (Al-Harthi et al., 1999; Kendrick et al., 2013b; Heap et al., 2014a, 2014b, 2016b; Schaefer et al., 2015), and to positively correlate with strain rate (Schaefer et al., 2015). In volcanic rocks, porosity is made up of vesicles and micro–fractures, which contribute to the mechanical behaviour and strength of the rock (Sammis and Ashby, 1986; Ashby and Sammis, 1990; Heap et al., 2014a; Bubeck et al., 2017; Collombet et al., 2017; Griffiths et al., 2017). Two models have gained traction to explain the strength of rocks. The pore–emanating crack model of Sammis and Ashby (1986), describes the case of a pore–only system where cracks nucleate from the pores and propagate in the direction parallel to the principal stress, when the applied stress overcomes the fracture toughness of a rock. As the applied stress increases, the micro–fractures propagate and coalesce, leading to macroscopic failure. An analytical estimation of this model was derived by Zhu et al., (2010) to estimate the uniaxial compressive stress ($\sigma$) of a sample, with a pore radius ($r$), as a function of its porosity ($\varphi$) and the fracture toughness ($K_{IC}$):

$$\sigma = \frac{1.325}{\varphi^{0.414}} \frac{K_{IC}}{\sqrt{\pi r}} \tag{2},$$

In contrast, the sliding wing–crack model of Ashby and Sammis (1990) considers only pre–existing micro–fractures inclined from the principal stress direction. The model describes that first, the frictional resistance of the crack must be overcome before wing–cracks can form, then the fracture toughness must be overcome for them to propagate and interact. The analytical approximation for this model was developed by Baud et al., (2014):

$$\sigma = \frac{1.346}{\sqrt{1+\mu^2}-\mu} \frac{K_{IC}}{\sqrt{\pi c}} D_0^{-0.256} \tag{3},$$

where $\mu$ is the friction coefficient of the crack, $c$ is the half–length of a pre–existing crack, and $D_0$ is an initial damage parameter (which takes into consideration the number of cracks per unit area and their angle with respect to the principal stress).

Heap et al., (2014a) experimentally demonstrated that neither model fully satisfied the mechanical data obtained for volcanic rocks and suggested that a microstructural model that combines the two mechanisms must be developed to permit the design of simulations considering the mechanical behaviour of microstructurally complex volcanic materials.

The problem of lava dome stability does not simply require knowledge of hot lavas or cold rocks; it further requires understanding of the effects of temperature (e.g. Harris et al., 2002); chemical alteration (e.g. Lopez & Williams 1993; Ball et al. 2015); pore pressure (Farquharson et al., 2016), thermal stressing (Heap et al., 2009, 2010, 2014a; Kendrick et al., 2013a; Schaefer et al., 2015) and mechanical stressing at different rates such as during seismic shaking (Cole et al., 1998; e.g. Voight, 2000; Calder et al., 2002) or magmatic intrusions (Walter et al., 2005) on the mechanical properties of the materials, many

aspects of which have been tested in the context of edifices. The cooling of crystalline lava bodies results in the generation of fractures (Fink and Anderson, 2000; Takarada et al., 2013; Eggertsson et al., 2018)  – leaving a highly fractured, blocky mass, the mechanical impact of which is difficult to quantify (Voight, 2000; Voight and Elsworth, 2000). Furthermore, thermal stressing cycles that could result from proximity of hot magma in a conduit, lava dome or edifice following a new eruptive episode, have been found to only weakly modify the strength of commonly micro–fractured volcanic rocks (Heap et al. 2009; Kendrick et al. 2013; Schaefer et al. 2015,), unless they contain thermally liable minerals (Heap et al., 2012, 2013a, 2013b). Recent experiments on porous basalt by Eggertsson et al. (2018) have shown that rocks that are essentially void of micro–cracks (likely due to slow cooling), are however susceptible to fracture damage by thermal stressing (i.e., forming cooling joints); in contrast, micro-fractured rocks, may not necessarily accumulate more damage during cooling, yet upon contraction, pre-existing fracture may widen to give way to the ingression of hydrothermal fluids (e.g. Lamur et al., 2018), further contributing to the stress balance and mechanical response.

### 1.4 Mt. Unzen lava dome

The Unzendake volcanic complex is situated on the Shimabara peninsula in South–Western Japan (Fig. 1a). The volcanic complex began to grow 0.5 Ma and now covers 20 km (E–W) by 25 km (N–S) (Takarada et al., 2013). Unzendake exhibits an intricate eruptive history of lava domes, flows and pyroclastic deposits (Nakada and Fujii, 1993) of predominantly dacitic composition (Nakada and Motomura, 1999).

On 17 November 1990, after 198 years of quiescence, a phreatic eruption occurred at Mt. Unzen, which was accompanied by multiple earthquake swarms (Matsushima and Takagi, 2000). This was followed shortly afterwards by a phreatomagmatic eruption along with intense edifice swelling, and on 20 May 1991, the extrusion of a lava spine initiated the growth of the Heisei–Shinzan dome complex (Nakada and Fujii, 1993; Takarada et al., 2013). This introduced a 45–month long period of lava dome activity with growth being primarily exogenous in periods of high extrusion rate, and endogenous in times of low effusion rate (Nakada et al., 1995b, 1999a). The final stage of growth was marked by the extrusion of a spine between October 1994 and February 1995 (which can be seen this present day; Fig. 1b–c), characterised by pulsatory ascent and seismicity (Umakoshi et al., 2008; Lamb et al., 2015), along fault zones defined by compactional shear (Ashworth et al., *in prep*) and mineral reactions, crystal plasticity and comminution (Wallace et al., *in review*). The end of the eruption was followed by cooling of the lava dome and thermal contraction that caused multiple joints (Takarada et al., 2013). Fumarole activity has continued to the present day, with temperatures decreasing from 300 °C in mid–2007 to 90 °C in 2011 (Takarada et al., 2013). In total, 13 lava lobes were formed, and, at its maximum size, the lava dome was 1.2 km (E–W) by 0.8 km (N–S) wide. In particular lobe 11, which dominated the Eastern side of the complex (Nakada et al., 1995a, 1999b) has long been unstable, which has led to partial collapses that generated several pyroclastic density currents (PDCs; Nakada et al., 1999a; Sakuma et al., 2008). The flows were estimated to have travelled at 200 km hr$^{-1}$, up to 5.5 km down the Oshigadani Valley (Yamamoto et al., 1993; Takarada et al., 2013). All in all, pyroclastic flows buried and/or burned approximately 800 buildings, with  debris flows destroying a further ~ 1,700, and in the summer of 1991 the number of evacuated persons exceeded 11,000 (Nakada et al., 1999a). The Committee of Survey and Countermeasure on Lava Dome Collapse at Mt. Unzen advises that the risk of collapse of lobe 11 is high, an exclusion zone remains active to the E of the summit and access to the lava dome is strictly limited. Data from electro–optical distance measuring instruments suggest that lobe 11 has advanced 1 m in 14 years (measurements from 1997–2011), and recent observations from ground-based synthetic aperture radar show the development of a shear fracture (Kohashi et al., 2012). Therefore, the complete or partial collapse of the lobe and the generation of block–and–ash flows are likely hazards, particularly after large regional earthquakes. The current uncertainty regarding the structural stability of the dome at Mt. Unzen, particularly after seismic activity, has led to recent field campaigns and mechanical studies of the dome material (e.g. Cordonnier et al., 2009; Hornby et al., 2015). The destabilisation of lava domes due to tectonic

activity is essentially a superficial process, meaning the stress balance may be considered as a uniaxial problem, and tested as such (e.g. Quane and Russell, 2005).

## 2 Materials and methods

### 2.1 Sample selection

225 Mt. Unzen lava dome is made up of porphyritic, dacite ($\sim$ 63 wt.% $SiO_2$) lava blocks which typically have large (> 3 mm) and abundant (> 25 vol.%) plagioclase phenocrysts, along with lesser amounts of amphibole ($\sim$ 5 vol.%), biotite ($\sim$ 2 vol.%) and quartz ($\sim$ 2 vol.%) phenocrysts and microphenocrysts set in a partially crystalline ($\sim$ 50 vol.%) groundmass of plagioclase, pyroxene, quartz pargasite, and Fe–Ti oxides in a rhyolitic interstitial glass (Nakada & Motomura 1999; Wallace et al. *in review*). However, as the dome was formed through both, exogenous and endogenous growth, the petrological history of the

230 eruptive products varies widely and as such the microstructure of the blocks forming the dome varies considerably. Furthermore, lasting heat sources and ongoing fumarolic activity have led to local thermal and hydrothermal alteration of the dome (Almberg et al., 2008). This heterogeneity calls for a variable sample suite to represent the dome material, and to constrain the processes of deformation and cooling that occurred throughout lava dome formation, that influences its current structural stability.

In this study, 9 samples were selected with different properties. Samples UNZ-1, 2, 4, 5, 7, and 8 were collected from easily accessible, June 1993 block–and–ash flow deposits in the Minami–Senbongi area, north–east of the spine; UNZ-13 was collected from the May-August 1991 deposits in the restricted area of the Mizunashi River, east of the spine (see Fig. 1b). These rocks were collected as they represent the freshest (unaltered) materials that originate from dome collapse events during eruption, prior to any chemical alteration (e.g. Cordonnier et al., 2008). Sample UNZ-11 was collected on lobe 11 of the dome,

selected as it showed signs of hydrothermal alteration (crusted, white and friable). UNZ-12 was collected on the dome, just east of the lava spine, and was chosen specifically for its reddish colour which suggested thermal alteration and oxidation. Each sample block was then cored to make multiple 20 mm diameter cylindrical cores, cut, and then ground parallel to 40 mm in length (Fig. S1) to maintain a 2:1 aspect ratio in accordance with the ISRM suggested method (ISRM Turkish National Group, 2015).

### 2.2 Sample characterisation and preparation

#### 2.2.1 Geochemistry

The bulk geochemical compositions of selected samples were determined in a PANalytical Axios Advanced X–Ray Fluorescence Spectrometer (XRF) at the University of Leicester (using fused glass beads prepared from ignited powders). Sample to flux ratio was kept at 1:5, 80 % Li metaborate: 20 % Li tetraborate flux. Results are quoted as component oxide

weight percent and re–calculated to include LOI (loss–on–ignition).

The geochemical composition of the interstitial glass in sample block UNZ-4 was determined using a Cameca SX–5 Field Emission Electron Probe Microanalyser (EPMA) at the University of Oxford. A variety of standards were used to calibrate the spectrometers, including Wollastonite for Ca, and Albite for Al, Na and Si. Secondary reference standards, of which the exact chemistry was known, were utilised for better precision and accuracy. These were Labradorite and kn18 glass (comendite

obsidian, Kenya), used as the chemistries were similar to those of the Mt. Unzen glass sample. Analyses used an accelerating voltage of 15 KeV, a beam current of 6 nA and a defocussed spot size of 10 μm. The data were checked for major element oxides' totals.

### 2.2.2 Porosity

The porosity and character of the pores (i.e., whether connected or isolated) was assessed using an AccuPyc 1340 helium pycnometer from Micromeritics. Firstly, height ($h$; in m), radius ($r$; in m) and mass ($m$; in kg) were recorded for each cylindrical core sample, providing a constraint on sample density ($\rho_s$; in kg/m$^3$):

$$\rho_s = {}^m/_{\pi r^2 h} \tag{4}.$$

Secondly, the solid density of the rocks ($\rho_0$) was constrained by measuring the mass and volume of a powdered lump from each rock in a pycnometer; from these measurements, the total porosity of each rock could be estimated via:

$$\varphi_T = 1 - ({}^{\rho_s}/_{\rho_0}) \tag{5}.$$

To constrain the fraction of isolated pores in the rocks, the skeletal volume ($V_{skeletal}$; in m$^3$) of each core was measured in the pycnometer. The porosity connected to the outside of the sample (henceforth termed connected porosity), $\varphi_O$, could then be calculated via:

$$\varphi_O = 1 - ({}^{V_{skeletal}}/_{\pi r^2 h}) \tag{6},$$

and isolated porosity, $\varphi_i$, via:

$$\varphi_i = \varphi_T - \varphi_O \tag{7},$$

The porosity determination was used to omit outliers from any sample block to ensure that the rocks of a given porosity were tested and compared to one another.

### 2.2.3 Microstructures

Thin sections of UNZ- 4,11,12 and 13 were prepared with a fluorescent dyed epoxy; selected as they cover a vast range of sample diversity; including both the lower and upper bounds of porosity, and collection site. Images were acquired using a DM2500P Leica microscope in plane–polarised light. To further constrain the microstructures of each sample block, backscattered electron (BSE) images were taken of each sample using a Philips XL30 tungsten filament scanning electron microscope (SEM), equipped with an energy–dispersive X–ray spectrometer (EDS), and a Hitachi TM3000 SEM at the University of Liverpool. Stubs of the samples were set in epoxy, polished and carbon coated, before being imaged in the Philips XL30 at a working distance of 13±0.1 mm using a 20 kV beam voltage, a 60–90 µA beam current and a spot size of 5. Thin sections of the samples were imaged with the Hitachi TM3000 using a 15 kV beam and 10 mm working distance.

### 2.2.4 Thermal Analysis

To constrain the conditions at which to carry out the high temperature uniaxial tests, we evaluated the softening point of the Mt. Unzen dome rock using a Netzsch 402 F1 Hyperion thermomechanical analysis (TMA) at the University of Liverpool. Under a 20 mL min$^{-1}$ argon flow, a 6.37 mm tall, 5.87 mm wide, cylindrical sample of UNZ-8 was placed under a constant load of 3 N and heated at 10 ˚C min$^{-1}$ to 1100 ˚C. The softening point of the material was found as the temperature at which the applied load counteracts sample expansion by inducing viscous flow (and sample shortening) during heating. This was detected at 824.6 °C, 80.6 minutes into the measurement (Fig. S5). An experimental temperature of 900 °C was selected as, being well above the softening point, this is high enough to allow for flow to occur on the timescales under investigation. This chosen temperature is close to the magmatic temperature (850–870 °C) constrained to have followed mixing (Venezky and

Rutherford, 1999) and above the glass transition of Unzen spine material (790 °C) measured by differential scanning calorimetry at a rate of 10 °C min$^{-1}$ (Wallace et al. *in review*), though the temperature profile within the conduit and dome during emplacement is poorly constrained.

### 2.2.5 Thermal stressing

Selected cores of pristine material were thermally stressed in a Carbolite box furnace to examine the effects of experimentally induced heating–cooling cycles on the residual strength of rock cores. Cores were subjected to heating at 4 °C min$^{-1}$ followed by 1–hr dwell at 900±3 °C (sample temperature) and cooling at 4 °C min$^{-1}$. The density and porosity of each sample were measured before and after thermal stressing, and the products were further subjected to uniaxial compressive strength tests.

## 2.3 Uniaxial compression experiments

Uniaxial compressive strength tests were carried out using a 50 kN 5969 Instron benchtop press and a 100 kN Instron 8862 uniaxial press with a three–zone, split cylinder furnace using the parallel plate method in the Experimental Volcanology and Geothermal Research Laboratory at the University of Liverpool (Fig. 2). Experiments were carried out both at ambient temperature (~20 °C) and at high temperature (900 °C, using a heating rate of 4 °C min$^{-1}$). Tests were conducted at constant strain rates of $10^{-1}$, $10^{-3}$ or $10^{-5}$ s$^{-1}$ (see Table 1 for the range of experimental conditions). The apparatus monitored the applied load and piston extension at 10–1000 Hz (depending on set experiment rate) and the Bluehill® 3 software was used to compute data and calculate strain ($\varepsilon$) and compressive stress from the input sample dimensions. [Note: all mechanical data have been corrected for the compliance of the setup at the relevant experimental temperature, quantified via Instron procedures that monitor length changes due to loading of the pistons in contact with one another]. The end of each experiment was defined by either (1) in the case of viscous flow, when there was a constant stress recorded for a significant amount of time (>1 hour), or (2) in the case of brittle behaviour, a stress drop exceeding 20 % of the monitored peak stress achieved, highlighting that failure had occurred. Repeat experiments were performed on samples with a similar porosity (i.e., within 0.01 of the other sample tested) at various conditions to verify findings.

### 2.3.1 High temperature experiments

Prepared cores were placed upright in between the pistons of the press; the furnace was closed around the sample which was heated at 4 °C min$^{-1}$ to 900±3 °C (sample temperature); a K–type thermocouple was left in contact with the sample at all times and the temperatures of the top, middle and bottom zones of the furnace were monitored throughout the experiment. Following thermal equilibration for 1 hour at target temperature, the piston was then brought into contact with the sample at low load (< 30 N), and the temperature of the sample was read from the thermocouple. A stepped strain–rate experiment (at $10^{-6}$ then $10^{-5}$, $10^{-4}$ and $10^{-3}$ s$^{-1}$) was first carried out to constrain the viscous–brittle transition of the material and inform subsequent testing at unique strain rates. Tests at unique strain rates were then carried out at $10^{-3}$, $10^{-4}$, $10^{-5}$ s$^{-1}$, after which, the samples were cooled to ambient temperature at 4 °C min$^{-1}$ [note: From here, samples deformed at high temperature will be defined as lavas, and those tested at room temperature as rocks].

### 2.3.2 Ambient temperature experiments

Ambient temperature experiments were carried out on all collected sample blocks. Prepared cores were placed upright between the pistons where they underwent compressive tests at various strain rates until failure. The thermally stressed samples were tested at a strain rate of $10^{-3}$ s$^{-1}$, whereas the remaining pristine specimens were axially loaded at strain rates of $10^{-1}$, $10^{-3}$, or $10^{-5}$ s$^{-1}$ until failure (see Table 1).

### 2.3.3 Treatment of data

The strain at failure for these samples was selected using a semi–automated MATLAB script which identified the strain value at peak stress. The static Young's Modulus was computed for each experiment that exhibited a brittle response (e.g. after Heap et al. 2014a) by calculating the slope of the linear portion of the stress–strain curve via an automated script written in MATLAB and available at https://doi.org/10.5281/zenodo.1287237. To ensure that only the linear portion was selected, points within 10 % of the maximum slope were considered to define the Young's modulus for that sample (Fig. S6), minimising the potential contribution of mechanical data obtained during crack closure (during initial loading) and during strain hardening (beyond the onset of dilation).

For samples that demonstrated a viscous response, the apparent viscosity ($\eta_a$; in Pa.s) was calculated using the equation of Gent (1960) developed for the parallel–plate viscometric method, given the absence of slip along the sample/piston interfaces:

$$\eta_a = \frac{2\pi F h^4}{3V\dot{\varepsilon}(V + 2\pi h^3)} \tag{8},$$

where $F$ (N) is the applied force on the sample; $h$ (m) is the height of the sample; $V$ (m$^3$) is the initial volume of the sample, assumed constant, and $\dot{\varepsilon}$ (s$^{-1}$) is the applied strain rate.

## 3. Results

### 3.1 Sample characterisation

### 3.1.1 Mineralogy and geochemistry

Normalised geochemical analysis for bulk and glass geochemistry, obtained by XRF and EPMA respectively, are displayed in Table 2. Optical examination of the samples reveals that they consist of 20–50 vol.% phenocrysts and microphenocrysts of plagioclase (> 25 vol.%), amphibole (~ 5 vol.%), biotite (~ 2 vol.%) and quartz (~ 2 vol.%) (Fig. 3), where plagioclase and amphibole are the largest of the phenocrysts, and are generally greater than 3 mm. These phenocrysts and microphenocrysts are set in a partially crystalline (~ 50 vol.%) groundmass containing microlites of plagioclase, pyroxene, quartz, pargasite, and Fe–Ti oxides in a peraluminous rhyolitic interstitial glass (as described in Cordonnier et al. 2009). Cristobalite is occasionally observed as pore infills (also recorded by Nakada and Motomura, 1999). The bulk chemistries of samples UNZ-11 and UNZ-12 (deemed visually altered) have slightly more (1.1–1.4 wt.%) SiO$_2$ and slightly less (0.55–0.63 wt.%) CaO than UNZ-4, whilst K$_2$O and Na$_2$O concentrations are almost identical.

### 3.1.2 Rock porosities

The total porosities of the samples determined by helium pycnometry measurements range from 0.10–0.32 (Table 3); a scatter which has previously been studied in an investigation of rock frictional properties (Hornby et al., 2015) and which is consistent with field measurements of Mt. Unzen 1991-1995 eruptive products (Kueppers et al., 2005). The pores of the denser products, notably UNZ-4 and UNZ-12, are fully connected, whereas the higher porosity blocks contain a portion (0.01–0.02) of isolated pores. The small standard deviation for the connected, isolated and thus total porosity of the rocks ensures the comparability of mechanical data obtained on samples with similar porosities during repeats.

### 3.1.3 Microstructures

Microstructural examination can be used to assess any pre–existing anisotropy or fabrics in the lavas. Photomicrographs along with SEM images, of a selected group of samples (UNZ-4,-11,-12,-13) can be seen in Figure 3. These samples are shown due

to their contrasting nature, covering the span of textures studied here: UNZ-11 and UNZ-12 are visually altered samples; UNZ-13 has a different pore anisotropy than UNZ-11, and UNZ-4 is a typical product of the block–and–ash flow and is representative of the remaining samples tested. The images in Figure 3 show the original materials, orientated so that the direction of principal stress, $\sigma_1$, applied to the cores prepared of each rock would be in the vertical direction.

It is evident from Figure 3 that the pores in the Mt. Unzen dome rock samples are preferentially elongate. In some cases, the elongation has a visually preferred orientation (e.g. UNZ-11,-13), while in others it is unsystematic (e.g. UNZ-4,-12). In UNZ-11 vesicles, and microlites, appear to bottleneck around phenocrysts in a horizontal direction (i.e., perpendicular to $\sigma_1$ imposed in the experiments), whereas in UNZ-13 their alignment is vertical (i.e., parallel to $\sigma_1$), indicating a sense of shear in those directions. UNZ-4,-11, and -13 have significant number of larger pores (>1 mm) when compared to UNZ-12, and across the

shown sample set, these larger pores appear as pressure shadows around the phenocrysts (e.g. see McKenzie and Holness, 2000). Fractures are only clearly visible in UNZ-4, this is most likely due to higher abundance and larger fracture widths in this sample, allowing them to be visible in both thin section and BSE images. The fractures appear to connect pores via the tip of their major axis.

The groundmass of UNZ-12 contains a scaly–textured silica polymorph that appears to have filled vesicles. Common silica

polymorphs seen at Mt. Unzen, and other domes across the world (e.g. Mt. St. Helens; Voight et al. 1981), are cristobalite precipitates, formed from hydrothermal activity (Nakada and Motomura, 1999; Voight et al., 1981, 2009; Yilmaz et al., *in review*). This silica deposit has filled a considerable amount (~ 50 vol.%) of the vesicles in UNZ-12, reducing its porosity (Fig. 3c). Although the polymorph is a sign of alteration, in the highest magnification BSE image, some glass appears to have remained vitreous between silica polymorph areas. In UNZ-11, neither the phenocrysts nor the groundmass show evidence of

alteration (Fig. 3b,c).

In UNZ-12 the phenocrysts are visually more abundant (> 20 vol.%) than in the other specimens (Fig. 3). Nakada and Motomura (1999a) observed that groundmass crystallinity increased from (33 to 50 vol.%) with decreasing effusion rate, as was the case towards the end of the eruption (Nakada et al., 1995b), consistent with ~ 55 vol.% groundmass crystallinity in the 1994–95 spine (Wallace et al. *in review*). These observations are also consistent with the crystal fractions measured in UNZ-

12, collected from the near–vent area.

### 3.1.4 The influence of thermal stressing

The skeletal volume, mass and dimensions of each core were measured before and after thermal stressing in order to assess changes in porosity that may accompany microstructural adjustment in the process. Results showed that over the 12 cores subjected to thermal stressing, the change in connected porosity was less than 0.001, which is within the resolution of the

method. Thus, it may be said that thermal stressing did not markedly create pores or connect isolated vesicles. It did however cause a slight decrease in the values of Young's Modulus.

### 3.2 Uniaxial compressive experiments

### 3.2.1 Mechanical response of Mt. Unzen dome rocks

Uniaxial compressive strength tests were conducted on 66 cores at ambient temperature. For those samples which had a brittle

response to uniaxial compression, the failure process can be segregated into 4 stages (Hoek and Bieniawski, 1965; e.g. Brace et al., 1966; Scholz, 1968; Heap et al., 2014a). An initial build–up of stress has been attributed to the closure of micro–cracks perpendicular to $\sigma_1$, this is the initial convex segment on the stress–strain curve (e.g. Fig. 4a,b). The second, linear increase in stress and strain has been attributed to dominantly elastic (recoverable) deformation. Strain hardening, marks the onset of micro–fracturing that imparts permanent, non–recoverable damage, causing deviation from the linear elastic regime during

loading (seen as the concave section of the stress–strain curve; Fig. 4a,b). Finally, a peak in stress is reached, followed by an abrupt stress drop, this is associated with through–going fracture propagation and coalescence before macroscopic failure is reached. This behaviour is seen in the stress–strain curves (Fig. 4a,b, Fig. S3, S4) of all samples deformed in the brittle regime, be it at ambient temperature (for all strain rates) or at high temperatures (for faster strain rates; see section 3.2.2).

The strength of the rocks was observed to decrease with porosity (Fig. 5a). The range of strength of dense rocks is higher than porous rocks. We observe that rock strength increases with applied strain rates at all porosities, although this effect is more pronounced for dense rocks. The data suggest that the rocks deemed altered (UNZ-11, UNZ-12) are not weaker, but indeed stronger than pristine rocks with equivalent porosities (see circled data points in Fig. 5a).

The overlap between the datasets obtained for thermally stressed and as–collected samples suggests that thermal stressing did not impart significant damage or mineralogical changes (if any) to modify the strength of these rocks (Fig. 5). Yet, a closer look at the mechanical data suggests that the initial convex increase in stress with strain is more pronounced for the thermally stressed samples than for their pristine equivalent (Fig. 4c), indicating that the thermally stressed samples have more cracks to close than their untreated equivalents. It is therefore likely that thermal stressing has caused the creation or opening of micro–fractures, dislocating the rocks slightly in the process, but not enough to cause a notable increase in porosity or decrease in strength

### 3.2.2 Rheological response of Mt. Unzen dome lavas

The mechanical data of lavas show a wider range of behaviour than those obtained on rocks at ambient temperature (Fig. 4a,b). At slower strain rates of $10^{-4}$ and $10^{-5}$ s$^{-1}$, samples may provide a viscous response. Initially, the stress-strain curves exhibit only a mildly convex stress build–up, but then deformation is dominated by a stress relaxation phase which results in a levelling of the stress to a steady value as strain amasses (orange and red lines, Fig. 4a, 4b). This represents the viscous flow of the suspension and, as such, its apparent viscosity can be calculated from the mechanical data. At faster strain rates $> 10^{-3}$ s$^{-1}$, samples may respond brittlely. In this regime, as at ambient (room) temperature, stress accumulation may eventually lead to failure and a significant stress drop (e.g. maroon line Fig. 4b). The behaviour that links the viscous and brittle response is termed transitional. If the plateau in the stress-strain curves is marked with minor stress drops, this signals a transitional response to deformation that is dominantly viscous. Where there is a major drop in the stress-strain curve that takes place over an extended period of strain, we termed this response brittle–dominated transitional (maroon line Fig. 4a). This interpretation of mechanical data can also be confirmed by analysing the microstructure of deformed samples (Fig. 6). Samples with pervasive macro–fractures that propagate through both groundmass and, to a lesser extent, the phenocrysts have likely undergone fully brittle deformation. Likewise, samples with a response classified as brittle–dominated transitional also have pervasive macro–fractures, however the phenocrysts are only slightly displaced along their cleavage planes, rather than shattered. Samples that have had a viscous–dominated transitional response to strain display microfractures in both the groundmass and phenocrysts, and those that have had a viscous response show little to no micro–fracturing. A viscous response may also lead to elongation of porosity parallel to the sense of shear.

The evolution of apparent viscosity is strain–rate dependent as shown by the stepped strain rate experiment (Fig. 7). An increase in the strain rate resulted in an order of magnitude decrease in viscosity– a thixotropy of similar magnitude as that described for highly crystalline magmas in Lavallée et al. (2007). In this experiment, deformation at low strain rates of $10^{-6}$ s$^{-1}$, $10^{-5}$ s$^{-1}$ and $10^{-4}$ s$^{-1}$, is marked by a non–linear increase in apparent viscosity (upon stress relaxation) and plateauing to a constant value for each strain–rate step; this value decreased with increase of the applied strain rate (Fig. 7). Deformation was pervasive (i.e., ductile), which, being above $T_g$, suggests that it may have dominantly occurred via a viscous response. At $10^{-3}$ s$^{-1}$, however, the apparent viscosity plummeted as the sample underwent failure along a localised fault, evidencing a transition into the brittle regime (Fig. 7b).

At strain rates of $10^{-3}$ s$^{-1}$ the sample suite tested reached peak stresses of ~ 20–80 MPa (Fig. 8a) and strength decreased inversely with porosity. Here the samples responded with a brittle and brittle–dominated transitional response to strain. The mechanical responses of samples tested at high temperature were more repeatable than those carried out at ambient temperature: the strength of samples (within a family with ~ 0.01 porosity range) was within ~ 2 % of each other at low porosities (< 0.20) and within ~ 5 % of each other at high porosities (> 0.20), whereas at ambient temperatures a variation of ~ 60 % is observed in the lower porosity regime (Table 1).

When a strain rate of $10^{-4}$ s$^{-1}$ was applied some of the samples reached peak stresses between ~ 10 and 35 MPa (Fig.7b), before relaxing the stress through substantial strain. Here, the lavas display a viscous and viscous–dominated transitional response to strain. In some samples, an initially viscous response transitioned to fracturing after a certain amount of strain, leading to macroscopic failure. Samples that did not fracture continued to flow viscously with increasing strain, with a component of strain hardening, similar to that seen by Kendrick et al. (2013b). Samples that were subjected to a strain rate of $10^{-5}$ s$^{-1}$ had a fully viscous response over the strain rates tested (Fig. 8c). Remarkably, the peak stresses of samples tested at $10^{-4}$ and $10^{-5}$ s$^{-1}$ were seemingly independent of porosity (Fig. 8d).

The apparent viscosities calculated from the responses at $10^{-5}$ and $10^{-4}$ s$^{-1}$ show an initial increase (due to relaxation in the first 0.7 % strain) and levelling to within a narrow range (see Fig. 9a,b). For a given strain rate, we note a small range of apparent viscosities, but importantly, no systematic change in viscosity as a function of sample porosity (within the range tested; Fig. 9c).

These results indicate that the transition in deformation mode from macroscopically ductile to brittle behaviour is straddled by our experiments in the range $10^{-5}$ to $10^{-3}$ s$^{-1}$.

## 4. Interpretation of dome rock mechanics

### 4.1 Mechanical responses of rocks and lavas in the brittle and brittle–dominated transitional regime

The experimental findings presented here suggest that the mechanical response of lavas and rocks is similar, but important differences remain. Experiments carried out on rocks at ambient temperature (all strain rates), and on some lavas at strain rates of $10^{-3}$ s$^{-1}$, resulted in brittle behaviour. However, there are significant differences in the mechanical response between the two (Fig. 4). (1) We noted a shorter convex portion at the onset of the stress–strain curve of tests at high temperature (Fig. S3), which we attribute to a narrowing of pre–existing cracks at high temperature (due to thermal expansion of the materials with heating; e.g. Fig. S5), resulting in a smaller extent of crack closure during initial loading; (2) most high temperature samples have a shallower linear portion of stress–strain build–up, which we hypothesise may reflect a contribution of viscous deformation upon loading, leading to a brittle–dominated transitional classification; and (3) we observed a less angular concave down portion of the stress–strain curve, which we attribute to more pervasive deformation (as seen by longer strain to failure; Fig. 10) and micro–fracturing leading to failure. The exception to these findings is in the highest porosity sample, UNZ-7, where there appears to be no significant change in shape between high and ambient temperature experiments (see Fig. S3, S4). This sample was classified in the fully brittle regime. It remains that at higher temperature, lavas are stronger (by 10–40 MPa; Figs 4–9) than their rock equivalents at ambient temperature. Before delving in their differences (section 4.1.4), we will first interpret the results on the strength (section 4.1.2) and Young's Modulus (section 4.1.3) of porous rocks at ambient temperature.

#### 4.1.2 The effect of porosity on material strength

From the results of the uniaxial compressive experiments it is evident that porosity is a major control on the strength of dome materials. Previous studies on volcanic rocks (Al-Harthi et al., 1999; Heap et al., 2014a, 2014b, 2016b; Schaefer et al., 2015) have found a similar correlation in which. to a first order, strength is inversely proportional to the porosity of the rock.

Here, the strength of samples with higher porosities display less scatter than those with lower porosities (Fig. 10a). Microstructural examination of the samples (Fig. 3) reveals the porosity of the porous specimens to be dominated by vesicles, whereas the porosity of the denser samples is dominated by microfractures, which may define a change in the microstructural control on the strength and failure of low and high porosity samples. In these lower porosity specimens, the non–systematic orientation of microfractures could be responsible for the large scatter in strength. The uniaxial compressive strength was calculated for the samples for both the pore–emanating crack model of Sammis & Ashby (1986) (Eq. 3) and the sliding wing crack model of Ashby & Sammis (1990) (Eq. 4). For the former, the uniaxial compressive strength was calculated with varying values of $\frac{K_{IC}}{\sqrt{\pi r}}$ from 5 MPa to 25 MPa (Fig. 11). For the latter, approximate values for $\mu, \frac{K_{IC}}{\sqrt{\pi c}}$ and $D_0$ were taken from Table 3 in Paterson and Wong (2005) as 0.51, 20–30 MPa and 0.3–44, respectively. This gave a range of estimated strength between 54 and 90 MPa (Fig. 11). At higher porosities, $> 0.25$, the pore–emanating crack model with $\frac{K_{IC}}{\sqrt{\pi r}} = 5$–10 MPa seems to fit the data well, whereas for most rocks with porosities of 0.12–0.2 $\frac{K_{IC}}{\sqrt{\pi r}} = 10$–15 MPa is a better fit. This could be explained by a decrease in the pore radius at these porosities, leading to higher values of $\frac{K_{IC}}{\sqrt{\pi r}}$, though, as the samples are heterogeneous and pore radius variability is high we cannot observe this (Figure 3). For the densest rocks in the study (~0.08–0.12), the UCS data would suggest yet a higher $\frac{K_{IC}}{\sqrt{\pi r}}$ of 20–25 MPa. The pore–emanating crack model could explain this switch in behaviour if there was a fundamental change in pore radius. However, the switch could also be explained by a transition in failure mechanism from pore–emanating cracks to wing cracks, meaning the wing–crack model would be more applicable. Alternatively, it may be a complex combination of the two. Although the solutions to the sliding wing–crack model are non–unique, as there are few experimentally constrained parameters, when combined with information gained from the pore structures (Fig. 3), the results of the modelling presented (Fig. 11) give us an insight into the dominant micromechanical failure mode of our samples.

It is likely that the complex pore structures of these lavas, generated by a combination of vesiculation, deformation and cooling-driven contraction require an as-yet undefined combination of the two models. The weighting towards one or the other, however indicates that for the higher porosity specimens the behaviour of failure could be described using the pore–emanating crack model of Sammis & Ashby (1986), whereas in the lower porosity samples deformed in uniaxial compression, the main failure mechanism is explained by the sliding wing–crack model of Ashby & Sammis (1990).

This transition in the preference of fracture nucleation site from pore to crack is likely to be gradual and dependent on the pore network architecture of a suite of samples; in these Mt. Unzen samples it is found at a porosity of ~ 0.2. Other studies have also alluded to such a transition when studying permeability, finding a transition from crack–dominated to pore–connectivity–dominated regime of fluid flow at values of ~ 0.14 (Farquharson et al., 2015), 0.155 (Heap et al., 2015), 0.105–0.31 (Kushnir et al., 2016), ~ 0.15 (Eggertsson et al., 2018), and 0.11–0.18 (Lamur et al., 2017).

Samples UNZ-11 (porosity: 0.30) and UNZ-13 (porosity: 0.32) both have elongated vesicles. The cores were cut so that the vesicles were either perpendicular or parallel to the applied principal stress, $\sigma_1$, for UNZ-11 and UNZ-13, respectively (Fig.3). The porosities of the two rocks are comparable, and there is no great difference in strength, indicating that pore orientation may not have a significant influence on strength within dome rocks. Although we do note that UNZ-11 undergoes a higher strain to failure (Fig. 10b) and thus lower Young's Modulus (Fig. 10c) than UNZ-13, indicating that it is less stiff. Sample UNZ-2 (porosity: 0.13) however, does have a remarkably larger uniaxial compressive strength (~20 MPa) and Young's Modulus (~5–10 GPa) than samples of similar porosity. This may be due to the high number of spherical isolated pores (Table

3,Fig. S2) which act as rigid bodies. However, it cannot be explicitly stated that pore anisotropy did not play a role in this and thus it is possible that the orientation of a pore may have a dominant effect on the  strength and stiffness of the dome rock (Bubeck et al., 2017; Griffiths et al., 2017) Thus, future studies on rock strength may benefit from an in–depth study of rock strength as a function of pore fraction, orientation and connectivity.

### 4.1.3 Static Young's Modulus

At ambient temperatures, the static Young's modulus decreases from > 15 GPa to < 5 GPa with increasing porosity (Fig. 10c). This is an indication that samples with lower porosities were stiffer than those with higher porosities. However, there were outliers to the data trend, UNZ-13 and UNZ-2 (with average porosities of 0.32 and 0.13, respectively) are stiffer and have higher (> 5 GPa) Young's Moduli than other rocks with similar porosities (see Table 1); in UNZ-13, this may be explained by the preferred orientation of pores parallel to the principal stress (Figure 3a) (cf, Griffiths et al., 2017). The naturally altered samples, tested at similar conditions, exhibited Young's Moduli trends like those of comparable fresh rocks (Fig. 10c).

Lavas deformed at 900 °C. at a strain rate of $10^{-3}$ s$^{-1}$ have systematically lower (~ 5–10 GPa) Young's Moduli. It is this malleability that allows the lava to be deformed to higher strains before macroscopic failure (Fig. 10b), an observation recognised in Schaefer et al., (2015) in tests on basaltic lavas.

In addition, thermally stressed samples have slightly lower (~ 0.5–1.5 GPa) Young's Moduli than their unstressed equivalents, as previously noted in dacites from Mt. St. Helens (Kendrick et al., 2013a) and andesites from Colima volcano (Heap et al., 2014a). The slight decrease in static Young's modulus with thermal stresses highlights a potential change in porosity distribution that was not recognised by other means (e.g. total porosity, strength).

### 4.1.4 The effect of temperature on sample strength

Remarkably, when in the brittle regime at high temperature, samples exhibited strengths ~ 10–40 MPa greater than at ambient temperature. This may be attributed to the way the samples respond to stress at higher temperatures. First, upon heating a rock, it expands, which may partially close pre–existing micro–fractures, thus modifying the resultant elastic response of the material (see section 4.1.1). Moreover, at 900 °C the presence of interstitial melt in a sample allows for considerably more strain than if it were deformed at ambient temperature (when in a solid, glassy state). The initial strain upon loading would be accommodated by both an instantaneous and a delayed elastic response (e.g. Dingwell and Webb, 1989) and perhaps minor micro–crack closure (e.g. Heap et al., 2014a), before the onset of viscous (e.g. Lavallée et al., 2007) and crystal plastic (e.g. Kendrick et al. 2017) deformation that results in permanent strain (and barrelling of the sample). Thus, at higher temperatures, more strain is accommodated upon loading than at ambient temperature (Fig. 7a), leading to higher strain to failure (Fig. 10b) and lower Young's Moduli than their rock counterparts (Fig. 10c). The Young's Moduli for lavas undergoing failure at high temperature are rate–dependent, perhaps as they may undergo further stress dissipation by viscous relaxation in the melt.

A similar increase in strength with temperature was also noted in basaltic rocks from Pacaya volcano (Schaefer et al., 2015). There, the authors attributed the increase in strength of the glass–poor rock to the closure of micro–cracks (likely formed upon cooling after their eruption) due to thermal expansion, a process that equally occurs in Mt. Unzen dome rocks. Rocks may also become weaker from thermal stressing, this can be due to crack initiation (Heap et al., 2016a), or alteration, via processes such as decarbonation and dehydroxylation (Heap et al., 2012, 2013a, 2013b). A recent study by Eggertsson et al., (2018), found that samples that hosted microfractures (like Mt. Unzen dome rock) were not affected by thermal stressing, while those that showed a trivial fraction of pre–existing micro–fractures were more readily fractured through thermal stressing and as a result became more permeable.

## 5. Rheology of dome lavas

### 5.1 Viscosity of dome lavas

The style of an eruption – effusive vs explosive – depends on the rheological response of magma (Dingwell, 1996). The urge to understand the alarmingly variable nature of volcanoes, and recent advances in experimental capabilities and computational modelling, have encouraged the community to focus efforts on the development of two and three–phase models of magma rheology (e.g. Lejeune and Richet, 1995; Caricchi et al., 2007; Lavallée et al., 2007; Costa et al., 2009; Mueller et al., 2011b; Truby et al., 2015). Truby et al., (2015) combined two, two–phase flow models (considering melt and crystals, and melt and gas bubbles) to elaborate a three–phase model of magmatic suspensions, further tested against a set of controlled analogue laboratory data. Their model shows that while the addition of crystals increases the viscosity of a suspension, leading to a shear thinning rheology, the addition of gas bubbles (which can deform during shear) has variable consequences. Depending upon the initial crystal volume and maximum packing fraction of those crystals, the addition of gas bubbles may result in a further increase in viscosity or, in other cases, a levelling or a decrease in the apparent viscosity of the suspension. Their model suggests that the addition of bubbles to lavas, above their glass transition, with high normalised crystal fractions, like those seen in volcanic domes, would likely decrease the viscosity of the suspension. However, here, the data show that the presence of vesicles (between 0.09 and 0.33) in dome lavas may not necessarily influence the apparent viscosity (at least not systematically). We advance that this could be due to the high connectivity of the pores present in dome lavas, which allows efficient outgassing, thus the gas cannot act as an isolated phase that can pressurise during shear. Thus, it may be that lavas hosting permeable porous networks may have mostly porosity–independent apparent viscosities (at least across the range examined here), as suggested by Lavallée et al. (2007). Current models relating porosity to viscosity, simply account for the presence of isolated gas bubbles via a capillary number, to calculate the apparent viscosity of a multi–phase suspension (e.g. Rust and Manga, 2002; Llewellin and Manga, 2005; Truby et al., 2015). However, this result highlights important shortcomings to the modelling of shallow magmas, where porous networks tend to develop connectivity, especially in sheared crystal–bearing lavas (e.g. Laumonier et al., 2011; Kushnir et al., 2017). This connectivity controls outgassing, and thus pressure build–up or release, which is responsible for rheological variations in magma and therefore eruption style (effusive vs explosive). Our findings suggest that we need to revise three–phase models to account for gas flow through evolving, deformable bubbles, that may also be connected, in order to constrain the apparent viscosity of magmas in lava domes and other open–system settings.

### 5.2 Failure criterion for porous lavas

During magma ascent, the strain rate, which is proportional to effusion rate (e.g. Goto 1999), plays a key role in determining whether the response of magmas and extruding lavas is that of a solid, or liquid (Webb and Dingwell, 1990). Here, the macroscopic deformation mode (viscous, viscous–dominated transitional, brittle–dominated transitional or brittle) of lavas was characterised based on their resulting stress–strain curve (section 3.2.2; Fig. 12a); these are further supported by micro-structural observations (see Fig. 6; Fig. 12a). [NOTE: sample UNZ-4-28 was not given a classification as its response to deformation was likely an experimental artefact due to a chipping of the sample edge]. The distinction between these rheological regimes can be made using the Deborah number (Eq. 2). In a recent study on the failure of single phase silicate melts, Wadsworth et al., (2017) suggest that fractures can propagate above $De \geq 10^{-2}$ when a sample begins to undergo brittle deformation, although these fractures are often blunted by viscous relaxation. When $De \geq 1$ brittle behaviour dominates over viscous deformation and violent rupture of the sample ensues. This dimensionless ratio of the relaxation timescale of the melt (Eq. 1) and the observation timescale can be rewritten as:

$$De = \frac{\eta_m}{G_\infty t_{obs}} \tag{9}$$

where the observation time, $t_{obs}$, is the inverse of the strain rate of magma deformation, $\dot{\varepsilon}_{obs}$. Thus Eq. 9 can be rewritten as:

$$De = \frac{\dot{\varepsilon}_{obs}\eta_m}{G_\infty} \tag{10}$$

Magmatic suspensions, like those described in this study, are non–Newtonian materials with a shear thinning response (Caricchi et al., 2007; Lavallée et al., 2007; Cordonnier et al., 2009; Avard and Whittington, 2012; Vona et al., 2013), hence their viscosity is strain rate dependent. It has previously been described that the peak stress, $\sigma$, shares a power law relationship with strain rate, $\dot{\varepsilon}_{exp}$, via:

$$\sigma = k\dot{\varepsilon}_{obs}^{\ b} \tag{11}$$

where $k$ is the flow consistency index (in Pa.s) and $b$ is the flow behaviour index, describing the rheology of the fluids (Ostwald, 1925; Lavallée et al., 2007). For Newtonian bodies $b = 1$, but for shear thinning suspensions, $b$ decreases below 1 (Caricchi et al., 2007) and reaches a minimum of $b = 0.5$ for crystal–rich materials (Lavallée et al., 2007; Cordonnier et al., 2009). In the present study the Mt. Unzen dome material tested at 900 °C, by fitting a power law to the peak stress–strain curve we obtained Ostwald constants of $k = 1653$ and $b = 0.5$ (Fig. 12b). So, we can rewrite Eq. 10, using Eq. 11, to obtain:

$$De = \frac{(\sigma/k)^{1/b}\eta_m}{G_\infty} \tag{12}$$

which permits the representation of the Deborah number of material failure as a function of strength (which was shown to be dependent on porosity), for a given temperature (and thus interstitial melt viscosity). For our samples, the interstitial melt viscosity can be estimated at $10^{9.42}$ Pa.s (using its chemistry and experimental temperature as an input parameter in the GRD viscosity calculator (Giordano et al., 2008)). In Figure 12c, we present the data using symbols that illustrate the response of 620 the samples. The onset of transitional behaviour, termed viscous–dominated transitional, is marked by the red line. Similarly, the onset of brittle behaviour, brittle–dominated transitional, is marked by the yellow line. These lines are linear regressions on a semi–log space plot, with their standard error of estimates marked by faded colour windows. Any point that plots between the red and yellow lines would be termed transitional and could demonstrate any type of hybrid behaviour. Above a porosity of 0.27 no transitional zone occurs, and behaviour would be classified as either viscous or brittle. This analysis demonstrates 625 that the critical Deborah number, $De_c$, which indicates the initiation of rupture, in dome lavas from Mt. Unzen decreases by just over half an order of magnitude over a 0.35 range in porosity; from $\sim 7.65\times10^{-5}$ in the densest sample measured to $4.1\times10^{-5}$ in the most porous, following the trend: $De_c = -1.7\times10^{-4}\varphi + 9.40\times10^{-5}$ (Fig. 12c). Such a magnitude is proportional to the strength decrease of material as a function of porosity (see Fig. 10a and Paterson and Wong, 2005 for a discussion), and thus relates the porosity to the ability of high temperature lavas to rupture. By extrapolating the trend and finding the $De_c$ for a 630 hypothetical, pore–free Mt. Unzen sample, we can compare our results to a two–phase (crystals and melt) model for rupture (Wadsworth et al., 2017). Given that the Mt. Unzen material has a crystal content (microlites and phenocrysts), $\phi_x$, of $\sim 0.75$, the bulk $De_c$ can be modelled via:

$$De_c = De_{cx}\left(1 - \frac{\phi_x}{\phi_m}\right) \tag{33}$$

where $De_{cx}$ is the critical Deborah number for a crystal and bubble free melt, $10^{-2}$, and $\phi_m$ is the maximum packing fraction 635 of the system.

For the Mt. Unzen material $\phi_m$ can be assumed to be in the range of $\sim 0.76 - > 0.99$, as it is clear from microstructural analysis that our material has not yet reached $\phi_m$ (see Fig. 3) [maximum packing is defined geometrically as the volume fraction at

which there is no space remining for further particles (Mader et al., 2013)]. This gives a modelled $De_c$ in the range of $\sim 1 \times 10^{-4}$ and $7.6 \times 10^{-4}$ which is in line with the $De_c$ found by the linear extrapolation of experimental results, $9.4 \times 10^{-5}$ for the onset of rupture and $6.6 \times 10^{-4}$ for full rupture (Fig. 12c).

Thus both, the addition of crystals (as seen by the fact that $De_c$ of dense dome lavas is reduced by over one order of magnitude compared to that suggested by Dingwell and Webb, (1990)) and vesicles (as shown by the above equation) contributes to an increased brittleness of lava during ascent and eruption at lava domes, and in many other eruptive scenarios.

## 6. Implications for volcanic scenarios

The findings observed here help constrain the impact of rheological evolution on lava domes as they erupt and cool following emplacement. The rheology of magma has a fundamental influence on the style of a volcanic eruption, be it explosive or effusive (Dingwell, 1996; Gonnermann and Manga, 2007). Understanding how magmas respond to changes in petrology, stress and eruptive shearing conditions that occur during ascent in a volcanic conduit may help to enhance models that aim to predict volcanic activity. The work undertaken here constrains the material behaviour of erupting dome lavas and the relics that remain once the lava cools.

As magma crystallises, its apparent viscosity (generally) increases as the melt evolves, and an increasing fraction of the suspension becomes solid (with slower diffusivity and lower rate of plasticity than the viscous liquid melt), thus the suspension becomes increasingly solid–like. For crystalline magmas, we would expect $De_c$ to be lower than that for silicate liquids (*i.e.*, $De_c < 10^{-2}$; e.g. Gottsmann et al., (2009)). Cordonnier et al., (2012a, 2012b) constrained the failure of silicate liquids with different crystal fractions, and they indeed showed that $De_c$ decreases when crystallinity increases. They suggest that $De_c$ linearly decreases from $10^{-2}$ to $2 \times 10^{-3}$ between 0 and 60 vol.% crystals. However, the viscosity used to estimate Maxwell's relaxation rate in the $De$ analysis was based on the suspension's apparent viscosity rather than the interstitial melt viscosity. To constrain how the addition of crystals shifts the onset of failure of a material whose rheology is well known it is advantageous to consider the pure melt. Given this, an even larger decrease of $De_c$ would be observed (perhaps down to $\sim 9.4 \times 10^{-5}$ as constrained by failure of our densest lavas). Since the strength of material is known to be strongly influenced by the presence of pores (commonly vesicles in volcanic materials) and micro–fractures (e.g. Paterson and Wong, 2005 for a review of material properties in the brittle field), here we demonstrate that the addition of porosity to magma shifts failure to lower strain rates; thus, under constant ascent conditions, magma may undergo failure simply by vesiculation, without the need for any increase in strain rate.

Upon extrusion, lava cools, contracts and fractures (Lamur et al., 2018). Here we show that the strength of a dome is reduced upon cooling due to contraction and micro–fracturing, leaving a weaker relic structure. This situation may favour the progressive creep of cooling dome structures, as observed in lobe 11 at Mt. Unzen (Kohashi et al., 2012).

Post–emplacement, through time and prolonged exposure to corrosive fluids, dome material may alter (Ball et al., 2015). In this study, the altered rocks tested showed a higher strength than pristine rocks with equivalent porosities. However, previous studies have found that altered volcanic rocks can also be weaker (e.g. Pola et al., 2012). From this distinction we surmise that the structure of the rocks as well as the type of alteration (developing under different conditions in cooling volcanic rocks) may have contrasting effect on the strength of cooled dome lavas. Thus, the data shown here begs for an increased focus on the impact of alteration on volcanic rock strength for improved lava dome structural stability models.

The rate of deformation imposed on dome materials is also an important variable to be considered. In this study, and in others (e.g. Schaefer et al., 2015; Lavallée et al., 2018), volcanic rocks have been shown to withstand higher stresses when deformed at higher strain rates. Previous studies have suggested earthquakes with high ground acceleration have provoked lava dome collapse (Voight, 2000), therefore, it is essential to understand the effect of strain rate on the strength of materials. This is of particular importance for Mt. Unzen as it is located in a very seismically active area. Slow, continuous strain (or recurring

stressing cycles) can induce fatigue in a material and promote brittle creep (e.g. Heap and Faulkner, 2008; Heap et al., 2009; Brantut et al., 2013; Kendrick et al., 2013a; Schaefer et al., 2015) thus weakening the rocks which undergo failure at lower stresses. Thus, over long periods (years) of deformation, such as for lobe 11 at Mt. Unzen, the actual strength of the dome rocks may be lower than those reported here at the lowest strain rate of $10^{-5}$ s$^{-1}$). Time–dependent deformation can importantly contribute to catastrophic collapse of volcanic structures (e.g. Mt. St. Helens, Reid et al. 2010). Here we advance that it is crucial for future failure models of volcanic materials to incorporate the effect of strain rate.

Volcanic structures are made of heterogeneous rocks and lavas, with intricate mineralogical assemblages, textures and fabrics, with variable degrees of coherence; thus, their mechanical response may vary widely. Although here we have only tested material from the 1991–1995 eruption of Mt. Unzen, this study has the potential to be applied to other dome–forming volcanoes of similar composition, crystallinity, and porosity. Additionally, the work can also be applied to parts of larger volcanic edifices dominantly constructed by the accumulation of lavas, which may be prone to collapse (Ball et al., 2015). The work presented here can help constrain the behaviour of lavas and rocks involved in lava dome eruptions. We anticipate that the results will form the basis for more advanced numerical simulations of dome eruption and related hazards.

## 7. Conclusion

Uniaxial experiments carried out at ambient and high temperature (900 °C) on a suite of natural lavas from Mt. Unzen have given significant insight into the behaviour of lava domes, both during extrusion and after emplacement. Ambient temperature experiments allowed for the investigation of brittle behaviour, and results from these experiments can be applied to cooling domes (and the relics that they leave in the record) allowing the development of volcanic edifice failure models. Conclusions drawn from experimentation are as follows:

1. In the brittle regime, strength decreases with increasing pore volume both at ambient and high temperatures;
2. Magmas deformed in the brittle regime at high temperature are stronger than rocks of equivalent porosity deformed at ambient temperature;
3. Thermal stressing did not affect the strength of dome rocks within the conditions tested (< 900 °C and 4 °C min$^{-1}$), it did however change the morphology of the stress strain curve, indicating the widening of cracks;
4. The presence of alteration may have variable effects, sometimes strengthening volcanic rocks;
5. The strength of rocks and lavas (in the brittle field at high temperature) increases with strain rate;
6. The viscosity of dome lavas decreased with strain rate (shear thinning) and did not vary for the range of material crystallinity and porosity studied;
7. Lavas deformed at high temperature and strain rates of > $10^{-4}$ s$^{-1}$ becomes increasingly brittle, and adopt fully brittle response above $10^{-3}$ s$^{-1}$; and
8. The critical Deborah number, $De_c$ of dense dome lavas was found to be ca. $1\times10^{-4}$. It decreases with porosity according to a linear relationship.

These results reveal that current stability models of cooling lava domes, like that of lobe 11 at Mt. Unzen, require an integration of the complex nature of the materials. The outcome of this study suggests that, as a primary control on rock strength, porosity heterogeneities must be included when modelling failure mechanisms. As secondary controls, it would also be beneficial to include deformation conditions such as temperature and strain rate. Conclusions drawn from high temperature experiments suggest that current three–phase models may not be fully applicable to dome lavas and other crystal–rich lavas. We suggest a new formulation of the Deborah number that applies to porous, crystal–rich lavas and propose that it may help refine the accuracy of models attempting to describe rheological evolution to explain geophysical data monitored during lava dome eruptions.

**Data availability**

Supplementary data are available in the Supporting Information "S1" to "S6". The script for the Young's Modulus calculation is freely available on Github (https://doi.org/10.5281/zenodo.1287237). Further information can be obtained on request from the corresponding author.

**Author contribution**

YL and JEK designed the experiments. RC, AJH, TM, JEK, PAW, and JDA carried out the mechanical experiments and
725 physical measurements. PAW collected microprobe data and conducted softening point determination. RC wrote the processing codes, analysed the data, and prepared the manuscript with contributions from all co–authors. All authors contributed to the collection and selection of samples.

**Competing interests**

The authors declare that they have no conflict of interest.

**Acknowledgements**

RC, JEK, PAW, AJH, TM, JDA, and YL acknowledges funding from the European Research Council (ERC) Starting Grant on Strain Localisation in Magma (SLiM; No. 306488), JEK was supported by an Early Career Fellowship of the Leverhulme Trust. PAW acknowledges the NERC ATSC training for providing time to conduct EPMA measurements. Fieldwork was funded by the DIAWA Anglo–Japanese Foundation (grant number: 11000/11740). The authors would like to thank Prof.
Hiroshi Shimizu for his guidance throughout the study. The AST14DEM used in Fig. 1 was retrieved from the online Data Pool, courtesy of the NASA Land Processes Distributed Active Archive Center (LP DAAC) and the NASA the Japan Ministry of Economy, Trade and Industry (METI), (https://lpdaac.usgs.gov/dataset_discovery/aster/aster_products_table/ast14dem_v003#tools)

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

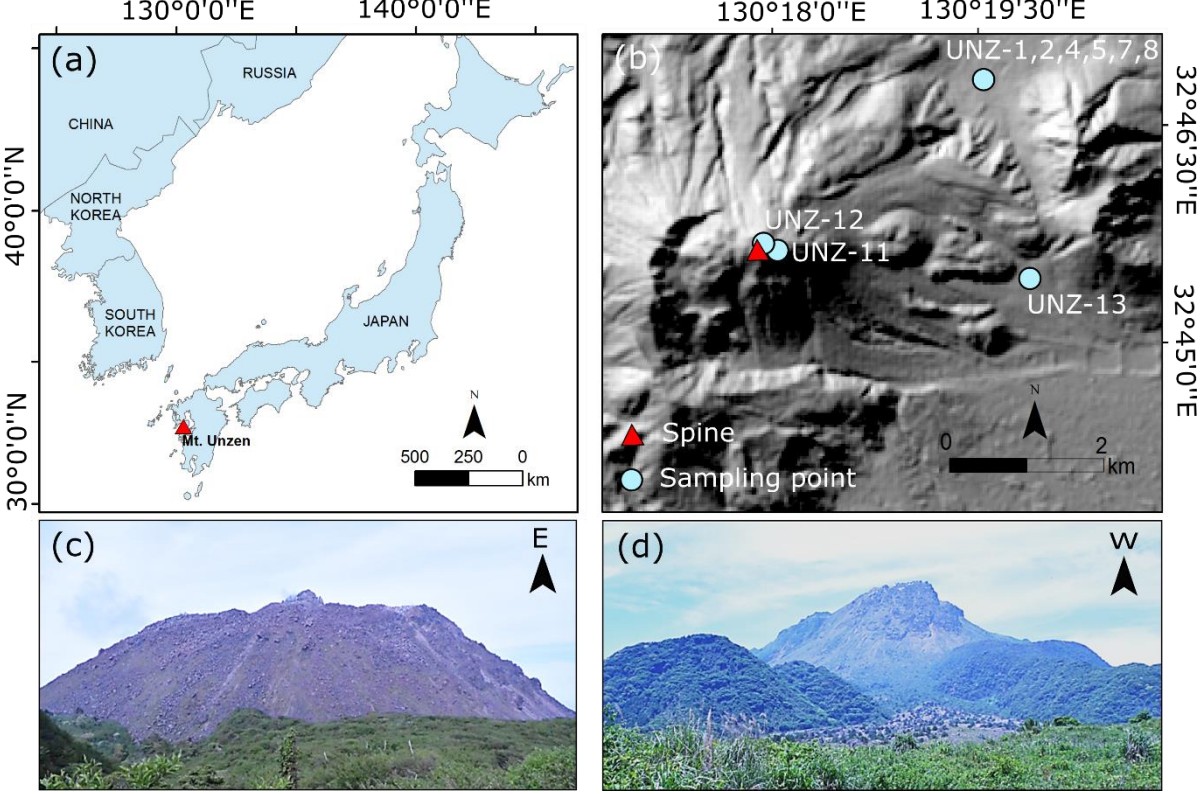

**Figure 1. (a) Location of Mt. Unzen in South Western Japan; (b) Sample collection locations and location of the erupted spine, the summit of Mt. Unzen at 1500 m above sea level (NASA/METI/AIST/Japan Spacesystems, 2001); view of Mt. Unzen lava dome looking East ~ 0.62 km from the spine (c) and West ~ 3.87 km from the spine (d) in 2016.**

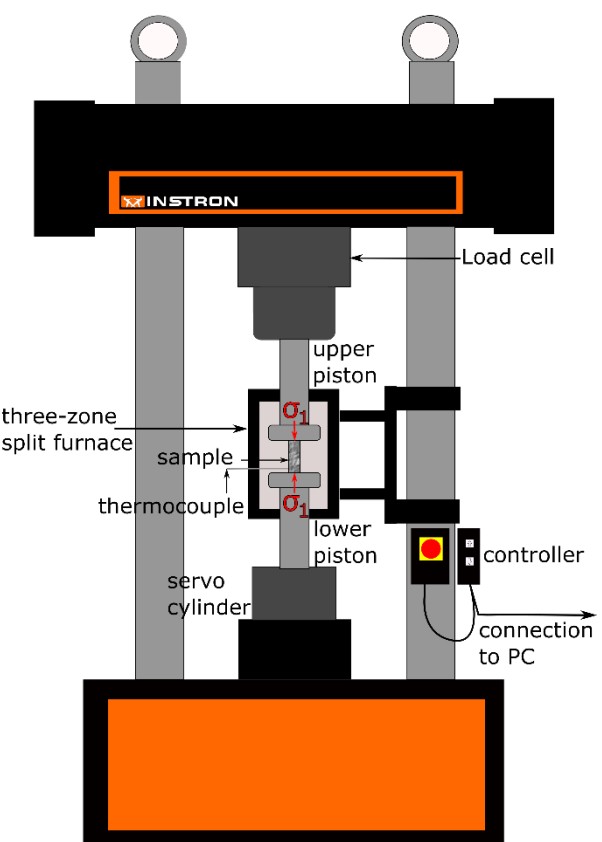


**Figure 2. Schematic of the uniaxial compressive strength testing set–up in the Experimental Volcanology and Geothermal Research Laboratory at the University of Liverpool. A 100 kN Instron 8862 uniaxial press with a three–zone, split cylinder furnace was used to perform experiments at varying strain rates and temperatures.**

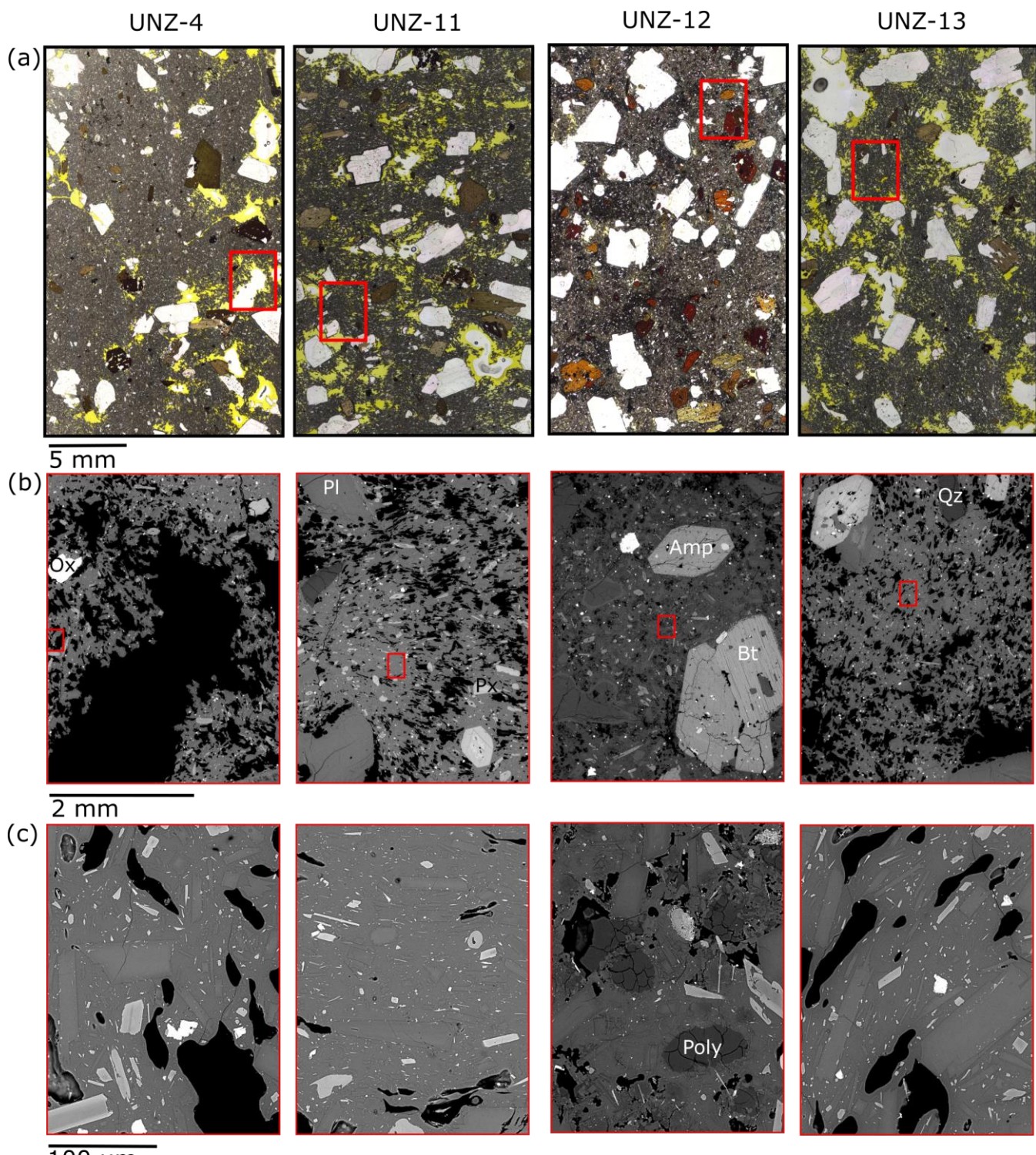

**Figure 3. Plane–polarised light (a) and backscattered electron images (b,c) of undeformed samples UNZ-4,-11, -12 and -13. (b) is a zoom into the red box in (a), and (c) is a zoom in of the red box in (b), displaying the groundmass textures. Amp: Amphibole, Bt: Biotite, Ox: Oxides, Pl: Plagioclase, Px: Pyroxene, Qz, Quartz, Poly: Silica polymorph. Images are orientated so that the later applied principal stress, $\sigma_1$, is in the vertical direction. [Note the scale that is below each set of images.]**


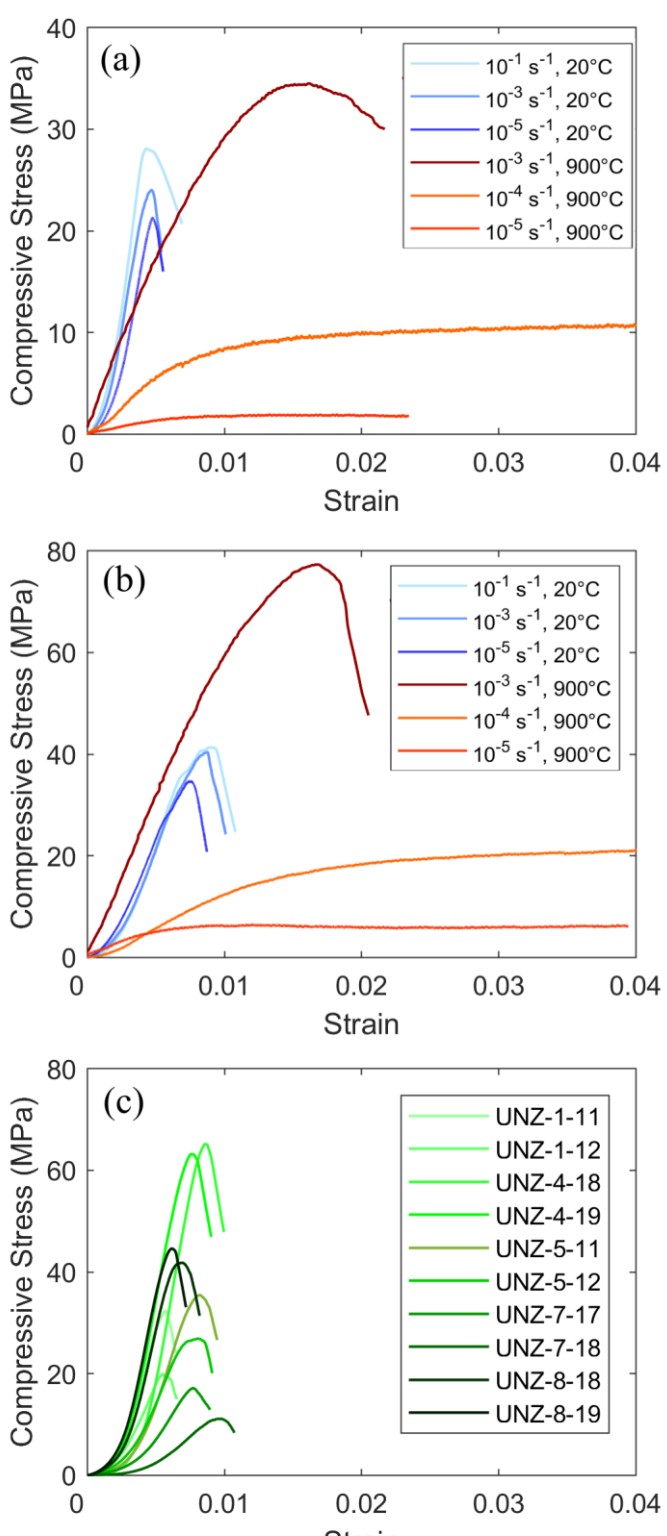

Figure 4. Examples of compressive stress–strain curves for (a) high porosity, UNZ-1 (0.21); (b) low porosity, UNZ-4 (0.12) at a range of rates and temperatures and (c) thermally stressed samples, all performed at a strain rate of $10^{-3}$ s$^{-1}$. Mechanical data for high temperature experiments are shown in shades of red, low temperature experiments in shades of blue and thermally stressed experiments in shades of green. At high temperature, faster strain rates cause the sample to break whereas at slower strain rates the sample flows. Brittle high temperature experiments fail at considerably higher peak stresses than those performed at ambient temperatures. In the brittle regime, samples deformed at faster rates failed at higher stresses. [Note: there is a difference in Y–scale between (a) and (b) & (c)]

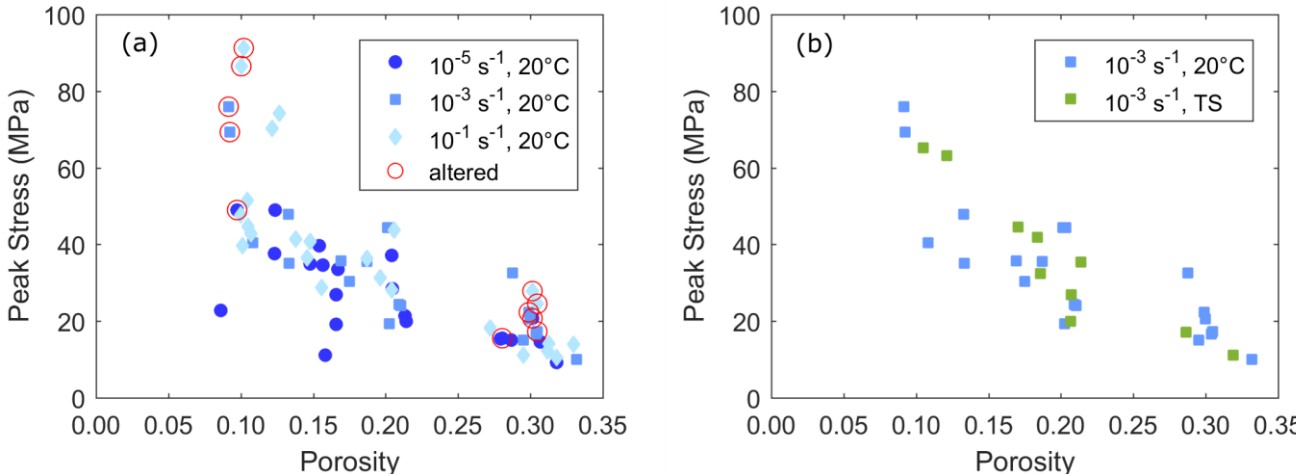

**Figure 5. (a)** The strength (peak stress) of samples tested at ambient temperatures at varying strain rates, highlighting the apparent strengthening of materials deformed at faster rate. Red rings circle the samples that are visibly altered. **(b)** A comparison of samples that were thermally stressed and those that were not, both tested at ambient temperatures and strain rates of $10^{-3}$ s$^{-1}$, demonstrating that there is no change in strength as a function of porosity due to thermal stressing.


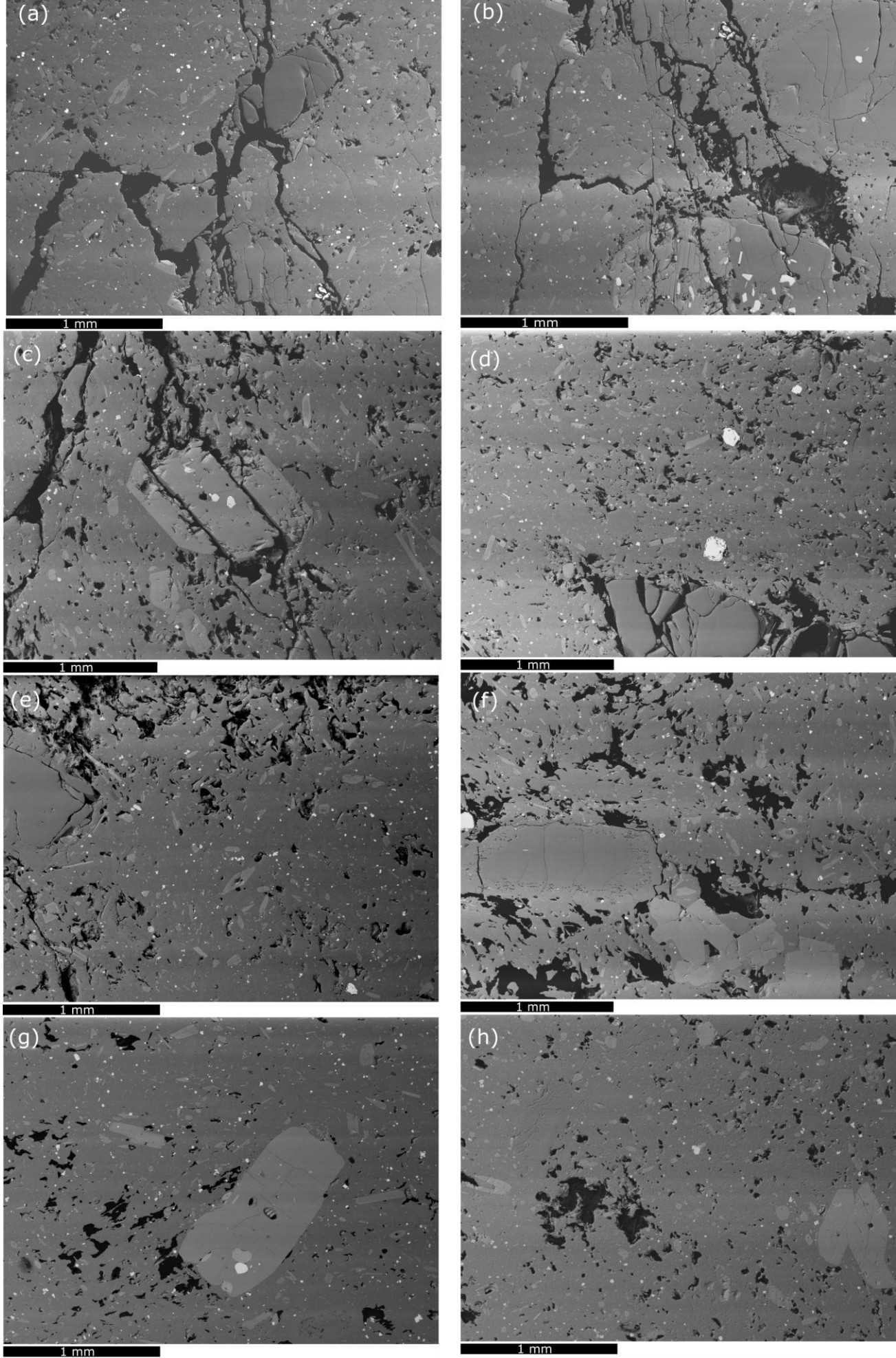

**Figure 6.** Backscattered electron images of polished stubs for samples after strain (a to f) and before strain (g and h). Panels a) and b) show sample UNZ-4-14 after experiencing brittle deformation at a strain rate of $10^{-3}$ s$^{-1}$; macroscopic cracks (> 100 μm in width) propagate through both the groundmass and phenocrysts. Panel c) shows sample UNZ-8-14 after experiencing brittle–dominated transitional behaviour at a strain rate of $10^{-3}$ s$^{-1}$; pervasive macroscopic fractures (> 100 μm in width) connect porosity and displace phenocrysts along their planes of weakness. Panels d) and e) are representative images of UNZ-8-21 which underwent viscous–dominated transitional behaviour when strained at $10^{-4}$ s$^{-1}$; small (< 200 μm in width) microfractures can be seen in the ground–mass glass, phenocrysts are pervasively fractured but show no sign of displacement. Panel f) is an image of sample UNZ-8-16 after experiencing viscous deformation at a strain rate of $10^{-5}$ s$^{-1}$; pores are aligned parallel to the direction of shear around phenocrysts with minor fractures < 100 μm in width. Panels g) and h) show UNZ-4 and UNZ-8, respectively, prior to deformation; with few, hairline fractures visible in the phenocrysts and little to no fractures in the smaller crystals or the groundmass glass.

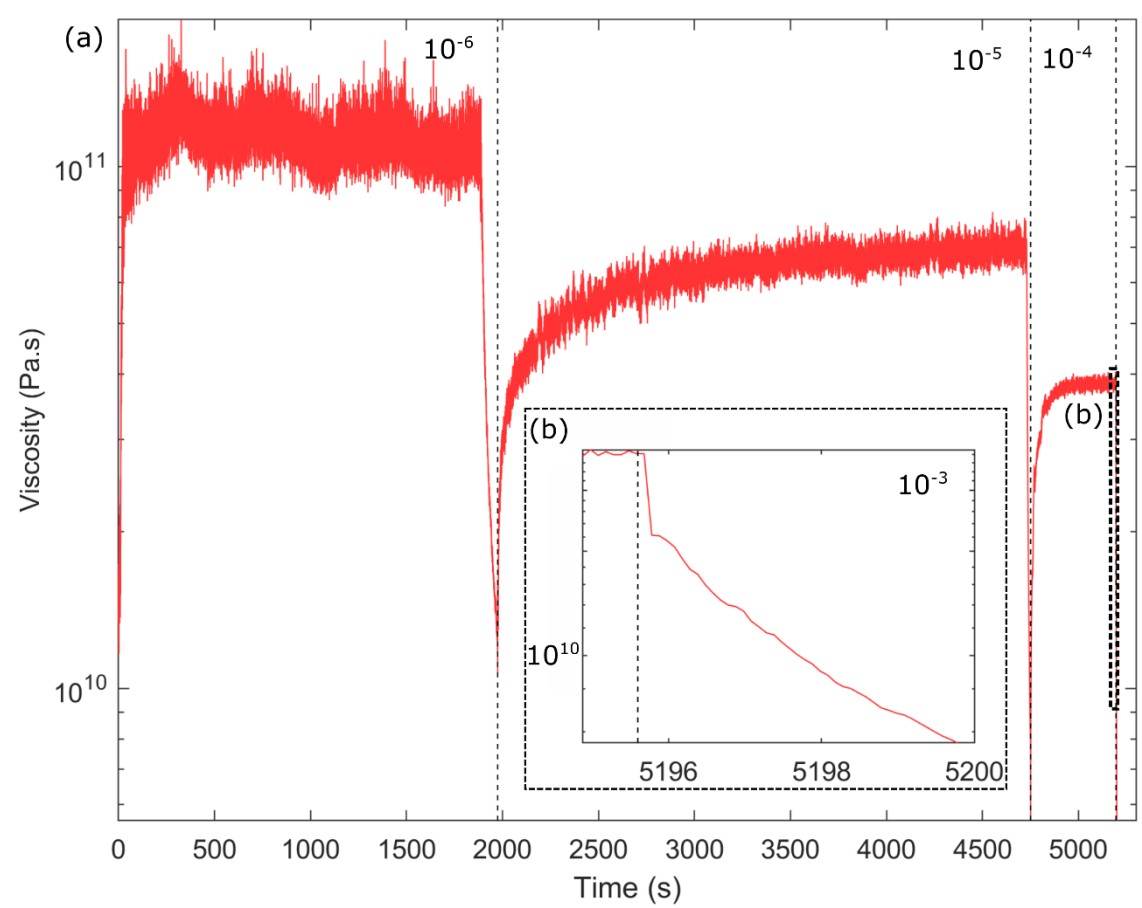

**Figure 7. Apparent viscosity evolution of UNZ-1 (porosity: 0.22) at 900 ˚C during a stepped strain–rate experiment ($10^{-6}$ s$^{-1}$, $10^{-5}$ s$^{-1}$, $10^{-4}$ s$^{-1}$, $10^{-3}$ s$^{-1}$); each step is separated by dashed lines. The insert zooms in on the apparent viscosity decrease that accompanies sample failure at $10^{-3}$ s$^{-1}$. The decrease in viscosity at each increasing strain rate increment highlights the shear thinning behaviour of these lavas.**

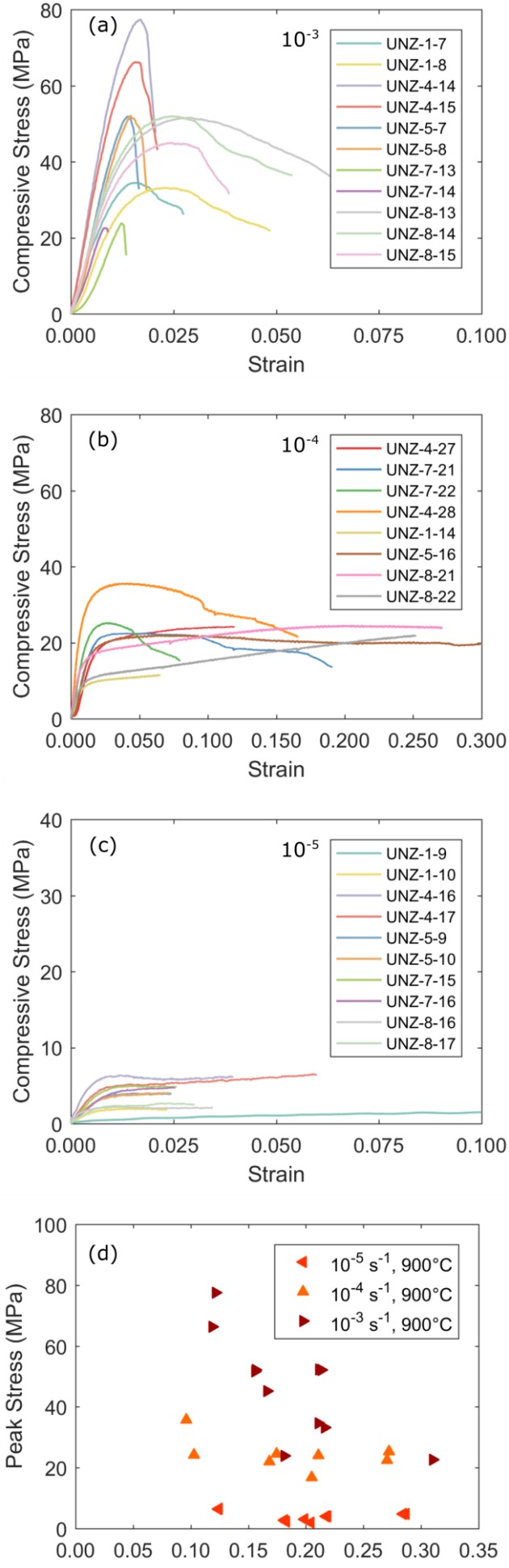

**Figure 8. High temperature uniaxial experiment results, including stress-strain curves for samples tested at a strain rates of (a) $10^{-3}$ $s^{-1}$, (b) $10^{-4}$ $s^{-1}$, and (c) $10^{-5}$ $s^{-1}$, demonstrating the shift from viscous flow at low rate to increasingly brittle deformation at faster rate. (d) The peak stresses achieved during each experiment carried out at 900 ˚C further highlights this observation and shows the porosity–dependence of strength in the brittle regime.**

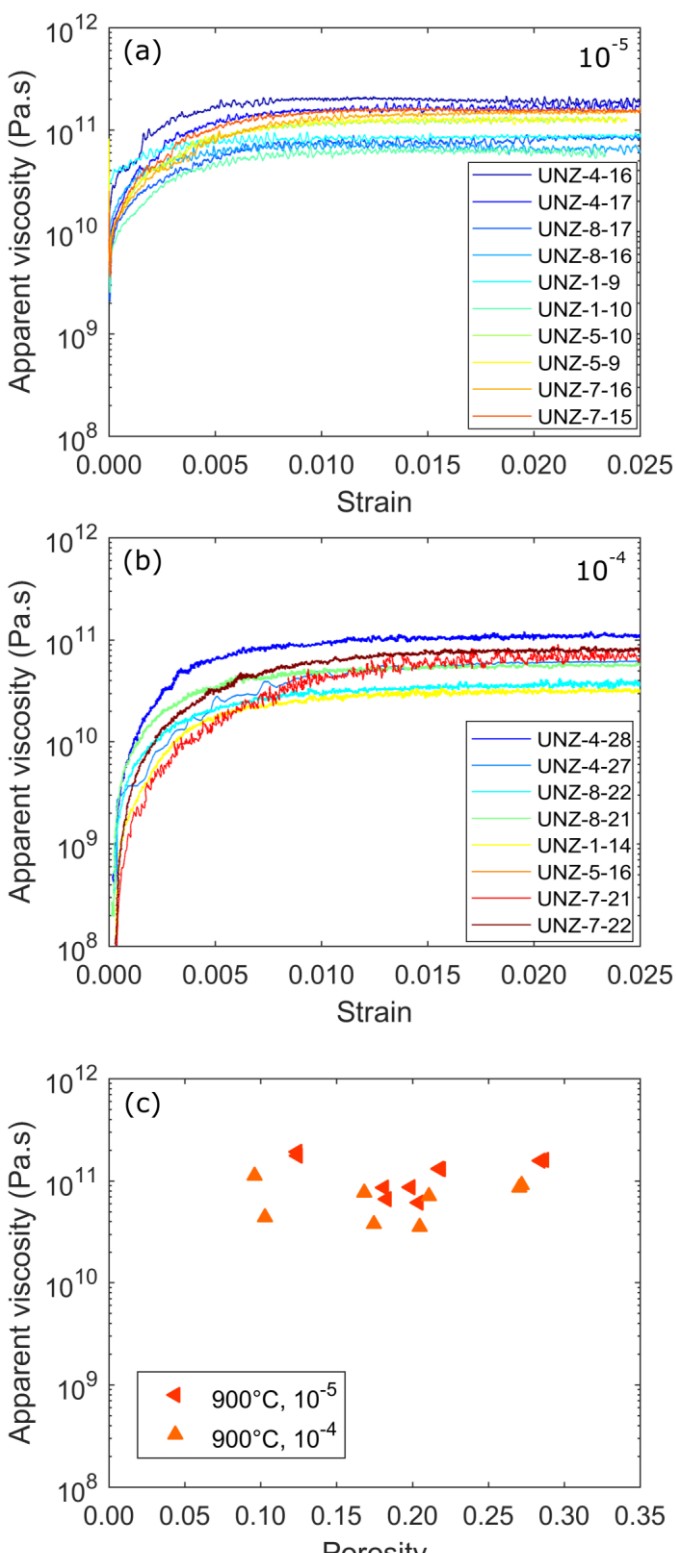

Figure 9. Apparent viscosities of porous lavas at 900 °C for strain rates of (a) $10^{-5}$ s$^{-1}$ and (b) $10^{-4}$ s$^{-1}$; colours warm from blue to red with increasing sample porosity. (c) Compilation of apparent viscosities as a function of porosity for samples tested at strain rates of $10^{-4}$ s$^{-1}$ and $10^{-5}$ s$^{-1}$. Viscosities decrease between strain rates of $10^{-5}$ to $10^{-4}$ s$^{-1}$, an example of shear thinning in the Unzen samples. Porosity has no control on the apparent viscosities of the samples tested here.

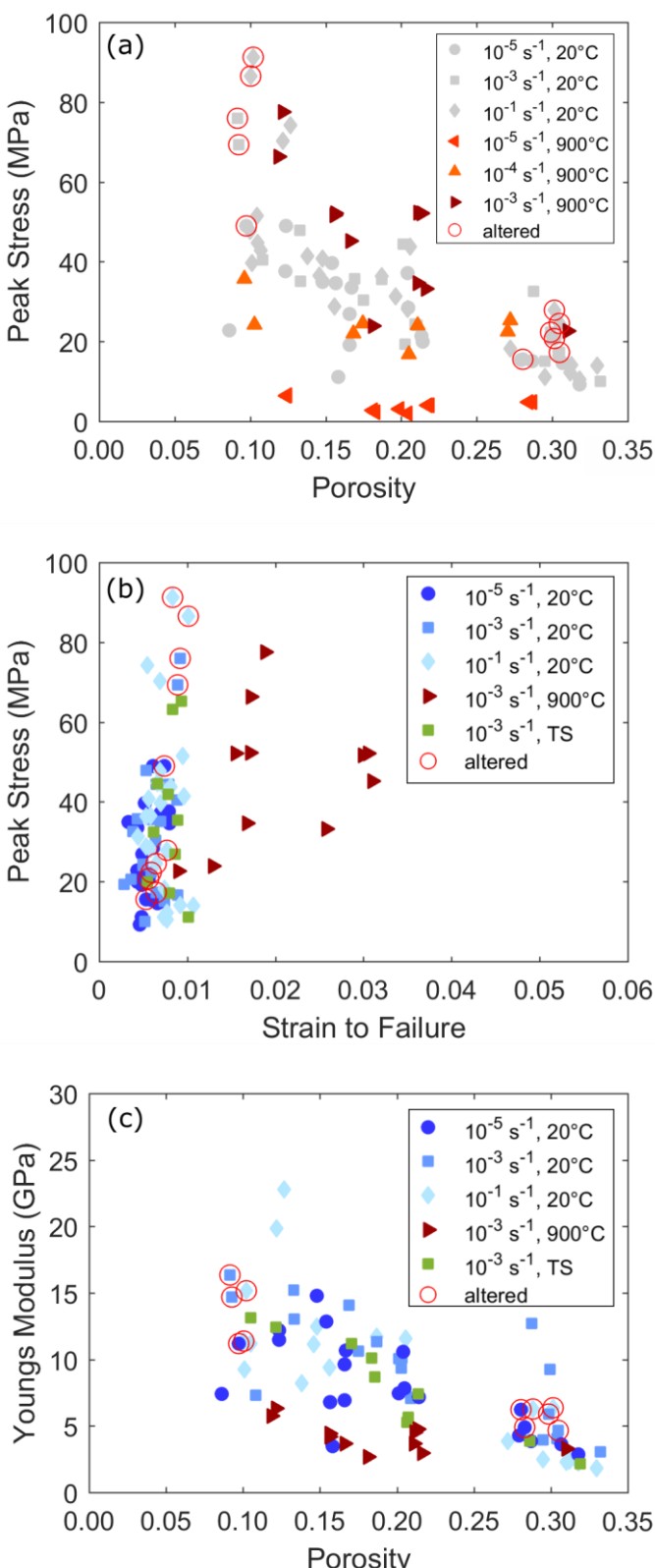


**Figure 10. Strength and Young's Moduli of Unzen rocks and lavas at different conditions. Shades of blue represent tests carried out at ambient temperatures, shades of red indicate those performed at 900 °C, and green depicts thermally stressed samples which were tested in ambient conditions. The red rings circle the samples which were deemed visibly altered at collection. (a) Peak stress with porosity for all completed experiments. Low temperature tests, as seen in Figure 5 (a), are faded to grey. (b) Peak stress with**

**strain at the point of sample failure (i.e. the stain at peak stress) for all experiments with a brittle response. (c) Young's modulus as a function of porosity for all samples that had a brittle response, calculated using the slope of the linear portion of the stress–strain curve (see Fig. S6).**

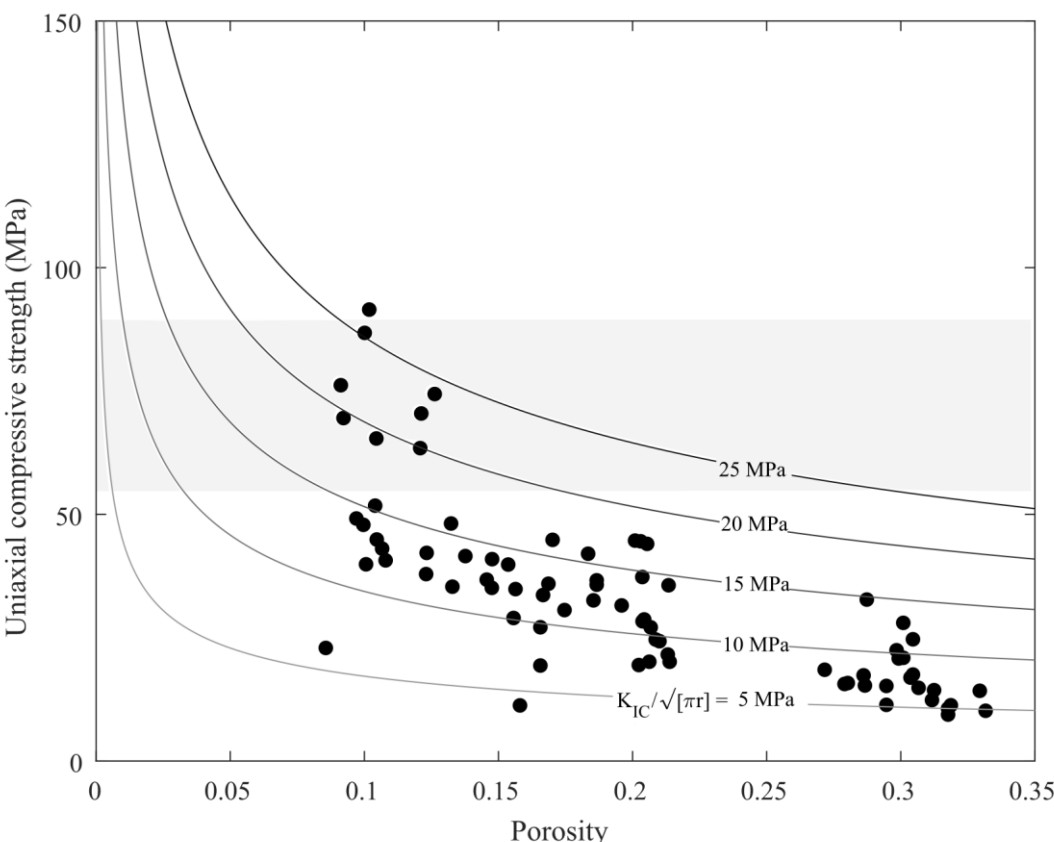

**Figure 11 Plot of uniaxial compressive stress against porosity showing the ambient temperature mechanical data (black dots), alongside contours of various values of $\frac{K_{IC}}{\sqrt{\pi r}}$ (5–25 MPa) from the pore–emanating crack model (Eq. 2). The range of UCS given by the wing-crack model is also plotted as a shaded region. The mechanical data are cross-cut by the contours, suggesting a change in the dominant porous structure. At porosities > 0.25 the UCS given by the pore-emanating crack model with $\frac{K_{IC}}{\sqrt{\pi r}}$ = 5–10 MPa seems to fit the data well. For porosities ranging from 0.12–0.2 the UCS given by the pore-emanating crack model with $\frac{K_{IC}}{\sqrt{\pi r}}$ = 10–15 MPa encloses the data. The UCS for the densest rocks in the study (~0.08–0.12) would suggest yet a higher $\frac{K_{IC}}{\sqrt{\pi r}}$ of 20–25 MPa. For porosities < 0.1 the UCS given by the wing–crack model is similar to the mechanical data (σ = 54.2–89.7 MPa).**

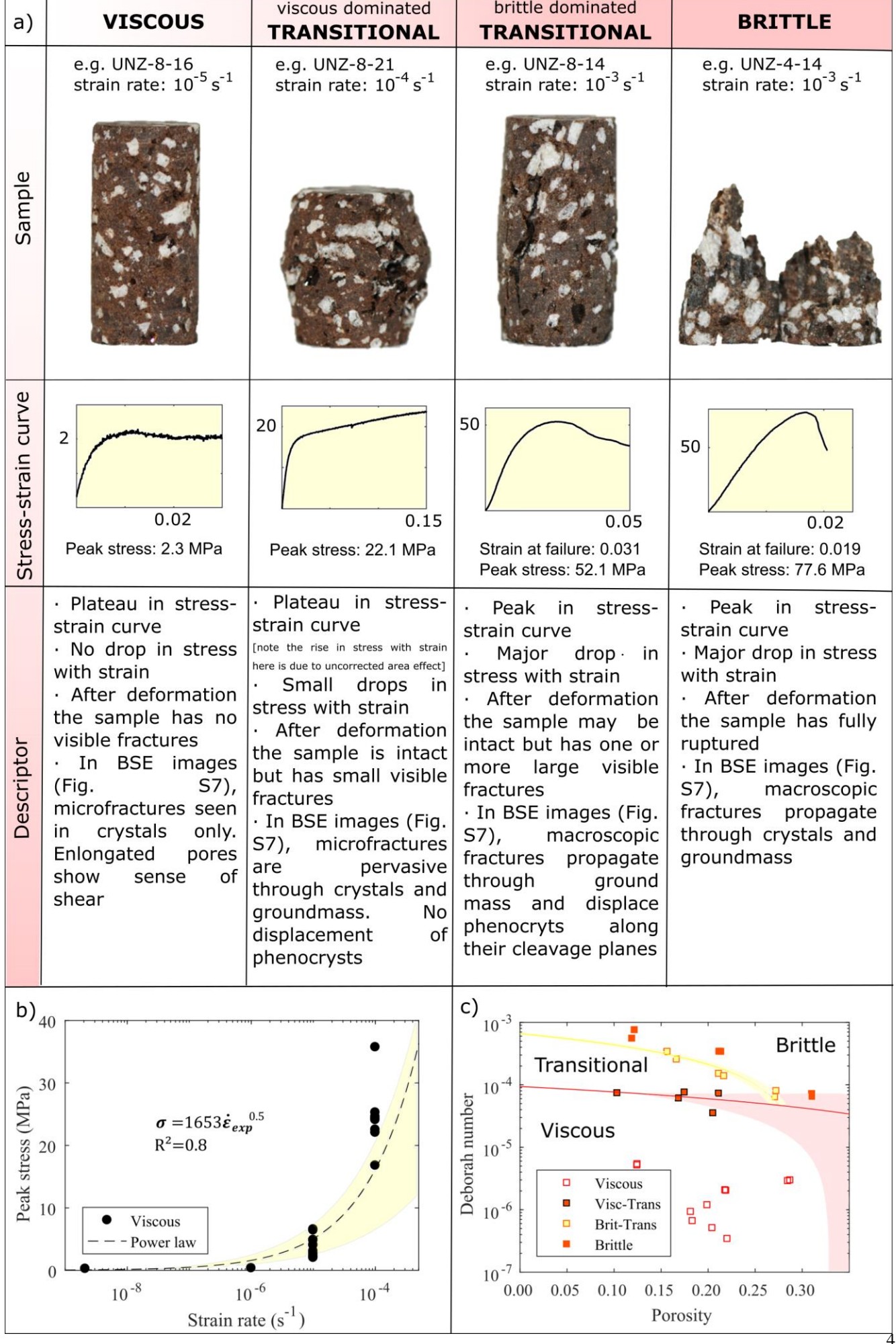

| | **VISCOUS** | viscous dominated **TRANSITIONAL** | brittle dominated **TRANSITIONAL** | **BRITTLE** |
|---|---|---|---|---|
| **Sample** | e.g. UNZ-8-16 strain rate: $10^{-5}$ $s^{-1}$ | e.g. UNZ-8-21 strain rate: $10^{-4}$ $s^{-1}$ | e.g. UNZ-8-14 strain rate: $10^{-3}$ $s^{-1}$ | e.g. UNZ-4-14 strain rate: $10^{-3}$ $s^{-1}$ |
| **Stress-strain curve** | 2 ... 0.02  Peak stress: 2.3 MPa | 20 ... 0.15  Peak stress: 22.1 MPa | 50 ... 0.05  Strain at failure: 0.031 Peak stress: 52.1 MPa | 50 ... 0.02  Strain at failure: 0.019 Peak stress: 77.6 MPa |
| **Descriptor** | · Plateau in stress-strain curve · No drop in stress with strain · After deformation the sample has no visible fractures · In BSE images (Fig. S7), microfractures seen in crystals only. Enlongated pores show sense of shear | · Plateau in stress-strain curve [note the rise in stress with strain here is due to uncorrected area effect] · Small drops in stress with strain · After deformation the sample is intact but has small visible fractures · In BSE images (Fig. S7), microfractures are pervasive through crystals and groundmass. No displacement of phenocrysts | · Peak in stress-strain curve · Major drop in stress with strain · After deformation the sample may be intact but has one or more large visible fractures · In BSE images (Fig. S7), macroscopic fractures propagate through ground mass and displace phenocryts along their cleavage planes | · Peak in stress-strain curve · Major drop in stress with strain · After deformation the sample has fully ruptured · In BSE images (Fig. S7), macroscopic fractures propagate through crystals and groundmass |

b)

$$\sigma = 1653 \dot{\varepsilon}_{exp}^{0.5}$$
$$R^2 = 0.8$$

Peak stress (MPa) vs Strain rate ($s^{-1}$)

● Viscous
- - Power law

c)

Deborah number vs Porosity

Transitional
Brittle
Viscous

□ Viscous
■ Visc-Trans
□ Brit-Trans
■ Brittle

**Figure 12. (a) A schematic demonstration of sample rheological classification [viscous, viscous–dominated transitional (visc-trans), brittle–dominated transitional (brit-trans) or brittle], depending on the respective shape of the stress-strain curve and the amount of strain experienced (b) Peak stress plotted with strain rate for completed experiments in semi-log space. The power law equation of the line is shown on the figure. Ostwald constants k and b are 1653 and 0.5 respectively. The standard error of estimate is shown on the plot as the yellow window, along with the $R^2$ value (c) The calculated Deborah number as a function of porosity for each**

**sample tested at high temperature, in semi-log space. The viscous–dominated transitional behaviour is marked by the red line bordered by a red window showing the standard estimate of error. The brittle–dominated transitional behaviour is marked by the yellow line bordered by a yellow window showing the standard estimate of error. The critical Deborah number, $De_c$, can thus be said to be between $1 \times 10^{-4}$ and $6.6 \times 10^{-4}$ for dense (pore-free) crystal-rich dome lavas, deceasing linearly with the addition of pores. We find that the two transitional zones converge at a porosity of approximately 0.27, beyond which, no transition zone exists**

**(although this coincide with the limit of the material properties studied). [NOTE: Sample UNZ-4-28 was omitted from this plot as its resulting stress-strain curve was likely due to an experimental artefact caused by chipping of the sample edge]**

| Sample | Total porosity | Connected porosity | Strain rate (s$^{-1}$) | Temperature (°C) | Peak force (N) | Peak Stress (MPa) | Strain to failure | Thermally treated | Altered | Viscosity (Pa.s) | De number | Young's modulus (GPa) |
|---|---|---|---|---|---|---|---|---|---|---|---|---|
| UNZ-1-2 | 0.21 | 0.19 | 1.E-05 | 20 | 6789 | 21.38 | 0.0049 | N | N | N/A | N/A | 7.45 |
| UNZ-4-13 | 0.09 | 0.07 | 1.E-05 | 20 | 7180 | 22.71 | 0.0043 | N | N | N/A | N/A | 7.40 |
| UNZ-5-1 | 0.20 | 0.18 | 1.E-05 | 20 | 11779 | 37.09 | 0.0070 | N | N | N/A | N/A | 10.58 |
| UNZ-5-5 | 0.20 | 0.19 | 1.E-05 | 20 | 9022 | 28.49 | 0.0061 | N | N | N/A | N/A | 7.84 |
| UNZ-7-1 | 0.29 | 0.28 | 1.E-05 | 20 | 4750 | 15.10 | 0.0071 | N | N | N/A | N/A | 3.85 |
| UNZ-7-10 | 0.31 | 0.30 | 1.E-05 | 20 | 4600 | 14.63 | 0.0066 | N | N | N/A | N/A | 3.63 |
| UNZ-7-12 | 0.32 | 0.31 | 1.E-05 | 20 | 2895 | 9.20 | 0.0046 | N | N | N/A | N/A | 2.86 |
| UNZ-7-6 | 0.28 | 0.28 | 1.E-05 | 20 | 4889 | 15.41 | 0.0060 | N | N | N/A | N/A | 4.27 |
| UNZ-8-1 | 0.17 | 0.17 | 1.E-05 | 20 | 6000 | 19.15 | 0.0048 | N | N | N/A | N/A | 6.93 |
| UNZ-8-10 | 0.15 | 0.15 | 1.E-05 | 20 | 12570 | 39.62 | 0.0052 | N | N | N/A | N/A | 12.84 |
| UNZ-8-12 | 0.17 | 0.14 | 1.E-05 | 20 | 10600 | 33.44 | 0.0043 | N | N | N/A | N/A | 10.67 |
| UNZ-8-6 | 0.17 | 0.17 | 1.E-05 | 20 | 8540 | 26.90 | 0.0049 | N | N | N/A | N/A | 9.65 |
| UNZ-4-25 | 0.16 | 0.14 | 1.E-05 | 20 | 3497 | 11.07 | 0.0048 | N | N | N/A | N/A | 3.48 |
| UNZ-4-26 | 0.16 | 0.11 | 1.E-05 | 20 | 10981 | 34.64 | 0.0080 | N | N | N/A | N/A | 6.79 |
| UNZ-1-0 | 0.21 | 0.19 | 1.E-05 | 20 | 6320 | 19.92 | 0.0043 | N | N | N/A | N/A | 7.16 |
| UNZ-2-4 | 0.12 | 0.10 | 1.E-05 | 20 | 13361 | 41.98 | 0.0061 | N | N | N/A | N/A | 12.19 |
| UNZ-2-5 | 0.12 | 0.10 | 1.E-05 | 20 | 11957 | 37.68 | 0.0079 | N | N | N/A | N/A | 11.50 |
| UNZ-12-4 | 0.10 | 0.09 | 1.E-05 | 20 | 15549 | 48.95 | 0.0074 | N | Y | N/A | N/A | 11.18 |
| UNZ-11-2 | 0.30 | 0.28 | 1.E-05 | 20 | 6592 | 20.71 | 0.0056 | N | Y | N/A | N/A | 6.21 |
| UNZ-11-3 | 0.28 | 0.27 | 1.E-05 | 20 | 4950 | 15.60 | 0.0053 | N | Y | N/A | N/A | 4.88 |
| UNZ-8-21 | 0.15 | 0.12 | 1.E-05 | 20 | 11073 | 34.90 | 0.0033 | N | N | N/A | N/A | 14.77 |
| UNZ-1-4 | 0.21 | 0.18 | 1.E-03 | 20 | 7681 | 24.42 | 0.0050 | N | N | N/A | N/A | 7.06 |
| UNZ-1-6 | 0.21 | 0.18 | 1.E-03 | 20 | 7639 | 24.08 | 0.0050 | N | N | N/A | N/A | 7.19 |
| UNZ-5-2 | 0.20 | 0.18 | 1.E-03 | 20 | 14081 | 44.47 | 0.0079 | N | N | N/A | N/A | 10.04 |
| UNZ-5-3 | 0.20 | 0.19 | 1.E-03 | 20 | 14065 | 44.33 | 0.0065 | N | N | N/A | N/A | 10.11 |
| UNZ-7-11 | 0.33 | 0.33 | 1.E-03 | 20 | 3150 | 9.98 | 0.0052 | N | N | N/A | N/A | 3.05 |

| | | | | | | | | | | | | |
|---|---|---|---|---|---|---|---|---|---|---|---|---|
| UNZ-7-2 | 0.30 | 0.29 | 1.E-03 | 20 | 5250 | 16.70 | 0.0089 | N | N | N/A | N/A | 4.05 |
| UNZ-7-7 | 0.30 | 0.29 | 1.E-03 | 20 | 4750 | 15.01 | 0.0074 | N | N | N/A | N/A | 3.97 |
| UNZ-8-11 | 0.19 | 0.17 | 1.E-03 | 20 | 11300 | 35.53 | 0.0059 | N | N | N/A | N/A | 11.32 |
| UNZ-8-3 | 0.18 | 0.15 | 1.E-03 | 20 | 9640 | 30.39 | 0.0064 | N | N | N/A | N/A | 10.63 |
| UNZ-8-7 | 0.17 | 0.14 | 1.E-03 | 20 | 11350 | 35.73 | 0.0043 | N | N | N/A | N/A | 14.08 |
| UNZ-4-24 | 0.11 | 0.09 | 1.E-03 | 20 | 12841 | 40.47 | 0.0088 | N | N | N/A | N/A | 7.30 |
| UNZ-2-1 | 0.13 | 0.11 | 1.E-03 | 20 | 15241 | 47.94 | 0.0053 | N | N | N/A | N/A | 15.22 |
| UNZ-2-6 | 0.13 | 0.11 | 1.E-03 | 20 | 11115 | 35.13 | 0.0070 | N | N | N/A | N/A | 13.04 |
| UNZ-13-1 | 0.29 | 0.27 | 1.E-03 | 20 | 10341 | 32.52 | 0.0038 | N | N | N/A | N/A | 12.70 |
| UNZ-13-2 | 0.30 | 0.29 | 1.E-03 | 20 | 6544 | 20.58 | 0.0036 | N | N | N/A | N/A | 9.25 |
| UNZ-12-1 | 0.09 | 0.09 | 1.E-03 | 20 | 22126 | 69.32 | 0.0089 | N | Y | N/A | N/A | 14.71 |
| UNZ-12-3 | 0.09 | 0.10 | 1.E-03 | 20 | 24227 | 75.97 | 0.0092 | N | Y | N/A | N/A | 16.33 |
| UNZ-11-4 | 0.30 | 0.30 | 1.E-03 | 20 | 7066 | 22.25 | 0.0059 | N | Y | N/A | N/A | 5.90 |
| UNZ-11-6 | 0.30 | 0.30 | 1.E-03 | 20 | 5507 | 17.32 | 0.0065 | N | Y | N/A | N/A | 4.65 |
| UNZ-1-14 | 0.20 | 0.18 | 1.E-03 | 20 | 7681 | 19.25 | 0.0028 | N | N | N/A | N/A | 9.34 |
| UNZ-1-11 | 0.19 | 0.16 | 1.E-03 | 20 | 10257 | 32.36 | 0.0062 | Y | N | N/A | N/A | 8.69 |
| UNZ-1-12 | 0.21 | 0.18 | 1.E-03 | 20 | 6334 | 19.92 | 0.0055 | Y | N | N/A | N/A | 5.26 |
| UNZ-4-18 | 0.10 | 0.09 | 1.E-03 | 20 | 20556 | 65.17 | 0.0093 | Y | N | N/A | N/A | 13.11 |
| UNZ-4-19 | 0.12 | 0.10 | 1.E-03 | 20 | 19939 | 63.22 | 0.0083 | Y | N | N/A | N/A | 12.40 |
| UNZ-5-11 | 0.21 | 0.21 | 1.E-03 | 20 | 11240 | 35.42 | 0.0089 | Y | N | N/A | N/A | 7.39 |
| UNZ-5-12 | 0.21 | 0.21 | 1.E-03 | 20 | 8515 | 26.89 | 0.0086 | Y | N | N/A | N/A | 5.64 |
| UNZ-7-17 | 0.29 | 0.29 | 1.E-03 | 20 | 5412 | 17.16 | 0.0080 | Y | N | N/A | N/A | 3.87 |
| UNZ-7-18 | 0.32 | 0.32 | 1.E-03 | 20 | 3515 | 11.10 | 0.0101 | Y | N | N/A | N/A | 2.14 |
| UNZ-8-18 | 0.18 | 0.16 | 1.E-03 | 20 | 13266 | 41.81 | 0.0078 | Y | N | N/A | N/A | 10.10 |
| UNZ-8-19 | 0.17 | 0.15 | 1.E-03 | 20 | 14175 | 44.63 | 0.0066 | Y | N | N/A | N/A | 11.19 |
| UNZ-1-1 | 0.20 | 0.17 | 1.E-01 | 20 | 9970 | 31.33 | 0.0044 | N | N | N/A | N/A | 9.83 |
| UNZ-1-3 | 0.20 | 0.18 | 1.E-01 | 20 | 8936 | 28.11 | 0.0058 | N | N | N/A | N/A | 8.66 |
| UNZ-4-20 | 0.10 | 0.09 | 1.E-01 | 20 | 16342 | 51.55 | 0.0095 | N | N | N/A | N/A | 8.51 |

| | | | | | | | | | | | | |
|---|---|---|---|---|---|---|---|---|---|---|---|---|
| UNZ-4-22 | 0.14 | 0.09 | 1.E-01 | 20 | 13050 | 41.33 | 0.0096 | N | N | N/A | N/A | 8.20 |
| UNZ-4-4 | 0.11 | 0.09 | 1.E-01 | 20 | 13580 | 42.84 | 0.0077 | N | N | N/A | N/A | 9.18 |
| UNZ-4-5 | 0.10 | 0.08 | 1.E-01 | 20 | 15160 | 47.67 | 0.0069 | N | N | N/A | N/A | 11.05 |
| UNZ-4-8 | 0.11 | 0.09 | 1.E-01 | 20 | 14200 | 44.69 | 0.0066 | N | N | N/A | N/A | 11.20 |
| UNZ-4-9 | 0.10 | 0.09 | 1.E-01 | 20 | 12580 | 39.67 | 0.0070 | N | N | N/A | N/A | 9.24 |
| UNZ-7-19 | 0.31 | 0.36 | 1.E-01 | 20 | 4492 | 14.16 | 0.0092 | N | N | N/A | N/A | 2.32 |
| UNZ-7-20 | 0.33 | 0.32 | 1.E-01 | 20 | 4442 | 14.03 | 0.0107 | N | N | N/A | N/A | 1.82 |
| UNZ-7-4 | 0.30 | 0.29 | 1.E-01 | 20 | 3546 | 11.18 | 0.0073 | N | N | N/A | N/A | 2.49 |
| UNZ-7-5 | 0.32 | 0.31 | 1.E-01 | 20 | 3300 | 10.43 | 0.0077 | N | N | N/A | N/A | 2.27 |
| UNZ-7-8 | 0.31 | 0.31 | 1.E-01 | 20 | 3858 | 12.15 | 0.0077 | N | N | N/A | N/A | 2.38 |
| UNZ-7-9 | 0.27 | 0.26 | 1.E-01 | 20 | 5802 | 18.29 | 0.0074 | N | N | N/A | N/A | 3.85 |
| UNZ-8-4 | 0.15 | 0.12 | 1.E-01 | 20 | 11600 | 36.56 | 0.0058 | N | N | N/A | N/A | 11.12 |
| UNZ-8-5 | 0.19 | 0.16 | 1.E-01 | 20 | 11540 | 36.42 | 0.0053 | N | N | N/A | N/A | 11.71 |
| UNZ-8-8 | 0.16 | 0.13 | 1.E-01 | 20 | 9125 | 28.79 | 0.0053 | N | N | N/A | N/A | 9.39 |
| UNZ-8-9 | 0.15 | 0.12 | 1.E-01 | 20 | 12910 | 40.71 | 0.0056 | N | N | N/A | N/A | 12.47 |
| UNZ-2-2 | 0.13 | 0.10 | 1.E-01 | 20 | 23562 | 74.18 | 0.0055 | N | N | N/A | N/A | 22.79 |
| UNZ-2-3 | 0.12 | 0.10 | 1.E-01 | 20 | 22309 | 70.24 | 0.0069 | N | N | N/A | N/A | 19.83 |
| UNZ-12-2 | 0.10 | 0.10 | 1.E-01 | 20 | 29086 | 91.30 | 0.0083 | N | Y | N/A | N/A | 15.15 |
| UNZ-12-5 | 0.10 | 0.10 | 1.E-01 | 20 | 27638 | 86.58 | 0.0101 | N | Y | N/A | N/A | 11.36 |
| UNZ-11-1 | 0.30 | 0.29 | 1.E-01 | 20 | 8840 | 27.80 | 0.0077 | N | Y | N/A | N/A | 6.35 |
| UNZ-11-5 | 0.30 | 0.29 | 1.E-01 | 20 | 7780 | 24.47 | 0.0065 | N | Y | N/A | N/A | 6.25 |
| UNZ-5-15 | 0.21 | 0.17 | 1.E-01 | 20 | 13805 | 43.81 | 0.0081 | N | N | N/A | N/A | 11.58 |
| UNZ-1-10 | 0.20 | 0.18 | 1.E-05 | 900 | 643 | 2.02 | N/A | N | N | 1.40E+11 | 5.17E-07 | N/A |
| UNZ-1-9 | 0.20 | 0.17 | 1.E-05 | 900 | 975 | 3.08 | N/A | N | N | 8.98E+10 | 1.20E-06 | N/A |
| UNZ-4-16 | 0.12 | 0.11 | 1.E-05 | 900 | 2041 | 6.41 | N/A | N | N | 2.87E+11 | 5.22E-06 | N/A |
| UNZ-4-17 | 0.12 | 0.11 | 1.E-05 | 900 | 2077 | 6.56 | N/A | N | N | 2.86E+11 | 5.46E-06 | N/A |
| UNZ-5-10 | 0.22 | 0.20 | 1.E-05 | 900 | 1294 | 4.06 | N/A | N | N | 1.77E+11 | 2.09E-06 | N/A |
| UNZ-5-9 | 0.22 | 0.20 | 1.E-05 | 900 | 1277 | 4.02 | N/A | N | N | 1.71E+11 | 2.05E-06 | N/A |

| UNZ-7-15 | 0.29 | 0.28 | 1.E-05 | 900 | 1540 | 4.87 | N/A | N | N | 2.06E+11 | 3.01E-06 | N/A |
|---|---|---|---|---|---|---|---|---|---|---|---|---|
| UNZ-7-16 | 0.28 | 0.27 | 1.E-05 | 900 | 1475 | 4.80 | N/A | N | N | 2.01E+11 | 2.93E-06 | N/A |
| UNZ-8-16 | 0.18 | 0.16 | 1.E-05 | 900 | 652 | 2.30 | N/A | N | N | 9.09E+10 | 6.70E-07 | N/A |
| UNZ-8-17 | 0.18 | 0.16 | 1.E-05 | 900 | 829 | 2.72 | N/A | N | N | 1.09E+11 | 9.40E-07 | N/A |
| UNZ-4-27 | 0.10 | 0.09 | 1.E-04 | 900 | 4696 | 24.28 | N/A | N | N | 4.55E+10 | 7.48E-05 | N/A |
| UNZ-7-21 | 0.27 | 0.26 | 1.E-04 | 900 | 7117 | 22.52 | 0.0851 | N | N | 8.38E+10 | 6.43E-05 | N/A |
| UNZ-4-28 | 0.10 | 0.08 | 1.E-04 | 900 | 11337 | 35.73 | 0.0529 | N | N | 1.20E+11 | 1.62E-04 | N/A |
| UNZ-7-22 | 0.27 | 0.26 | 1.E-04 | 900 | 8010 | 25.27 | 0.0285 | N | N | 8.62E+10 | 8.10E-05 | N/A |
| UNZ-8-21 | 0.17 | 0.15 | 1.E-04 | 900 | 7012 | 22.05 | N/A | N | N | 7.19E+10 | 6.17E-05 | N/A |
| UNZ-8-22 | 0.17 | 0.14 | 1.E-04 | 900 | 7843 | 24.60 | N/A | N | N | 3.83E+10 | 7.68E-05 | N/A |
| UNZ-5-16 | 0.21 | 0.19 | 1.E-04 | 900 | 7625 | 24.08 | N/A | N | N | 7.17E+10 | 7.36E-05 | N/A |
| UNZ-1-14 | 0.20 | 0.18 | 1.E-04 | 900 | 5278 | 16.78 | N/A | N | N | 3.60E+10 | 3.57E-05 | N/A |
| UNZ-1-7 | 0.21 | 0.19 | 1.E-03 | 900 | 11044 | 34.70 | 0.0168 | N | N | N/A | 1.53E-04 | 3.65 |
| UNZ-1-8 | 0.22 | 0.19 | 1.E-03 | 900 | 10637 | 33.23 | 0.0258 | N | N | N/A | 1.40E-04 | 2.93 |
| UNZ-4-14 | 0.12 | 0.10 | 1.E-03 | 900 | 24575 | 77.60 | 0.0189 | N | N | N/A | 7.64E-04 | 6.32 |
| UNZ-4-15 | 0.12 | 0.11 | 1.E-03 | 900 | 21048 | 66.40 | 0.0172 | N | N | N/A | 5.59E-04 | 5.75 |
| UNZ-5-7 | 0.21 | 0.19 | 1.E-03 | 900 | 16503 | 52.06 | 0.0155 | N | N | N/A | 3.44E-04 | 4.76 |
| UNZ-5-8 | 0.21 | 0.09 | 1.E-03 | 900 | 16566 | 52.21 | 0.0171 | N | N | N/A | 3.46E-04 | 4.53 |
| UNZ-7-13 | 0.31 | 0.31 | 1.E-03 | 900 | 7583 | 23.90 | 0.0129 | N | N | N/A | 7.24E-05 | 2.67 |
| UNZ-7-14 | 0.31 | 0.30 | 1.E-03 | 900 | 7187 | 22.70 | 0.0090 | N | N | N/A | 6.54E-05 | 3.30 |
| UNZ-8-13 | 0.16 | 0.13 | 1.E-03 | 900 | 16384 | 51.63 | 0.0299 | N | N | N/A | 3.38E-04 | 4.19 |
| UNZ-8-14 | 0.16 | 0.13 | 1.E-03 | 900 | 16571 | 52.07 | 0.0305 | N | N | N/A | 3.44E-04 | 4.43 |
| UNZ-8-15 | 0.17 | 0.14 | 1.E-03 | 900 | 14382 | 45.19 | 0.0310 | N | N | N/A | 2.59E-04 | 3.68 |
| UNZ-1-x | 0.22 | 0.21 | 1.E-06 | 900 | 127 | 0.34 | N/A | N | N | 1.20E+11 | 3.47E-07 | N/A |
| UNZ-1-x | 0.22 | 0.21 | 1.E-05 | 900 | 610 | 1.94 | N/A | N | N | 7.00E+10 | N/A | N/A |
| UNZ-1-x | 0.22 | 0.21 | 1.E-04 | 900 | 3682 | 11.72 | N/A | N | N | 3.85E+10 | N/A | N/A |
| UNZ-1-x | 0.22 | 0.21 | 1.E-03 | 900 | 8383 | 26.68 | 0.9180 | N | N | N/A | N/A | N/A |

**Table 1. Sample properties, measurement data, experimental conditions, mechanical response and resulting properties of each sample.**

| | XRF | | | Microprobe | |
|---|---|---|---|---|---|
| | UNZ-4 | UNZ-11 | UNZ-12 | UNZ-4 glass | standard deviation |
| **SiO$_2$** | 64.07 | 65.2 | 65.48 | 79.20 | 0.20 |
| **TiO$_2$** | 0.67 | 0.66 | 0.61 | 0.40 | 0.01 |
| **Al$_2$O$_3$** | 16.34 | 15.98 | 16.39 | 11.13 | 0.02 |
| **Fe$_2$O$_3$** | 4.84 | 4.67 | 4.35 | - | - |
| **FeO** | - | - | - | 0.92 | 0.01 |
| **MnO** | 0.10 | 0.10 | 0.09 | 0.01 | 0.01 |
| **MgO** | 2.57 | 2.37 | 2.02 | 0.07 | 0.02 |
| **CaO** | 5.18 | 4.55 | 4.63 | 0.56 | 0.02 |
| **Na$_2$O** | 3.61 | 3.56 | 3.69 | 2.83 | 0.09 |
| **K$_2$O** | 2.31 | 2.55 | 2.46 | 4.87 | 0.07 |
| **P$_2$O$_5$** | 0.17 | 0.12 | 0.15 | - | - |
| **LOI** | 0.14 | 0.23 | 0.14 | - | - |
| **Total** | 100 | 100 | 100 | 100 | 0 |

**Table 2. Normalised chemical composition of bulk rocks obtained by XRF analysis and interstitial glass obtained by EPMA. UNZ-4 was selected as it is representative of fresh lavas tested in this study; in contrast, UNZ-11 and UNZ-12 were deemed to display a certain degree of alteration. Original totals were 99.97, 100.39, 100.09, and 99.95 for UNZ-4, UNZ-12, UNZ-11, UNZ-4 glass, respectively, before normalisation for direct comparison. The standard deviation of the UNZ-4 glass was taken from 2 measurements.**



| Sample block name | Average total porosity | Standard deviation | Average connected porosity | Standard deviation | Average isolated porosity | Standard deviation | No. samples |
|---|---|---|---|---|---|---|---|
| UNZ-1 | 0.21 | 0.011 | 0.18 | 0.012 | 0.02 | 0.002 | 17 |
| UNZ-2 | 0.13 | 0.016 | 0.11 | 0.018 | 0.02 | 0.003 | 7 |
| UNZ-4 | 0.12 | 0.016 | 0.12 | 0.021 | 0.00 | 0.016 | 30 |
| UNZ-5 | 0.21 | 0.006 | 0.19 | 0.009 | 0.02 | 0.006 | 18 |
| UNZ-7 | 0.30 | 0.024 | 0.29 | 0.025 | 0.01 | 0.002 | 23 |
| UNZ-8 | 0.16 | 0.016 | 0.14 | 0.016 | 0.02 | 0.003 | 24 |
| UNZ-11 | 0.30 | 0.009 | 0.29 | 0.011 | 0.01 | 0.004 | 8 |
| UNZ-12 | 0.10 | 0.025 | 0.09 | 0.026 | 0.00 | 0.002 | 7 |
| UNZ-13 | 0.32 | 0.005 | 0.30 | 0.004 | 0.01 | 0.004 | 6 |

Table 3. Average total, connected, and isolated porosities for each sample block used. A larger number of cores were measured to calculate the average porosities than those used in strength tests. *Note: values are presented to 2 d.p. but were calculated with 4 d.p.*