# Peer review of "Failure criteria for porous dome rocks and lavas: a study of Mt. Unzen, Japan"

_Solid Earth, 2018_

## Referee Comment (RC1) · J. Farquharson (Referee) · 3 Apr 2018

Dear Editor and authors,

I have read the manuscript *Failure criteria for porous dome rocks and lavas: a study of Mt. Unzen, Japan* with great interest. By way of a targeted experimental campaign, Coats *et al*. map out the failure conditions for suites of variably-porous crystalline andesite as a function of different temperatures and strain rates. The conclusion of the study is an empirical threshold for the failure of these materials derived from their data.

The article is assidious and well written, the experimental protocol appears rigorous, and the study yields a wealth of interesting new data. Overall, this article represents a commendable research effort from the authors. Where the authors perhaps do themselves a disservice is in the analysis of their data, which could be more comprehensive. Below I outline minor comments or concerns that I feel the authors should address or clarify. Pending these changes, I recommend this article for publication in *Solid Earth*.

Yours faithfully,

Jamie Farquharson

**Comments**

(Please forgive the clumsy formatting, I was forced to write this report on my phone)

Lines 39-56: Sparks (1997) is highly relevant to this study, and is a surprising omission here.

> Sparks (1997) The causes and consequences of pressurisation in lavas done eruptions. *Earth and Planetary Science Letters* 150(3-4): 177-189

Line 66: how do the authors define the "temperature range of interest"?

Line 101: it might be useful to provide the equation for *Ca* here.

Lines 134-144, and elsewhere: the authors describe two inclusion models which they highlight may explain their data: the pore-emanating crack model of Sammis and Ashby, and the sliding wing crack model of Ashby and Sammis. However, the authors do not go on to employ either of these models subsequently. As I intimated previously, it seems something of a shame that there is not a more involved analysis of these data. Analytical solutions for both these models are provided by Zhu *et al*. 2010 *JGR* and Baud *et al*. 2014 *IJRMS*, respectively. Previous authors have utilised one or other in order to describe the failure behavior of volcanic materials or analogues, for example Zhu *et al*. 2011 *JGR* (for tuffs), Vasseur *et al*. 2013 *GRL* (sintered glass), Heap *et al*. 2014 *JGR* (andesite), Zhu *et al*. 2016 *JGR* (basalts). Moreover, Zhu *et al*. 2011 extend the analytical solution to a dual-porosity medium, and Zhu *et al*. 2016 combine both models so as to have a representative element volume comprised of an effective medium including a pore surrounded by many cracks. If Coats *et al*. were to interrogate their data in a similar manner, they may be able to glean valuable information about the governing microstructural elements in their samples contributing to failure (for example by contouring for different values of $K_{Ic}/\sqrt{\pi r}$).

Line 146: the authors should state clearly that they here define "lava" as something at high temperature (volcanic rocks may often also be referred to as lava).

Line 184: in which direction has the lobe moved (*i.e.* rotation, inflation, advance)?

Line 279, and elsewhere: is this a change of 1 % or 1 vol.%?

Line 296 and 298: which software was used to run the scripts? Are they publicly available?

Line 322: the observation that in dense materials the connectivity is higher than in porous materials is somewhat counter-intuitive, and runs counter to results from previous researchers (Farquharson *et al*. 2015 *JVGR* and Collombier *et al*. 2017 *EPSL*, both of which noted that connectivity generally increased with increasing porosity). Do the authors have any comment on this difference?

Line 342: cristobalite isn't a polymorph of quartz. Rather, both quartz and cristobalite are silica polymorphs.

Line 355-359: did sample volume change upon heating? This really is surprising, as one would anticipate thermal cracking upon heating and subsequent cooling, due to thermal expansion mismatch between the constituent microstructural components (*e.g.* Browning *et al*. 2016 *GRL*). This may be especially pronounced given the existence of cristobalite in your samples, which undergoes a significant volumetric change as it transitions between its alpha and beta forms (a function of temperature: see for example, Damby *et al*. 2014 *JAC*).

Line 378-380: it is not immediately clear from the figure, but does the rock stiffness increase with thermal stressing? Here you identity microfractures as the culprit, yet previously you indicate that thermal stressing doesn't affect the porosity of the samples. Moreover, I would think that induced cracking would serve to decrease the material stiffness, rather than increase it. I suggest the authors re-word this section for clarity and consistency.

Line 400-401: as previously, it would by useful if the authors were to distinguish between % and vol. %.

Line 431: Young's modulus ought to be capitalised. Also, there is a full stop missing at the end of this sentence.

Line 441-445: see comment above concerning these inclusion models.

Line 470: could this observation be due to pore anisotropy (*e.g.* Bubeck *et al*. 2017; Griffiths *et al*. 2017)?

Line 480-481: do the authors have any information on the pore size distribution or pore anisotropy that could help explain this?

Line 525: shear-induced development of pore connectivity was also shown experimentally by Kushnir *et al*. 2017 *EPSL*.

Line 535-560: as with my previous comments concerning the inclusion models of Ashby and Sammis and Sammis and Ashby, I believe the authors could extend this Deborah number analysis some more.

- As far as I can tell, the dimensions of Equation 10 only balance out if $b = 1$. Pa = [Pa.s × s^-1] is fine ($b = 1$), but Pa = $\sqrt{[Pa.s × s^{-1}]}$ is not, as would be the case were $b = 0.5$. To me, this highlights a serious shortcoming in the empirical approach adopted here, especially as there has been more recent work on the topic which adopt physical rather than empirical parameters, for example the Wadsworth *et al*. 2017 chapter referenced elsewhere in the manuscript. For example, that chapter addresses the physical scaling of *De* with crystal content. Ultimately, a similar physical approach could yield a much more generally applicable failure criterion for porous materials.
- Line 535: using *exp* as a subscript is a little ambiguous (at first glance I presumed it signified an exponential).
- Line 546: Oswald should presumably be Ostwald.
- Line 546: here, the authors state values of *k* and *b* of 1653 and 0.5, yet in the caption for Figure 10, the values are $k = 1606$ and $b = 0.7755$. Which of these are correct? What is the sensitivity of the the following analysis to variations on *k* and/ or *b*?
- Equation 11: Based on equation 9, shouldn't $\sigma$ relate to strain rate × viscosity (or equivalently, $De \times G_\infty$)?

Line 618: this is in contrast to existing theory, models, and experimental data. Perhaps this effect is masked in your data by sample heterogeneity? I would be wary of including this point as a key conclusion of the study.

Figure 4: there appears to be some data obscured by the legend in panels *a* and *b*.

Figure 6: for clarity, perhaps the authors could plot viscosity as a function of strain rate (similar to Figure 10a; perhaps with symbols coloured for time).

Figure 10: in panel *a*, the authors state that the equation is shown on the figure, but it is missing. In panel *b* (and line 555), is the relation given by the yellow line based on only three data (i.e. the transitional data)? What is the r2 value of this relation? Can the authors comment on the theoretical value of *De_c* for a nonporous material? Would this fit on the trend? Likewise, how do the authors anticipate *De_c* evolving for highly porous materials?

---

## Referee Comment (RC2) · S. Quane (Referee) · 25 May 2018

This paper is potentially an excellent contribution to the experimental literature on natural, complex, multi-component, volcanic rocks. Initially, the authors provide an excellent treatment of the relevant rheological behavior of high-temperature deformation in glassy samples which is followed by a good characterization of representative, pre-experiment cores. The study comprises a substantial number of experiments both at ambient room (20oC) and magmatic temperatures (900oC). The rheological data is of high quality and the experimental conditions and span a reasonable range of deformation rates and timescales. The authors do an excellent job of presenting the results separately from analysis and a thorough job of explaining the major decisions and assumptions they had to make in the process of running the experiments (i.e., how and

why samples were chosen based on porosity, connected porosity, etc., detail on how experimental charges were loaded and the effect that has on the experiment). The rheological analysis is solid and prima facie, the interpretations seem sound and lead to a number of logical conclusions about the behavior of these multicomponent systems under conditions relevant to Unzen eruptions. However, I see one main oversight in their otherwise detailed and robust analysis; there are no descriptions, photos, or representative images and quantitative measurements (porosity, density) of experimental run products. This is a major issue for several reasons: a) Deformation in experimental charges can not be interpreted by using the rheological data alone. For example, seemingly "viscous" behavior and "brittle" behavior were interpreted (starting is section 4.1) based on the "mechanical responses" of the rocks. The authors make assumptions and "attributions" about the actual mechanical behavior of the samples with no empirical evidence. For example they "attribute to a narrowing of pre-existing cracks" and "hypothesize may reflect a contribution of viscous deformation upon loading". It is possible that these interpretations are correct, however, it would be relatively easy to test the attributions and hypotheses by halting an experiment at the requisite place on the deformation path and doing microstructural analysis. b) Post experiment analysis of end products can lead to surprising conclusions about mechanical behavior. In these multicomponent systems, deformation can occur via several mechanisms. Bubble collapse, brittle fracturing, viscous flow of groundmass glass, microcracking, rotation of grains, grain boundary sliding, internal grain deformation. All of these are factors in accommodating strain in the samples. Hence, strain can be accommodated homogenously (throughout the sample evenly) or it can be localized into bands or disparate parts of the sample. Without post experiment analysis, these important rheological behaviors cannot be determined. The authors are making the most logical conclusions about their "brittle" and "viscous" determinations based on the rheological data, however, without visual analysis of bulk properties and microstructures, the authors cannot confirm behavior. In addition, they are losing a considerable amount of important information about the nature of the deformation. c) Post experiment analysis of physical

properties (i.e., density, porosity) can yield important information on the nature of deformation. Certainly, for the cores that were not destroyed during brittle failure, the authors can make density and porosity determinations via the methods they used on the pre-experiment cores. Bulging of cores may cause a little consternation, however, established methods exist in the volcanology literature to measure density and porosity on irregular samples. d) Characterizing the amount of strain in the samples is an independent measure of machine strain. Does the sample show the same amount of strain as the machine? This can be determined through post-experiment analysis of density, porosity and core geometry. It is an important check on the experimental apparatus to ensure all strain from the machine is going into the sample. Quane and Russell, 2005 (cited by authors) and Quane et al., 2004 from American Mineralogist go through these procedures in detail.

Without post-experiment characterization (on samples that will allow it-sometimes even brittle deformation samples can be salvaged and epoxyed), the authors cannot speak with authority on the types of deformation occurring. Unfortunately, by not having that authority, the Conclusions they draw come into question. Certainly, the authors can do an analysis of the run products and produce a figure or two (like Figure 3 does for pre-experiment cores) to describe the major mechanisms of deformation and strain accommodation. Without this, this otherwise very strong, methodical and detailed contribution falls incomplete.

Technical corrections in this manuscript a very minimum. Found one spelling mistake, but I lost it!

---

## Editor Comment (EC1) · J. Gottsmann (Editor) · 5 Jun 2018

Dear authors,

As you have seen we have received two independent reviews of your manuscript and both recommend publication after revision. While focusing on different weaknesses in the current manuscript, both reviewers provide concrete suggestions for improvement. One reviewer asks for additional analyses to substantiate some claims made in the ms about the mechanical behaviour of the samples during the experiments. Performing additonal analyses may or may not be feasible during revision. It will be down to the authors to decide in which way these criticisms can be adressed. A careful edit of parts of the narrative that according to one reviewer lack empirical underpinning will be

necessary.

I expect to receive a carefully revised manuscript including a rebuttal letter that provides details on how the reviewers' comments have been addressed. I will then consider your manuscript for a final decision.

With best wishes, J. Gottsmann Executive Editor Solid Earth

————————————————

---

## Author Response (AR1)

**Point-by-point response to the reviews**

**Response to Farquharson**

Dear Editor and authors,

- 5 I have read the manuscript *Failure criteria for porous dome rocks and lavas: a study of Mt. Unzen, Japan* with great interest. By way of a targeted experimental campaign, Coats *et al.* map out the failure conditions for suites of variably-porous crystalline andesite as a function of different temperatures and strain rates. The conclusion of the study is an empirical threshold for the failure of these materials derived from their data.
- 10 The article is assiduous and well written, the experimental protocol appears rigorous, and the study yields a wealth of interesting new data. Overall, this article represents a commendable

research effort from the authors. Where the authors perhaps do themselves a disservice is in the analysis of their data, which could be more comprehensive. Below I outline minor

comments or concerns that I feel the authors should address or clarify. Pending these changes, I recommend this article for 15 publication in *Solid Earth*.

Yours faithfully, Jamie Farquharson

20 (Please forgive the clumsy formatting, I was forced to write this report on my phone)

Lines 39-56: Sparks (1997) is highly relevant to this study, and is a surprising omission here. Sparks (1997) The causes and consequences of pressurisation in lavas done eruptions. *Earth and Planetary Science Letters* 150(3-4): 177-189

25 - We thank the reviewer for bringing this to our attention and have added the reference in question, an oversight given its relevance to the fundamentals to this study.

**Line 66: how do the authors define the "temperature range of interest"?**

- Webb and Dingwell (1989) compiled a large dataset of elastic modulus at infinite frequency,  $G_{\infty}$ , for silicate melts and many other compounds, over a range of temperatures. Previous studies found that  $G_{\infty}$  ranges from 5 to 42 GPa at room temperature for glasses with silica contents ranging between 5-99 mole% (Bansal and Doremus, 1986), and at temperatures between 400 and 1600 °C,  $G_{\infty}$ = 33 GPa ± 5% (Bucaro and Dardy, 1974). Webb and Dingwell (1989) added data to find that for silicate melts and glasses,  $G_{\infty}$  ranges between 3.2 and 32 GPa and thus can be approximated at  $10^{10\pm0.5}$ as it only weakly varies with temperature (unlike viscosity). This brings an important simplification to the modelling of viscoelastic melts, which has been pivotal in its integration to volcanology. Therefore, we can assume  $G_{\infty} = 10^{10\pm0.5}$  for all

35 viscoelastic melts, which has been pivotal in its integration to volcanology. Therefore, we can assume  $G_{\infty} = 10^{-5.50}$  for all silicate melts (and glasses) at ~20-1600 °C, which extends beyond the temperature range of most (contemporaneous) volcanic systems reported in the literature. Hence, the "temperature range of interest" is all volcanic temperatures.

**Line 101: it might be useful to provide the equation for *Ca* here.**

- The paragraph details our knowledge of complex lavas (such as dome lavas) containing crystals and bubbles. The capillary number is highly relevant in aphyric lavas, but in the lavas discussed here, it remains to be adapted, which is beyond the scope of this study. Hence, we wish not to introduce the equation here and believe a qualitative description is sufficient.

45

50

Lines 134-144, and elsewhere: the authors describe two inclusion models which they highlight may explain their data: the pore-emanating crack model of Sammis and Ashby, and the sliding

wing crack model of Ashby and Sammis. However, the authors do not go on to employ either of these models subsequently. As I intimated previously, it seems something of a shame that

there is not a more involved analysis of these data. Analytical solutions for both these models are provided by Zhu *et al.* 2010 *JGR* and Baud *et al.* 2014 *IJRMS*, respectively. Previous

authors have utilised one or other in order to describe the failure behavior of volcanic materials or analogues, for example Zhu *et al.* 2011 *JGR* (for tuffs), Vasseur *et al.* 2013 *GRL* (sintered

55 glass), Heap *et al.* 2014 *JGR* (andesite), Zhu *et al.* 2016 *JGR* (basalts). Moreover, Zhu *et al.* 2011 extend the analytical solution to a dual-porosity medium, and Zhu *et al.* 2016 combine both

models so as to have a representative element volume comprised of an effective medium including a pore surrounded by many cracks. If Coats *et al.* were to interrogate their data in a

similar manner, they may be able to glean valuable information about the governing microstructural elements in their samples

60 contributing to failure (for example by contouring for different values of  $KIc/\sqrt{\pi r}$ ).

-We thank Dr Farquharson for his detailed comments. Indeed, we had previously investigated the comparison of our data with the pore and wing-crack models but had excluded it from the results based on the assumptions of parameters we had to make in fitting the wing crack model. However, as the reviewer asked for this data we felt it important to extend our analysis. The following has been added to the memory in action 1.2.

following has been added to the manuscript in section 1.3:

[revised manuscript text omitted]

Line 146: the authors should state clearly that they here define "lava" as something at high temperature (volcanic rocks may often also be referred to as lava).

110 -We thank the reviewer for this comment and agree the definition between cold lava and high temperature lava was unclear at certain points in the manuscript. We have edited the manuscript with the following note: '[note: From here, samples deformed at high temperature will be defined as lavas, and those tested at room temperature as rocks].'

**115 Line 184: in which direction has the lobe moved (*i.e.* rotation, inflation, advance)?**

-We have now added that: The mechanism of movement is thought to be down-slope advancement to the East (Matsushima and Takagi, 2000).

**Line 279, and elsewhere: is this a change of 1 % or 1 vol.%?**

120 -It is a change of 1 vol.%. We have changed the manuscript to read 0.01 in place of 1%, in line with the method of reporting porosity used throughout the manuscript (e.g. Table 1). We have also noted this change elsewhere in the manuscript so that porosities are reported as a fraction. Where crystals are concerned we have reported these as vol.%

**Line 296 and 298: which software was used to run the scripts? Are they publicly available?**

125 -The software used to run the scripts was MATLAB. A version of the script is now freely available on Github (https://doi.org/10.5281/zenodo.1287237). This has now been made clear in the manuscript.

Line 322: the observation that in dense materials the connectivity is higher than in porous materials is somewhat counterintuitive, and runs counter to results from previous researchers

130 (Farquharson et al. 2015 JVGR and Collombier et al. 2017 EPSL, both of which noted that connectivity generally increased with increasing porosity). Do the authors have any comment on this difference?

-We agree it seems counter intuitive, but the difference is very small: a matter of 0.01-0.02 unconnected pores. We attribute this to the presence of more small, isolated pores in the high porosity samples than in the low porosity samples. We also note

that the porosity range of our rocks is considerably smaller (0.09-0.32) than Farquharson et al., (2015) (0.025-0.73) and 135 Collombier et al., (2017) (~0.0-1.0), and so note that we are not evaluating the same range as these other studies. Our 'high porosity' samples could be considered low porosity in comparison to these other studies and so a comparison cannot be clearly drawn.

**140 Line 342: cristobalite isn't a polymorph of quartz. Rather, both quartz and cristobalite are silica polymorphs.**

-We thank the reviewer for bringing this to our attention and agree the text would be clearer if the polymorph was referred to as a silica polymorph. The manuscript and Figure 3 have been edited in accordance with this.

Line 355-359: did sample volume change upon heating? This really is surprising, as one would anticipate thermal cracking upon heating and subsequent cooling, due to thermal expansion

mismatch between the constituent microstructural components (*e.g.* Browning *et al.* 2016 *GRL*). This may be especially pronounced given the existence of cristobalite in your samples,

which undergoes a significant volumetric change as it transitions between its alpha and beta forms (a function of temperature: see for example, Damby *et al.* 2014 *JAC*).

- 150 We refer Dr Farquharson to the supplementary data, S5. Thermo-mechanical analysis (TMA) of the Mt. Unzen dome rock shows thermal expansion upon heating until the softening point of the material. We also refer him to lines 503-506 in the original manuscript ("A recent study by Eggertsson et al., *in review*, found that the strength and porosity of samples that hosted microfractures (like Mt. Unzen dome rock) were not affected by thermal stressing, while those that showed a trivial fraction of pre-existing micro-fractures were more significantly influenced through thermal stressing and as a result became more
- 155 permeable."). In accordance with Eggertsson et al., (2018), as our samples contained a pre-existing network of microfractures they were not readily fractured by thermal expansion and contraction.

We also would like to point out that samples containing cristobalite, UNZ-13 (see Figure 3), were not tested at high temperature nor thermally stressed to avoid adding effects of mineralogical reactions in this study that would not be relevant in the lava dome setting – i.e. the cristobalite post-dates high-temperature emplacement. We also would like to state that the heating and cooling rates used (4 °C min-1) were low enough to limit the differential expansion of the samples (that is, caused by temperature gradients across the samples, which also contribute to thermal cracking and which are not always considered independently from differential expansion of the constituent phases).

Line 378-380: it is not immediately clear from the figure, but does the rock stiffness increase with thermal stressing? Here you identity microfractures as the culprit, yet previously you

indicate that thermal stressing doesn't affect the porosity of the samples. Moreover, I would think that induced cracking would serve to decrease the material stiffness, rather than increase it.

I suggest the authors re-word this section for clarity and consistency.

145

165

- No, the rock stiffness (i.e. Young's Modulus) does not increase with thermal stressing, it in fact decreases (Figure 9c and
   lines 479-498, original manuscript). We attribute the initial convex portion of the stress-strain curves to the closure of
   microcracks perpendicular and sub-parallel to the principal stress (as in Heap et al., 2014, JGR). This initial section of the
   stress-strain curve is more pronounced in the case of the thermally stressed samples (Figure 4c), indicating that either 1) the
   width of pre-existing macro-fractures increased or that 2) more microfractures (generated by thermal stressing) are available
   to close; however, the porosity determined from post-thermal stressing pycnometric measurements suggests that no micro-
- 175 fracturing took place. We suggest that thermal stressing did indeed slightly modify the network of micro-fractures but on toosmall a scale to affect the UCS and pycnometry results. The manuscript has been edited to highlight this observation and changes have been made to section 3.2.1.
  - Line 400-401: as previously, it would by useful if the authors were to distinguish between % and vol. %.
- 180 We have amended this throughout, as previously mentioned.

Line 431: Young's modulus ought to be capitalised. Also, there is a full stop missing at the end of this sentence. -These typos have now been corrected in the manuscript.

185 Line 441-445: see comment above concerning these inclusion models.

-We have added analysis of these micromechanical models; see reply to comments above.

**Line 470: could this observation be due to pore anisotropy (e.g. Bubeck et al. 2017; Griffiths et al. 2017)?**

- Here the reviewer is referring to the statement "This suggests that these samples are stiffer than the other specimens tested, 190 and indeed those experiments reached unusually high peak stresses at relatively low strains to failure.", which is discussing samples UNZ-2 and UNZ-13. We looked at the thin section of UNZ-13 and conclude that the pore anisotropy may indeed contribute. Here the pores are preferentially aligned toward the principal stress direction (Fig.3) and so this is a likely case for stiffening. The manuscript has been amended at this line and in the conclusions section, see the reviewer's comment below. For UNZ-2 it is possible that these vesicles had a pore anisotropy that could have led to minor strengthening and an increase 195 in stiffness, like that seen by Bubeck et al. (2017) and Griffiths et al. (2017). Yet, our investigation was not sufficient to

constrain and explicitly state this as fact.

Line 480-481: do the authors have any information on the pore size distribution or pore anisotropy that could help explain this?

200 -The reviewer is referring to the following statement "In addition, thermally stressed samples have slightly lower ( $\sim 0.5-1.5$ GPa) Young's Moduli than their unstressed equivalents, as previously noted in dacites from Mt. St. Helens (Kendrick et al., 2013a). This highlights a potential change in

porosity distribution that was not recognised by other means (e.g. total porosity, strength)."

did not indicate any change - and the mechanical data which shows a decrease in Young's modulus.

205 The decrease in Young's Modulus is very slight, therefore information on pore size distribution or pore anisotropy, e.g. from a thin section, is unlikely to help in explaining this decrease. As the samples are very heterogenous any change in pore behaviour would be difficult to quantify as the change would be expected to be small and comparing the exact material before and after thermal stressing is not possible in thin section. This would rely on pre and post stressing CT scans which are time consuming and would likely provided reconstruction with spatial resolution too low to accurately distinguish slight pore 210 morphology changes. Hence, the most sensitive measure of change we have is the pycnometry - which as stated previously,

Line 525: shear-induced development of pore connectivity was also shown experimentally by Kushnir et al. 2017 EPSL. -We regret the oversight in not citing this article and have now added it to the manuscript.

**215**

225

Line 535-560: as with my previous comments concerning the inclusion models of Ashby and Sammis and Ashby, I believe the authors could extend this Deborah number

analysis some more. As far as I can tell, the dimensions of Equation 10 only balance out if b = 1. Pa = [Pa.s × s^-1] is fine (b = 1), but  $Pa = \sqrt{[Pa.s \times s^{-1}]}$  is not, as would be the case were b = 0.5. To me,

220 this highlights a serious shortcoming in the empirical approach adopted here, especially as there has been more recent work on the topic which adopt physical rather than empirical

parameters, for example the Wadsworth et al. 2017 chapter referenced elsewhere in the manuscript. For example, that chapter addresses the physical scaling of De with crystal

content. Ultimately, a similar physical approach could yield a much more generally applicable failure criterion for porous materials.

-With regards to the units of Equation 11, the equation balances if constant k has units of MPa.s1/2, as shown below:  $\sigma(MPa) = k(MPa.s^{1/2}).\dot{\varepsilon}_{obs}(s^{-1})^{b=1/2}$

In the Wadsworth et al. 2017 chapter, only the single or two-phase cases are discussed. Within this chapter, the equation for 230 the Deborah number is only given for a two-phase, crystal-bearing medium (Equation 3, Wadsworth et al., 2017). Here we are

- working with a three-phase medium, for which no models exist and thus an empirical approach must be adopted. For the sake of this argument, if we assume that the material from Mt. Unzen is a two-phase, crystal-bearing medium we can use Figure 2 in Wadsworth et al., (2017) to calculate the critical Deborah number expected. Given that the Unzen material has a crystal content,  $\phi_x$ , (microlites+phenocrysts) of ~0.75 we can use this to find the critical Deborah number if no bubbles were present. 235 To find this critical value, the maximum packing fraction,  $\phi_m$ , also has to be known. As  $\phi_m$  is defined as "the volume fraction of particles beyond which there is no space remaining which would accommodate further particles" (Mader et., 2013), it is clear from thin section and SEM images that our material has not yet reached  $\phi_m$  (see Fig. 3). Therefore, it can be assumed that for the Mt. Unzen material investigated here,  $\phi_m \approx 0.76 - 0.99$ , and  $\frac{\phi_x}{\phi_m} = 0.76 - 0.99$ . According to Figure 2 in Wadsworth et al. (2017), this range gives the range of critical Deborah number as 9.9x10-5-7.6x10-4. Therefore, our estimation of the critical
- 240 Deborah number of the dense material as  $9.4 \times 10^{-5}$  -6.6  $\times 10^{-4}$  is a very reasonable one indeed. The method used in this manuscript to find the Deborah number and then critical Deborah numbers is empirical but, as well as giving a working solution to find the Deborah number for a three-phase material, it shows that there is a reduction in critical Deborah number due to the addition of particles and also provides a linear relationship for the critical Deborah number and the addition of pores. This study is a novel, first-step approach into characterising the De number and failure constraints on 245
- real, volcanic samples using mechanical testing data.

**Line 535: using exp as a subscript is a little ambiguous (at first glance I presumed it signified an exponential).**

-For clarity, we have changed the exp subscript to obs to stand for observation in line with the symbol for observation time tobs.

250

255

Line 546: Oswald should presumably be Ostwald. -Yes, this has now be changed in the manuscript.

Line 546: here, the authors state values of k and b of 1653 and 0.5, yet in the caption for Figure 10, the values are k = 1606and b = 0.7755. Which of these are correct? What is the

sensitivity of the following analysis to variations on k and/ or b?

-We thank the reviewer for pointing out this typo. The correct k and b values were those written within the manuscript. These values shifted due to an addition of data to the plot. We have now added the standard error of estimate for these values to the plot to show the variation of k and b.

260

Equation 11: Based on equation 9, shouldn't  $\sigma$  relate to strain rate  $\times$  viscosity (or equivalently,  $De \times G \infty$ )?

-We believe the reviewers confusion lies with the presentation of the equations, Equation 8 has now been edited from:

 $De = \frac{\eta_m}{G_{\infty}t_{obs}}$

to:

265

 $De = \frac{\eta_m}{G_\infty t_{obs}}$ (now Equation 9) And Equation 9 from:

 $De = \dot{\varepsilon}_{obs} \eta_m / G_{\infty}$ to:

**270 $De = \frac{\dot{\varepsilon}_{obs} \eta_m}{G_{\infty}}$**

(now Equation 10)

To get to Equation 11 from Equation 9 a substitution is made for  $\dot{\varepsilon}_{obs}$ , which from Equation 10 is  $\dot{\varepsilon}_{obs} = \left(\frac{\sigma}{k}\right)^{1/b}$ . (now Equation 11)

275 Line 618: this is in contrast to existing theory, models, and experimental data. Perhaps this effect is masked in your data by sample heterogeneity? I would be wary of including this point as

a key conclusion of the study.

-The reviewer is referring to conclusion 3: The orientation of a vesicle may not necessarily have a discerning control on the strength of a rock, however it does have an influence on the strains reached at failure and, as such, the Young's Modulus. Here
we agree with the reviewer and this point has now been removed as a key conclusion, in light of the discussion regarding anisotropy in the reply to an earlier reviewer comment above made regarding line 470. The manuscript has also been edited to reflect this change.

Figure 4: there appears to be some data obscured by the legend in panels *a* and *b*.

-We thank the reviewer for his keen eye and have clarified the figure accordingly.

Figure 6: for clarity, perhaps the authors could plot viscosity as a function of strain rate (similar to Figure 10a; perhaps with symbols coloured for time).

-As the stress (which is proportional to viscosity) is plotted against strain rate on Figure 12b (previously Figure 10a). The b value obtained from the curve in Fig 12b is 0.5 which matched the values obtained from previous studies on crystalline dome material (Caricchi et al., 2007; Lavallée et al., 2007, 2012), therefore we feel that there is no more information to be gained by alternative plots.

Figure 10: in panel a, the authors state that the equation is shown on the figure, but it is missing. In panel b (and line 555), is the relation given by the yellow line based on only three data

(i.e. the transitional data)? What is the r2 value of this relation? Can the authors comment on the theoretical value of  $De_c c$  for a nonporous material? Would this fit on the trend? Likewise,

how do the authors anticipate *De\_c* evolving for highly porous materials?

- -We thank the reviewer for highlighting this point which has spurred a number of changes (also in light of reviewer 2's comments). We now refer the reviewer to the revised Figure 12 and additional manuscript changes which have been edited to show two separate transitional regimes: viscous-dominated and brittle-dominated. Figure 12 shows clearly the evolution of failure and at which critical Deborah numbers this occurs. Figure 12c shows the standard error of estimate windows of this data and their R2 values.
- For a non-porous material, the data suggest the Critical Deborah number would lie between  $De_c = 1 \times 10^{-4} 6.6 \times 10^{-4}$ . This is 305 approximately two orders of magnitude lower than that reported by Webb and Dingwell (1989), which we attribute to the high crystal content as crystals also decrease this critical Deborah number (Cordonnier et al., 2012; Wadsworth et al., 2017), see also the reply to reviewer's comments on lines 535-560 of the manuscript.

For the porosity range of material tested herein, we expect the critical Deborah number to follow a linear trend as shown in Figure 12c.

310

**Response to Quane**

315

320

This paper is potentially an excellent contribution to the experimental literature on natural, complex, multi-component, volcanic rocks. Initially, the authors provide an excellent treatment of the relevant rheological behavior of high-temperature deformation in glassy samples which is followed by a good characterization of representative, pre-experiment cores. The study comprises a substantial number of experiments both at ambient room (20 °C) and magmatic temperatures (900°C). The rheological data is of high quality and the experimental conditions and span a reasonable range of deformation rates and timescales. The authors do an excellent job of presenting the results separately from analysis and a thorough job of explaining the major decisions and assumptions they had to make in the process of running the experiments (i.e., how and why samples were chosen based on porosity, connected porosity, etc., detail on how experimental charges were loaded and the effect that has on the experiment). The rheological analysis is solid and prima facie, the interpretations seem sound and lead to a number of logical conclusions about the behavior of these multicomponent systems under conditions relevant to Unzen eruptions.

**-We thank the reviewer, Steve Quane, for his concise summary and descriptive comments which are answered below.**

- 325 However, I see one main oversight in their otherwise detailed and robust analysis; there are no descriptions, photos, or representative images and quantitative measurements (porosity, density) of experimental run products. This is a major issue for several reasons:
- a) Deformation in experimental charges cannot be interpreted by using the rheological data alone. For example, seemingly
   "viscous" behavior and "brittle" behavior were interpreted (starting is section 4.1) based on the "mechanical responses" of the rocks. The authors make assumptions and "attributions" about the actual mechanical behavior of the samples with no empirical evidence. For example, they "attribute to a narrowing of pre-existing cracks" and "hypothesize may reflect a contribution of viscous deformation upon loading". It is possible that these interpretations are correct, however, it would be relatively eas y to test the attributions and hypotheses by halting an experiment at the requisite place on the deformation path and doing microstructural analysis.
- 555 millionasia analys

-We thank the reviewer for bringing this point to our attention. With reference to the high temperature viscous responses we refer the reviewer to work by Cordonnier et al., (2012), referenced in the text, who look, in detail, at the mechanical response curves of deformed samples and label them 'viscous', 'transitional' and 'brittle' according to X-ray microcomputed tomography scans of deformed samples. We particularly refer to Figures 1 and 2 in Cordonnier et al., (2012) – as such we are not defining new regimes but simply categorising our samples according to regimes already defined. With reference to the brittle experiments, we refer the reviewer to publications mentioned in the text (Hoek and Bieniawski, 1965; e.g. Brace et al., 1966; Scholz, 1968; Heap et al., 2014) who describe, in detail, the four stages of mechanical loading and brittle failure with reference to the mechanical stress-strain curves. Although the authors agree that 'halting an experiment at the requisite place on the deformation path and doing microstructural analysis' would be a very informative study, this work has already previously been tackled, stress-strain curves have been dissected with respect to sample attributes, and for brittle experiments, the regimes of crack-closure, elastic deformation, strain hardening, and failure are well defined and identifiable by the mechanical curves. Therefore, we reassure the reviewer that care has been taken when labelling a response as 'viscous' or 'brittle' by referring to previous work in rock and lava deformation, based on their stress-strain curves.

350

That said, we understand the need to examine the experimental run products and have taken the reviewers concerns on board. We have illustrated our examination of the experimental products acquired by SEM imaging to create a new Figure 5, along

with further descriptions and photographs of samples after deformation to Figure 12 (previously Figure 10). This analysis, based on the reviewers' comments made us revisit our labelling of samples, and we have issued a new comprehensive, visual 355 description of failure, as seen in Figure 12. We also further concluded that the state of 'transitional' can be further sub-divided to clearly express that it is a spectrum leading from viscous behaviour, indicated from a continuous plateau in stress with substantial strain, to brittle behaviour, defined by a sharp drop in stress with little strain beyond an initial elastic loading response. Therefore, we suggest that a sample can either be in the viscous dominated regime while undergoing a transitional behaviour, where the stress plateaus with strain but there are small stress drops along the way, or the brittle dominated regime 360 where the stress-drop is poorly defined and 'curves' before reaching high strain at failure (Figure 5; Figure 11). Although the post deformation photographs and SEM images are a useful guide, as the same strains before experiment termination/sample failure were not met by every sample (we chose our end strains based on characteristics of the stress-strain curves not on a set total strain) the results are not entirely comparable e.g. a viscous sample experiment would be terminated at much shorter strains (than a transitional sample) as its curve was already defining viscous behaviour. Thus, we consider the stress-strain curves a better method for quantitatively of defining the deformation mode of lavas. 365

b) Post experiment analysis of end products can lead to surprising conclusions about mechanical behavior. In these multicomponent systems, deformation can occur via several mechanisms. Bubble collapse, brittle fracturing, viscous flow of groundmass glass, microcracking, rotation of grains, grain boundary sliding, internal grain deformation. All of these are factors in accommodating strain in the samples. Hence, strain can be accommodated homogenously (throughout the sample evenly) or it can be localized into bands or disparate parts of the sample. Without post experiment analysis, these important rheological behaviors cannot be determined. The authors are making the most logical conclusions about their "brittle" and "viscous" determinations based on the rheological data, however, without visual analysis of bulk properties and microstructures, the authors cannot confirm behavior. In addition, they are losing a considerable amount of important information about the nature of the deformation.

-We thank the reviewer for his comments and agree, understanding the complex mechanisms that led to failure in volcanic rocks is important, yet, here, this paper is not trying to decipher the deformation mechanism (e.g., viscous, plastic, brittle) but 380 the deformation mode of lavas (i.e, ductile vs brittle) necessary to constrain (and distinguish between) flow and fragmentation processes. [Please note that the distinction between the two is that of scale: a deformation mode refers to the macroscopic character of sample deformation whereas a deformation mechanism refers to microscopic deformation processes. Thus, unfortunately in this field of laboratory testing, brittle may be used when refereeing to both a deformation mode (sample failure) and a deformation mechanism (i.e., a cracking event); see also Rutter in Tectonophysics (1986) and Heap et al. in Bull. 385 Volc. (2015) for clarity] We have conducted further analysis of the experimental products as described above. We also guide the reviewer to Figure 2 in Lavallee et al., (2007) where post-experiment textures have been viewed and deformation mechanisms discussed, and to Figure 2 in Kendrick et al., (2013), as well as Figure 2 in Kendrick et al., (2017), where textural evolution with strain is depicted and the deformation mechanisms are interpreted. Appreciating the need for a more in-depth explanation of the overarching deformation mechanisms in the deformation mode discussed (e.g. 'brittle' and 'viscous' and offer the reviewer the new Figure 12 (previously Figure 10) with accompanying edits in the manuscript. A detailed study of 390 the exact microstructural deformation mechanisms at play across all samples is beyond the scope of this paper seeking to constrain deformation mode (not mechanism), and as it has already previously been discussed in other studies, we chose to highlight samples representative of each regime and map the textures associated with the different deformation regimes and link these to the stress-strain curve characteristics used to define the remaining samples.

c) Post experiment analysis of physical properties (i.e., density, porosity) can yield important information on the nature of deformation. Certainly, for the cores that were not destroyed during brittle failure, the authors can make density and porosi ty determinations via the methods they used on the pre-experiment cores. Bulging of cores may cause a little consternation, however, established methods exist in the volcanology literature to measure density and porosity on irregular samples.

400

-We advise the reviewer that samples that remained completely intact (only those with a completely 'viscous' response, i.e. those carried out at strain rates of  $10^{-5}$  s-1) were re-measured to constrain changes in connected porosity. However, the results showed no significant change in porosity, nor in the volume of the sample determined by pycnometry (Table 1); hence, we mention this in the text but do not present the data in the study. Due to minor loss of volume from the experimental process (removal of sample from pistons etc.) the pycnometer readings are within error and thus we concluded, cannot be considered.

405

Table 1. Example of volume measurements made using pycnometery on samples that remained intact after deformation. Fractional change in volume is  $< \pm 0.05\%$  of the measured volume

| Sample   | Initial  | Strain rate               | Temperature | Measured                  | Measured     | Fractional |
|----------|----------|---------------------------|-------------|---------------------------|--------------|------------|
|          | porosity | tested (s -1 ) | tested (°C) | volume                    | volume after | change in  |
|          |          |                           |             | before (cm 3 ) | (cm³)        | measured   |
|          |          |                           |             |                           |              | volume     |
| UNZ-4-16 | 0.12     | 1.00E-05                  | 900         | 12.17                     | 12.14        | 0.003      |
| UNZ-4-17 | 0.12     | 1.00E-05                  | 900         | 11.67                     | 11.72        | -0.005     |
| UNZ-8-16 | 0.18     | 1.00E-05                  | 900         | 11.38                     | 11.32        | 0.005      |

410

415

d) Characterizing the amount of strain in the samples is an independent measure of machine strain. Does the sample show the same amount of strain as the machine? This can be determined through post-experiment analysis of density, porosity and core geometry. It is an important check on the experimental apparatus to ensure all strain from the machine is going into the sample. Quane and Russell, 2005 (cited by authors) and Quane et al., 2004 from American Mineralogist go through these procedures in detail.

We refer the reviewer to the first paragraph in section 2.3. of the manuscript: '[Note: all mechanical data have been corrected for the compliance of the setup, quantified via Instron procedures that monitor length changes due to loading of the pistons in contact with one another]'. This compliance method is carried out at all temperatures tested in our laboratory. Following the application of the compliance correction, the total strain referred to in the manuscript is the sample strain and not machine strain. Post-deformation sample geometry (i.e. final sample length, for the in-tact samples) was measured for the samples to

420 strain. Post-deformation sample geometry (i.e. final sample length, for the in-tact samples) was measured for the samples to confirm final strains were correct. This point has been added to the manuscript, as well as "...at the relevant experimental temperature" in the sentence describing compliance, to clarify that the different behaviour of the machine at temperature is also accounted for. To clarify, the method to quantify strain in deforming porous samples in Quane et al., (2004) and Quane and Russell (2005) may be applied for glass-bead compacts, but unfortunately not for natural multi-phase material.

425

Without post-experiment characterization (on samples that will allow it-sometimes even brittle deformation samples can be salvaged and epoxyed), the authors cannot speak with authority on the types of deformation occurring. Unfortunately, by not having that authority, the Conclusions they draw come into question. Certainly, the authors can do an analysis of the run products and produce a figure or two (like Figure 3 does for pre-experiment cores) to describe the major mechanisms of

**430 deformation and strain accommodation. Without this, this otherwise very strong, methodical and detailed contribution falls incomplete.**

-As mentioned above, this study is first and foremost concerned with a description of the macroscopic deformation modes of lava, not the deformation mechanism. Yet, we fully agree that textural information provides insight into the underlying 435 microscopic deformation mechanism. We draw the reviewer's attention to the newly created Figure 5 and Figure 12 a). Due to the fragmental nature of the samples, particularly those marked as having a brittle or brittle-dominated response, it was impracticable to reconstruct the position of each fragment with epoxy, yet we looked at some fragments (new Figure 5) taken from the inner part of the sample. We provide new data in Figure 10 (now Figure 12), containing photographs of the runproducts and accompanying comments in the manuscript. With the photographs, the now more detailed explanation of the curves, and the SEM images in Figure 5, we believe we have satisfied the reviewers concerns about sample characterisation.

440

**Technical corrections in this manuscript a very minimum. Found one spelling mistake, but I lost it!**

445 -We thank the reviewer for searching the document for typos, we have found the assaulting spelling mistake mentioned and have track changed it in the manuscript.

**450 List of relevant changes**

The following manuscript has been edited with tracked changes to indicate what amendments have been made after reviewer's comments. Minor text edits have been made throughout the manuscript to aid with flow and to correct for typing issues, major changes include:

- The addition of a translated abstract from English to Japanese
- 455 The addition of references suggested by reviewers
  - The extension of our analysis of our data with comparison to the pore-emanating crack model and wing-crack model, this includes text additions to sections 1.3 and 4.1.2 and the addition of a new figure, Figure 11.
  - The addition of an accessible MATALB code for the calculation of Young's Modulus, found at:
  - https://doi.org/10.5281/zenodo.1287237
    A more detailed description of the rheological response of the dome lavas, after comments by Steve Quane. This involved carrying out more SEM work, as seen in Figure 6, and the addition of text in section 3.2.2
    - An updated description of the effect of pore anisotropy on rock strength, section 4.1.2, after comments by Jamie Farquharson
  - An updated description of the failure criterion of porous lavas in section 5.2. This included the newly updated Figure 10, which is now labelled Figure 12.
  - Edits were made to Figure 3, Figure, 9 (Now Figure 10), and Figure 4 for consistency

The manuscript now includes 12 figures, 6 supplementary figures, and  $\sim$ 13,000 words excluding references and figure captions.

470

460

**Failure criteria for porous dome rocks and lavas: a study of Mt. Unzen, Japan**

Rebecca Coats1, Jackie E. Kendrick1, Paul A. Wallace1, Takahiro Miwa2, Adrian J. Hornby1,3, James D. Ashworth1, Takeshi Matsushima4, Yan Lavallée1

1Department of Earth, Ocean and Ecological Sciences, The University of Liverpool, Liverpool, L69 3GP, UK
 2National Research Institute for Earth Science and Disaster Prevention, Ibaraki, 305-0006, Japan
 3Now at Department of Earth and Environmental Sciences, Ludwig-Maximilians-Universität München, Munich, 80333, Germany

4Institute of Seismology and Volcanology, Kyushu University, Nagasaki, 855 0843, Japan

480 Correspondence to: Rebecca Coats (r.coats@liverpool.ac.uk)

**Abstract**

The strength and macroscopic deformation mode (brittle vs ductile) of rocks is generally related to the porosity and pressure conditions, with occasional considerations of strain rate. At high temperature, molten rocks abide to Maxwell's viscoelasticity and their deformation mode (brittle vs ductile) is generally defined by strain rate or reciprocally, by comparing the relaxation 485 timescale of the material (for a given condition) to the observation timescale - a dimensionless ratio known as the Deborah (De) number. Volcanic materials are extremely heterogeneous, with variable concentrations of crystals, glass/ melt and vesicles (of different sizes), and a complete description of the conditions leading to flow or rupture as a function of temperature, stress and strain rate (or timescale of observation) eludes us. Here, we examined the conditions which lead to the macroscopic failure for of variably vesicular (0.09-0.35-%), crystal-rich (~ 75 vol.%), pristine and altered, dome rocks (at ambient temperature) 490 and lavas (at 900 °C) from Mt. Unzen Volcano, Japan. We found that the strength of the dome rocks decreases with porosity and is commonly independent of strain rate; when comparing pristine and altered rocks, we found that the precipitation of secondary mineral phases in the original pore space alteration caused minor strengthening. The strength of the lavas (at 900 °C) also decreases with porosity. Importantly, the results demonstrate that these dome rocks are weaker at ambient temperatures than when heated and deformed at 900 °C (for a given strain rate resulting in brittle behaviour). Thermal stressing 495 (by heating and cooling a rock up to 900 °C at a rate of 4 °C min-1, before testing its strength at ambient temperature) was

found not to affect the strength of rocks. In the magmatic state (900 °C), the rheology of the dome lavas is strongly strain rate dependent. Under low experimental strain rate conditions ( $\leq 10^4$  s-1) ductile deformation dominated the lavas behaved ductilly (i.e., the material sustained substantial, pervasive deformation) and displayed a non–Newtonian, shear thinning behaviour. In this regime, the apparent viscosities of the dome lavas were found to be essentially equivalent, independent of vesicularity, likely due to the lack of pore pressurisation

- the dome lavas were found to be essentially equivalent, independent of vesicularity, likely due to the lack of pore pressurisation and efficient pore collapse during shear. At high experimental strain rates ( $\geq 10^4 \text{ s}^{-1}$ ) the lavas displayed an increasingly brittle response (i.e., deformation resulted in failure along localised faults); we observed an increase in strength and a decrease in strain-to-failure as a function of strain rate. To constrain the conditions leading to failure of the lavas, we analysed and compared the critical Deborah number at failure ( $De_c$ , the ratio between the relaxation time and the experimental observation time) of these lavas to that of pure melt ( $De_{melt}=10^{-3}-10^{-2}$ ; Webb & Dingwell, 1990). We found that the presence of crystals
- decreases  $De_c$  to between  $6.6 \times 10^4$  -2.11×104. The vesicularity ( $\varphi$ ), which dictates the strength of lavas, further controls  $De_c$  following a linear trend-5.1×104 $\varphi$ +2.11×104. We discuss the implications of these findings for the case of magma ascent and lava dome structural stability.

| 510 | 多孔質な岩石及び溶岩の破壊基準:雲仙火山溶岩ドームでの研究                                                                                                                                                                                                                                                                                                       |
|-----|--------------------------------------------------------------------------------------------------------------------------------------------------------------------------------------------------------------------------------------------------------------------------------------------------------------------------------------------|
| 515 | マグマ(溶岩)と岩石のレオロジーと強度は、応力の蓄積と散逸を支配し、噴火様式や山体の構造的安定性に影響
を与える、火山噴出物は極端に不均質であり、様々な量・サイズの結晶、ガラス(メルト)、気泡を含む、そのため
、温度・応力・歪速度の関数として、その流れや亀裂形成を引き起こす状態を完全に記載することは難しい、こ
こで我々は、雲仙火山において溶岩ドームを形成し様々な発泡度(9-35%)を有する高結晶度(~75%)な岩石(常温)
と溶岩(900度)について、その破壊を引き起こす状態を検討した。その結果、我々は岩石の強度は空隙率とともに                                                    |
| 520 |  <li>減少し、金速度に依存しないことを発見した: 新鮮な岩石と変負したものでは、夜者でわすかに強度か大さい</li> <li>. また、溶岩(900°C)の強度も空隙率とともに減少する. この結果は重要なことに、脆性的振る舞いを起こす歪</li> <li>速度において、常温における岩石の強度は、それを900℃まで加熱し変形させたときの強度よりも弱いことを示</li> <li>している. このとき、熱応力は岩石の強度に影響を与えない.</li> <li>高温条件(900°C)では、溶岩のレオロジーは歪速度に強く依存する. 低歪速度下(<104 s4)では、溶岩は塑性的に振</li>  |
| 525 | る舞い(物質が広範な固体変形を持続させる),非ニュートン流体としてずり粘減の振る舞いを示した.このレジ
ームでは,溶岩の見かけ粘性は,おそらく剪断時の効率的な空隙崩壊のため,発泡度に依存しない.高歪速度下
(>10 -4 s -1 )では,溶岩は益々の脆性的な応答(局所的な断層に沿った破壊による変形)を示す;歪速度の関数として,
強度の増加と破壊へ至るときの歪の減少が観察された.溶岩の破壊を引き起こす状態を制約するため,これら溶
岩における破壊時の臨界デポラ数(De, 緩和時間と実験観察時間の比)を解析し、メルトにおけるそれ(De-no =10 -3 -      |
|     | 10 -2 ; Webb & Dingwell, 1990)と比較した. 我々は結晶の存在がDecを6.6×10 4 –1×10 4 まで減少させることを発見し
た. またさらに,溶岩の強度に影響する発泡度( $\varphi$ )もDecを線形傾向のようにコントロールする. 我々はこれらの
発見が与える,マグマ上昇と溶岩ドームの構造的安定性への示唆を議論する.                                                                                                              |

[revised manuscript text omitted]

$$De = \frac{\varepsilon_{abs}}{G_{co}} \frac{\eta_m}{G_{co}}$$

Magmatic suspensions, like those described in this study, are non-Newtonian materials with a shear thinning response (Caricchi et al., 2007; Lavallée et al., 2007; Cordonnier et al., 2009; Avard and Whittington, 2012; Vona et al., 2013), hence their viscosity is strain rate dependent. It has previously been described that the peak stress,  $\sigma$ , shares a power law relationship with strain rate,  $\dot{\varepsilon}_{exp}$ , via:

125
$$\sigma = k \dot{\varepsilon}_{obsexp}^{b}$$

where k is the flow consistency index (in Pa.s) and b is the flow behaviour index, describing the rheology of the fluids (Ostwald, 1925; Lavallée et al., 2007). For Newtonian bodies b = 1, but for shear thinning suspensions, b decreases below 1 (Caricchi et al., 2007) and reaches a minimum of b = 0.5 for crystal-rich materials (Lavallée et al., 2007; Cordonnier et al., 2009). In the present study the Mt. Unzen dome material tested at 900 °C, by fitting a power law to the peak stress-strain curve we obtained Ostwald constants of k = 1653 and b = 0.5 (Fig. 10a12b). So, we can rewrite Eq. 910, using Eq. 1011, to obtain:

$$\mathbf{D}\mathbf{e} = \frac{\left(\sigma_{/k}\right)^{1/b} \eta_{\mathrm{m}}}{\mathbf{G}_{\infty}},\tag{11}$$

which permits the representation of the Deborah number of material failure as a function of strength (which was shown to be dependent on porosity), for a given temperature (and thus interstitial melt viscosity). For our samples, the interstitial melt viscosity can be estimated at 109.42 Pa.s (using its chemistry and experimental temperature as an input parameter in the GRD 1135 viscosity calculator (Giordano et al., 2008)). In Figure 10Figure 12bc, we present the data using symbols that illustrate the response of the samples; whether it flows or fails near instantaneously, or after some amount of strain. The onset of transitional behaviour, termed viscous-dominated transitional, is marked by the red line. Similarly, the onset of brittle behaviour, brittledominated transitional, is marked by the yellow line. These lines are linear regressions on a semi-log space plot, with their standard error of estimates marked by faded colour windows. Any point that plots between the red and yellow lines would be 140 termed transitional and could demonstrate any type of hybrid behaviour. Above a porosity of 0.27 no transitional zone occurs, and behaviour would be classified as either viscous or brittle. This analysis demonstrates that the critical Deborah number, Dec, which indicates the initiation of rupture, inof dome lavas from Mt. Unzen decreases by just over half analmost one order of magnitude over a 0.35 difference range in porosity; from ~  $7.652 \times 10^{-54}$  in the densest sample measured to  $4.18 \times 10^{-5}$  
[revised manuscript text omitted]

---

## Referee Report (RR1)

Review of "Failure criteria for porous dome rocks and lavas: a study of Mt. Unzen, Japan"

Coats and her co-authors have addressed the majority of my previous comments satisfactorily, either by well-argued rebuttal or by making amendments to the text, for which I applaud them. In particular, they have improved the clarity of the manuscript in many parts, and performed a more in-depth interrogation of their data using the various micromechanical damage models discussed in the text.

I only have one outstanding concern, which relates to the balancing of units in Eq. 11. As highlighted in my original review, the units (as stated) do not balance out if $b$ does not equal 1. This is a fundamental problem stemming from the use of an exponent model. The authors counter this comment by couching their constant $k$ in units of $\sqrt{Pa.s}$ (i.e. $Pa.s^b$). While this is not particularly satisfactory (the "flow consistency index" has an ambiguous physical meaning if it is not in measurable units, i.e. Pa.s as it is currently explicitly defined in the manuscript), it does solve the immediate unit balancing problem. However, it is not a suitable solution as later in the manuscript their non-unity value of $b$ appears again (in the Deborah number equation). The authors indicate that

$$De = \frac{(\sigma/k)^{1/b}}{G_\infty} \eta_m.$$

If the authors use units of $Pa.s^b$ to define $k$, then the units balance thus:

$$De = \frac{\left(Pa/\sqrt{Pa.s}\right)^2}{Pa} Pa.s$$

when $b = 0.5$, which is to say $De = Pa$. The Deborah number is a dimensionless ratio (a timescale divided by a timescale), so presenting it in units of pressure is clearly not desirable, and I'm sure was not the authors' intention. Moreover, this assumes that $b$ is a "neat" fraction, so that $1/b$ is resolved into an integer and the degree of the $k$ radical is also an integer. Things become more complex if $0.5 < b < 1.0$.

I urge the authors to look more critically at this problem, and perhaps reconsider the use of a power-law model, which propagates problems when incorporated into more involved analyses. Failing this, the authors should at least take care that their representation of $k$ and $b$ do not lead to errors later in the manuscript. For example, defining a critical strain rate $\lambda$ such that $\lambda = 1$ s$^{-1}$, viscosity could be presented as

$$\eta_A = k \left(\frac{\dot{\epsilon}}{\lambda}\right)^{b-1}$$

such that the units balance out without the need to redefine $k$:

$$Pa.s = Pa.s \left(\frac{s^{-1}}{s^{-1}}\right)^{0.5} \rightarrow Pa.s = Pa.s\sqrt{1}.$$

I acknowledge that this may not be a perfect (or even correct) solution, but it may be a useful avenue for the authors to explore. In any case, the authors ought to discuss some of the shortcomings of their power-law approach.

This point aside, I recommend this article for publication in *Solid Earth*.

Yours faithfully,

Jamie Farquharson

---

## Author Response (AR2)

**Point-by-point response to the reviews**

**Response to Farquharson #2**

Coats and her co-authors have addressed the majority of my previous comments satisfactorily, either by well-argued rebuttal or by making amendments to the text, for which I applaud them. In particular, they have improved the clarity of the manuscript in many parts, and performed a more in-depth interrogation of their data using the various micromechanical damage models discussed in the text. I only have one outstanding concern, which relates to the balancing of units in Eq. 11. As highlighted in my original review, the units (as stated) do not balance out if $b$ does not equal 1. This is a fundamental problem stemming from the use of an exponent model. The authors counter this comment by couching their constant k in units of $\sqrt{Pa.s}$ (i.e. $Pa.s^b$). While this is not particularly satisfactory (the "flow consistency index" has an ambiguous physical meaning if it is not in measurable units, i.e. Pa.s as it is currently explicitly defined in the manuscript), it does solve the immediate unit balancing problem. However, it is not a suitable solution as later in the manuscript their non-unity value of b appears again (in the Deborah number equation). The authors indicate that

$$De = \frac{(\sigma/k)^{1/b}\eta_m}{G_\infty}$$

If the authors use units of $Pa.s^b$ to define k, then the units balance thus:

$$De = \frac{\left(Pa/\sqrt{Pa.s}\right)^2}{Pa} Pa.s$$

when b = 0.5, which is to say $De$ = Pa. The Deborah number is a dimensionless ratio (a timescale divided by a timescale), so presenting it in units of pressure is clearly not desirable, and I'm sure was not the authors' intention. Moreover, this assumes that b is a "neat" fraction, so that 1/b is resolved into an integer and the degree of the k radical is also an integer. Things become more complex if 0.5 < b < 1.0. I urge the authors to look more critically at this problem, and perhaps reconsider the use of a power-law model, which propagates problems when incorporated into more involved analyses. Failing this, the authors should at least take care that their representation of k and b do not lead to errors later in the manuscript. For example, defining a critical strain rate λ such that λ = 1 s⁻¹, viscosity could be presented as

$$\eta_A = k\left(\frac{\dot{\epsilon}}{\lambda}\right)^{b-1}$$

such that the units balance out without the need to redefine k:

$$Pa.s = Pa.s\left(\frac{s^{-1}}{s^{-1}}\right)^{0.5} \rightarrow Pa.s = Pa.s\sqrt{1}$$

I acknowledge that this may not be a perfect (or even correct) solution, but it may be a useful avenue for the authors to explore. In any case, the authors ought to discuss some of the shortcomings of their power-law approach. This point aside, I recommend this article for publication in Solid Earth.

Yours faithfully,

Jamie Farquharson

-We thank the reviewer for taking the time to review the manuscript for a second time. We acknowledge his concerns over unit balance and refer him to the reference Jahangiri et al., 2012, now referred to in the manuscript. Here it is stated that the Ostwald–de Waele or power-law fluid relationship is defined as: $\tau = K\gamma\,n$, where $\tau$ is shear stress (Pa); $K$ is the flow consistency

index (Pa.s$^n$); $\gamma$ is the Shear rate (s$^{-1}$) and $n$ is the flow behaviour index. The reviewer has inadvertently taken the units of k as $(Pa.s)^b \neq Pa.s^b$. We have found a technical error in our text on line 610 which stated that the unit of k were Pa.s, this has now been amended so that the unit of k are defined as Pa.s$^b$.

This simple unit correction means that the units of Equation 12, the Deborah number equation, balance as follows:

$$ De = \frac{(\frac{Pa}{Pa.s^b})^{1/b}}{Pa} Pa.s \;\rightarrow\; De = \frac{(\frac{1}{s^b})^{1/b}}{Pa} Pa.s \;\rightarrow\; De = \frac{s^{-1}}{Pa} Pa.s \rightarrow De = 1 $$

Thus, the Deborah number is dimensionless as expected.

We thank the reviewer for his efforts which have much improved our manuscript.

Jahangiri, P., Streblow, R. and Müller, D.: Simulation of Non-Newtonian Fluids using Modelica, Proc. 9th Int. Model. Conf., 57–62, doi:10.3384/ecp1207657, 2012

**Response to Farquharson**

Dear Editor and authors,

I have read the manuscript *Failure criteria for porous dome rocks and lavas: a study of Mt. Unzen, Japan* with great interest. By way of a targeted experimental campaign, Coats *et al.* map

out the failure conditions for suites of variably-porous crystalline andesite as a function of different temperatures and strain rates. The conclusion of the study is an empirical threshold for the

failure of these materials derived from their data.

The article is assiduous and well written, the experimental protocol appears rigorous, and the study yields a wealth of interesting new data. Overall, this article represents a commendable

research effort from the authors. Where the authors perhaps do themselves a disservice is in the analysis of their data, which could be more comprehensive. Below I outline minor

comments or concerns that I feel the authors should address or clarify. Pending these changes, I recommend this article for publication in *Solid Earth*.

Yours faithfully,

Jamie Farquharson

(Please forgive the clumsy formatting, I was forced to write this report on my phone)

Lines 39-56: Sparks (1997) is highly relevant to this study, and is a surprising omission here.

Sparks (1997) The causes and consequences of pressurisation in lavas done eruptions. *Earth and Planetary Science Letters* 150(3-4): 177-189

- We thank the reviewer for bringing this to our attention and have added the reference in question, an oversight given its relevance to the fundamentals to this study.

Line 66: how do the authors define the "temperature range of interest"?

75    - Webb and Dingwell (1989) compiled a large dataset of elastic modulus at infinite frequency, $G_\infty$, for silicate melts and many other compounds, over a range of temperatures. Previous studies found that $G_\infty$ ranges from 5 to 42 GPa at room temperature for glasses with silica contents ranging between 5-99 mole% (Bansal and Doremus, 1986), and at temperatures between 400 and 1600 °C, $G_\infty = 33$ GPa $\pm 5\%$ (Bucaro and Dardy, 1974). Webb and Dingwell (1989) added data to find that for silicate melts and glasses, $G_\infty$ ranges between 3.2 and 32 GPa and thus can be approximated at $10^{10\pm0.5}$

80    as it only weakly varies with temperature (unlike viscosity). This brings an important simplification to the modelling of viscoelastic melts, which has been pivotal in its integration to volcanology. Therefore, we can assume $G_\infty = 10^{10\pm0.5}$ for all silicate melts (and glasses) at ~20-1600 °C, which extends beyond the temperature range of most (contemporaneous) volcanic systems reported in the literature. Hence, the "temperature range of interest" is all volcanic temperatures.

85

Line 101: it might be useful to provide the equation for *Ca* here.

- The paragraph details our knowledge of complex lavas (such as dome lavas) containing crystals and bubbles. The capillary number is highly relevant in aphyric lavas, but in the lavas discussed here, it remains to be adapted, which is beyond the scope

90    of this study. Hence, we wish not to introduce the equation here and believe a qualitative description is sufficient.

Lines 134-144, and elsewhere: the authors describe two inclusion models which they highlight may explain their data: the pore-emanating crack model of Sammis and Ashby, and the sliding

95    wing crack model of Ashby and Sammis. However, the authors do not go on to employ either of these models subsequently. As I intimated previously, it seems something of a shame that

there is not a more involved analysis of these data. Analytical solutions for both these models are provided by Zhu *et al.* 2010 *JGR* and Baud *et al.* 2014 *IJRMS*, respectively. Previous

authors have utilised one or other in order to describe the failure behavior of volcanic materials or analogues, for example Zhu

100    *et al.* 2011 *JGR* (for tuffs), Vasseur *et al.* 2013 *GRL* (sintered

glass), Heap *et al.* 2014 *JGR* (andesite), Zhu *et al.* 2016 *JGR* (basalts). Moreover, Zhu *et al.* 2011 extend the analytical solution to a dual-porosity medium, and Zhu *et al.* 2016 combine both

models so as to have a representative element volume comprised of an effective medium including a pore surrounded by many cracks. If Coats *et al.* were to interrogate their data in a

105    similar manner, they may be able to glean valuable information about the governing microstructural elements in their samples contributing to failure (for example by contouring for different

values of $K_{Ic}/\sqrt{[\pi r]}$).

-We thank Dr Farquharson for his detailed comments. Indeed, we had previously investigated the comparison of our data with the pore and wing-crack models but had excluded it from the results based on the assumptions of parameters we had to make

110    in fitting the wing crack model. However, as the reviewer asked for this data we felt it important to extend our analysis. The following has been added to the manuscript in section 1.3:

*An analytical estimation of this model was derived by Zhu et al., (2010) to estimate the uniaxial compressive stress ($\sigma$) of a sample, with an average pore radius ($r$), as a function of its porosity ($\varphi$) and the fracture toughness ($K_{IC}$):*

115    $\sigma = \frac{1.325}{\varphi^{0.414}} \frac{K_{IC}}{\sqrt{\pi r}},$          *(2)*

*The analytical approximation for this model was developed by Baud et al., (2014):*

$$\sigma = \frac{1.346}{\sqrt{1+\mu^2}-\mu} \frac{K_{IC}}{\sqrt{\pi c}} D_0^{-0.256}, \quad (3)$$

*where $\mu$ is the friction coefficient of the crack, $c$ is the half–length of a pre–existing crack, and $D_0$ is an initial damage parameter (which takes into consideration the number of cracks per unit area and their angle with respect to the principal stress).*

And the following in section 4.1.2:

*The uniaxial compressive strength was calculated for the samples for both the pore–emanating crack model of Sammis & Ashby (1986) (Eq. 3) and the sliding wing crack model of Ashby & Sammis (1990) (Eq. 4). For the former, the uniaxial compressive strength was calculated with varying values of $\frac{K_{IC}}{\sqrt{\pi r}}$ from 5 MPa to 25 MPa (Fig. 11). For the latter, approximate values for $\boldsymbol{\mu}, \frac{K_{IC}}{\sqrt{\pi c}}$ and $D_0$ were taken from Table 3 in Paterson and Wong (2005) as 0.51, 20–30 MPa and 0.3–44, respectively. This gave a range of estimated strength between 54 and 90 MPa (Fig. 11). At higher porosities, > 0.25, the pore–emanating crack model with $\frac{K_{IC}}{\sqrt{\pi r}}$ = 5–10 MPa seems to fit the data well, whereas for most rocks with porosities of 0.12–0.2 $\frac{K_{IC}}{\sqrt{\pi r}}$ = 10–15 MPa is a better fit. This could be explained by a decrease in the pore radius at these porosities, leading to higher values of $\frac{K_{IC}}{\sqrt{\pi r}}$, though, as the samples are heterogeneous and pore radius variability is high we cannot observe this (Figure 3). For the densest rocks in the study (~0.08–0.12), the UCS data would suggest yet a higher $\frac{K_{IC}}{\sqrt{\pi r}}$ of 20–25 MPa. The pore–emanating crack model could explain this switch in behaviour if there was a fundamental change in pore radius. However, the switch could also be explained by a transition in failure mechanism from pore–emanating cracks to wing cracks, meaning the wing–crack model would be more applicable. Alternatively, it may be a complex combination of the two. Although the solutions to the sliding wing–crack model are non–unique, as there are few experimentally constrained parameters, when combined with information gained from the pore structures (Fig. 3), the results of the modelling presented (Fig. 11) give us an insight into the dominant micromechanical failure mode of our samples. It is likely that the complex pore structures of these lavas, generated by a combination of vesiculation, deformation and cooling-driven contraction require an as-yet undefined combination of the two models. The weighting towards one or the other, however indicates that for the higher porosity specimens the behaviour of failure could be described using the pore–emanating crack model of Sammis & Ashby (1986), whereas in the lower porosity samples deformed in uniaxial compression, the main failure mechanism is explained by the sliding wing–crack model of Ashby & Sammis (1990).*

The following figure and caption were also added to the manuscript as new Figure 11:

[Figure]

**Figure 11 Plot of uniaxial compressive stress against porosity showing the ambient temperature mechanical data (black dots), along-side contours of various values of $\frac{K_{IC}}{\sqrt{\pi r}}$ (5−25 MPa) from the pore−emanating crack model (Eq. 2). The range of UCS given by the wing-crack model is also plotted as a shaded region. The mechanical data are cross-cut by the contours, suggesting a change in the dominant porous structure. At porosities > 0.25 the UCS given by the pore-emanating crack model with $\frac{K_{IC}}{\sqrt{\pi r}}$= 5−10 MPa seems to fit the data well. For porosities ranging from 0.12−0.2 the UCS given by the pore-emanating crack model with $\frac{K_{IC}}{\sqrt{\pi r}}$ = 10−15 MPa encloses the data. The UCS for the densest rocks in the study (~0.08−0.12) would suggest yet a higher $\frac{K_{IC}}{\sqrt{\pi r}}$ of 20−25 MPa. For porosities < 0.1 the UCS given by the wing−crack model is similar to the mechanical data (σ = 54.2−89.7 MPa).**

Line 146: the authors should state clearly that they here define "lava" as something at high temperature (volcanic rocks may often also be referred to as lava).

-We thank the reviewer for this comment and agree the definition between cold lava and

high temperature lava was unclear at certain points in the manuscript. We have edited the manuscript with the following note:

'[note: From here, samples deformed at high temperature will be defined as lavas, and those tested at room temperature as rocks].'

Line 184: in which direction has the lobe moved (*i.e.* rotation, inflation, advance)?

-We have now added that: The mechanism of movement is thought to be down-slope advancement to the East (Matsushima and Takagi, 2000).

Line 279, and elsewhere: is this a change of 1 % or 1 vol.%?

-It is a change of 1 vol.%. We have changed the manuscript to read 0.01 in place of 1%, in line with the method of reporting porosity used throughout the manuscript (e.g. Table 1). We have also noted this change elsewhere in the manuscript so that porosities are reported as a fraction. Where crystals are concerned we have reported these as vol.%

170   Line 296 and 298: which software was used to run the scripts? Are they publicly available?

-The software used to run the scripts was MATLAB. A version of the script is now freely available on Github (https://doi.org/10.5281/zenodo.1287237). This has now been made clear in the manuscript.

175   Line 322: the observation that in dense materials the connectivity is higher than in porous materials is somewhat counter-intuitive, and runs counter to results from previous researchers
(Farquharson *et al.* 2015 *JVGR* and Collombier *et al.* 2017 *EPSL*, both of which noted that connectivity generally increased with increasing porosity). Do the authors have any comment on
this difference?

-We agree it seems counter intuitive, but the difference is very small: a matter of 0.01-0.02 unconnected pores. We attribute
180   this to the presence of more small, isolated pores in the high porosity samples than in the low porosity samples. We also note that the porosity range of our rocks is considerably smaller (0.09-0.32) than Farquharson et al., (2015) (0.025-0.73) and Collombier et al., (2017) (~0.0-1.0), and so note that we are not evaluating the same range as these other studies. Our 'high porosity' samples could be considered low porosity in comparison to these other studies and so a comparison cannot be clearly drawn.

185

Line 342: cristobalite isn't a polymorph of quartz. Rather, both quartz and cristobalite are silica polymorphs.

-We thank the reviewer for bringing this to our attention and agree the text would be clearer if the polymorph was referred to as a silica polymorph. The manuscript and Figure 3 have been edited in accordance with this.

190   Line 355-359: did sample volume change upon heating? This really is surprising, as one would anticipate thermal cracking upon heating and subsequent cooling, due to thermal expansion
mismatch between the constituent microstructural components (*e.g.* Browning *et al.* 2016 *GRL*). This may be especially pronounced given the existence of cristobalite in your samples,
which undergoes a significant volumetric change as it transitions between its alpha and beta forms (a function of temperature:
195   see for example, Damby *et al.* 2014 *JAC*).

- We refer Dr Farquharson to the supplementary data, S5. Thermo-mechanical analysis (TMA) of the Mt. Unzen dome rock shows thermal expansion upon heating until the softening point of the material. We also refer him to lines 503-506 in the original manuscript ("A recent study by Eggertsson et al., *in review*, found that the strength and porosity of samples that hosted microfractures (like Mt. Unzen dome rock) were not affected by thermal stressing, while those that showed a trivial fraction
200   of pre-existing micro-fractures were more significantly influenced through thermal stressing and as a result became more permeable."). In accordance with Eggertsson et al., (2018), as our samples contained a pre-existing network of microfractures they were not readily fractured by thermal expansion and contraction.

We also would like to point out that samples containing cristobalite, UNZ-13 (see Figure 3), were not tested at high temperature nor thermally stressed to avoid adding effects of mineralogical reactions in this study that would not be relevant in the lava
205   dome setting – i.e. the cristobalite post-dates high-temperature emplacement. We also would like to state that the heating and cooling rates used (4 °C min$^{-1}$) were low enough to limit the differential expansion of the samples (that is, caused by temperature gradients across the samples, which also contribute to thermal cracking and which are not always considered independently from differential expansion of the constituent phases).

 it is not immediately clear from the figure, but does the rock stiffness increase with thermal stressing? Here you identify microfractures as the culprit, yet previously you

indicate that thermal stressing doesn't affect the porosity of the samples. Moreover, I would think that induced cracking would serve to decrease the material stiffness, rather than increase it.

I suggest the authors re-word this section for clarity and consistency.

215   - No, the rock stiffness (i.e. Young's Modulus) does not increase with thermal stressing, it in fact decreases (Figure 9c and lines 479-498, original manuscript). We attribute the initial convex portion of the stress-strain curves to the closure of microcracks perpendicular and sub-parallel to the principal stress (as in Heap et al., 2014, JGR). This initial section of the stress-strain curve is more pronounced in the case of the thermally stressed samples (Figure 4c), indicating that either 1) the width of pre-existing macro-fractures increased or that 2) more microfractures (generated by thermal stressing) are available

220   to close; however, the porosity determined from post-thermal stressing pycnometric measurements suggests that no micro-fracturing took place. We suggest that thermal stressing did indeed slightly modify the network of micro-fractures but on too-small a scale to affect the UCS and pycnometry results. The manuscript has been edited to highlight this observation and changes have been made to section 3.2.1.

225   Line 400-401: as previously, it would by useful if the authors were to distinguish between % and vol. %.

  - We have amended this throughout, as previously mentioned.

Line 431: Young's modulus ought to be capitalised. Also, there is a full stop missing at the end of this sentence.

-These typos have now been corrected in the manuscript.

230

Line 441-445: see comment above concerning these inclusion models.

-We have added analysis of these micromechanical models; see reply to comments above.

Line 470: could this observation be due to pore anisotropy (*e.g.* Bubeck *et al.* 2017; Griffiths *et al.* 2017)?

235   - Here the reviewer is referring to the statement "This suggests that these samples are stiffer than the other specimens tested, and indeed those experiments reached unusually high peak stresses at relatively low strains to failure.", which is discussing samples UNZ-2 and UNZ-13. We looked at the thin section of UNZ-13 and conclude that the pore anisotropy may indeed contribute. Here the pores are preferentially aligned toward the principal stress direction (Fig.3) and so this is a likely case for stiffening. The manuscript has been amended at this line and in the conclusions section, see the reviewer's comment below.

240   For UNZ-2 it is possible that these vesicles had a pore anisotropy that could have led to minor strengthening and an increase in stiffness, like that seen by Bubeck *et al.* (2017) and Griffiths *et al.* (2017). Yet, our investigation was not sufficient to constrain and explicitly state this as fact.

Line 480-481: do the authors have any information on the pore size distribution or pore anisotropy that could help explain

245   this?

-The reviewer is referring to the following statement "In addition, thermally stressed samples have slightly lower (~ 0.5–1.5 GPa) Young's Moduli than their unstressed equivalents, as previously noted in dacites from Mt. St. Helens (Kendrick et al., 2013a). This highlights a potential change in

porosity distribution that was not recognised by other means (e.g. total porosity, strength)."

250

The decrease in Young's Modulus is very slight, therefore information on pore size distribution or pore anisotropy, e.g. from a thin section, is unlikely to help in explaining this decrease. As the samples are very heterogenous any change in pore behaviour would be difficult to quantify as the change would be expected to be small and comparing the exact material before and after thermal stressing is not possible in thin section. This would rely on pre and post stressing CT scans which are time consuming and would likely provided reconstruction with spatial resolution too low to accurately distinguish slight pore morphology changes. Hence, the most sensitive measure of change we have is the pycnometry – which as stated previously, did not indicate any change – and the mechanical data which shows a decrease in Young's modulus.

Line 525: shear-induced development of pore connectivity was also shown experimentally by Kushnir *et al*. 2017 *EPSL*.
-We regret the oversight in not citing this article and have now added it to the manuscript.

Line 535-560: as with my previous comments concerning the inclusion models of Ashby and Sammis and Sammis and Ashby, I believe the authors could extend this Deborah number
analysis some more. As far as I can tell, the dimensions of Equation 10 only balance out if $b = 1$. Pa = [Pa.s × s^-1] is fine ($b = 1$), but Pa = √[Pa.s × s^-1] is not, as would be the case were $b = 0.5$. To me,
this highlights a serious shortcoming in the empirical approach adopted here, especially as there has been more recent work on the topic which adopt physical rather than empirical
parameters, for example the Wadsworth *et al*. 2017 chapter referenced elsewhere in the manuscript. For example, that chapter addresses the physical scaling of *De* with crystal
content. Ultimately, a similar physical approach could yield a much more generally applicable failure criterion for porous materials.
-With regards to the units of Equation 11, the equation balances if constant k has units of MPa.s$^{1/2}$, as shown below:

$$\sigma(MPa) = k\left(MPa.s^{1/2}\right).\dot{\varepsilon}_{obs}(s^{-1})^{b=1/2}$$

In the Wadsworth *et al*. 2017 chapter, only the single or two-phase cases are discussed. Within this chapter, the equation for the Deborah number is only given for a two-phase, crystal-bearing medium (Equation 3, Wadsworth et al., 2017). Here we are working with a three-phase medium, for which no models exist and thus an empirical approach must be adopted. For the sake of this argument, if we assume that the material from Mt. Unzen is a two-phase, crystal-bearing medium we can use Figure 2 in Wadsworth et al., (2017) to calculate the critical Deborah number expected. Given that the Unzen material has a crystal content, $\phi_x$, (microlites+phenocrysts) of ~0.75 we can use this to find the critical Deborah number if no bubbles were present. To find this critical value, the maximum packing fraction, $\phi_m$, also has to be known. As $\phi_m$ is defined as "the volume fraction of particles beyond which there is no space remaining which would accommodate further particles" (Mader et., 2013), it is clear from thin section and SEM images that our material has not yet reached $\phi_m$ (see Fig. 3). Therefore, it can be assumed that for the Mt. Unzen material investigated here, $\phi_m \approx 0.76 - 0.99$, and $\frac{\phi_x}{\phi_m} = 0.76$-0.99. According to Figure 2 in Wadsworth et al. (2017), this range gives the range of critical Deborah number as 9.9x10$^{-5}$-7.6x10$^{-4}$. Therefore, our estimation of the critical Deborah number of the dense material as 9.4x10$^{-5}$ -6.6 x10$^{-4}$ is a very reasonable one indeed.

The method used in this manuscript to find the Deborah number and then critical Deborah numbers is empirical but, as well as giving a working solution to find the Deborah number for a three-phase material, it shows that there is a reduction in critical Deborah number due to the addition of particles and also provides a linear relationship for the critical Deborah number and the addition of pores. This study is a novel, first-step approach into characterising the De number and failure constraints on real, volcanic samples using mechanical testing data.

Line 535: using *exp* as a subscript is a little ambiguous (at first glance I presumed it signified an exponential).

-For clarity, we have changed the *exp* subscript to *obs* to stand for observation in line with the symbol for observation time

295   $t_{obs}$.

Line 546: Oswald should presumably be Ostwald.

-Yes, this has now be changed in the manuscript.

300   Line 546: here, the authors state values of *k* and *b* of 1653 and 0.5, yet in the caption for Figure 10, the values are *k* = 1606
and *b* = 0.7755. Which of these are correct? What is the
sensitivity of the the following analysis to variations on *k* and/ or *b*?

-We thank the reviewer for pointing out this typo. The correct k and b values were those written within the manuscript. These

values shifted due to an addition of data to the plot. We have now added the standard error of estimate for these values to the

305   plot to show the variation of k and b.

Equation 11: Based on equation 9, shouldn't σ relate to strain rate × viscosity (or equivalently, *De*× *G*∞)?

-We believe the reviewers confusion lies with the presentation of the equations, Equation 8 has now been edited from:

$$De = {}^{\eta_m}/G_\infty t_{obs}$$

310   **to:**

$$De = \frac{\eta_m}{G_\infty t_{obs}}$$

(now Equation 9)

**And Equation 9 from:**

$$De = \dot{\varepsilon}_{obs}\, \eta_m / G_\infty$$

315   to:

$$De = \frac{\dot{\varepsilon}_{obs}\, \eta_m}{G_\infty}$$

(now Equation 10)

To get to Equation 11 from Equation 9 a substitution is made for $\dot{\varepsilon}_{obs}$, which from Equation 10 is $\dot{\varepsilon}_{obs} = \left(\frac{\sigma}{k}\right)^{1/b}$.

(now Equation 11)

320

Line 618: this is in contrast to existing theory, models, and experimental data. Perhaps this effect is masked in your data by
sample heterogeneity? I would be wary of including this point as
a key conclusion of the study.

-The reviewer is referring to conclusion 3: The orientation of a vesicle may not necessarily have a discerning control on the

325   strength of a rock, however it does have an influence on the strains reached at failure and, as such, the Young's Modulus. Here

we agree with the reviewer and this point has now been removed as a key conclusion, in light of the discussion regarding

anisotropy in the reply to an earlier reviewer comment above made regarding line 470. The manuscript has also been edited to

reflect this change.

330   Figure 4: there appears to be some data obscured by the legend in panels *a* and *b*.

-We thank the reviewer for his keen eye and have clarified the figure accordingly.

Figure 6: for clarity, perhaps the authors could plot viscosity as a function of strain rate (similar to Figure 10a; perhaps with symbols coloured for time).

335 -As the stress (which is proportional to viscosity) is plotted against strain rate on Figure 12b (previously Figure 10a). The b-value obtained from the curve in Fig 12b is 0.5 which matched the values obtained from previous studies on crystalline dome material (Caricchi et al., 2007; Lavallée et al., 2007, 2012), therefore we feel that there is no more information to be gained by alternative plots.

340 Figure 10: in panel *a*, the authors state that the equation is shown on the figure, but it is missing. In panel *b* (and line 555), is the relation given by the yellow line based on only three data
(i.e. the transitional data)? What is the r2 value of this relation? Can the authors comment on the theoretical value of *De_c* for a nonporous material? Would this fit on the trend? Likewise,
how do the authors anticipate *De_c* evolving for highly porous materials?

345 -We thank the reviewer for highlighting this point which has spurred a number of changes (also in light of reviewer 2's comments). We now refer the reviewer to the revised Figure 12 and additional manuscript changes which have been edited to show two separate transitional regimes: viscous-dominated and brittle-dominated. Figure 12 shows clearly the evolution of failure and at which critical Deborah numbers this occurs. Figure 12c shows the standard error of estimate windows of this data and their $R^2$ values.

350 For a non-porous material, the data suggest the Critical Deborah number would lie between $De_c = 1\times10^{-4}$-$6.6\times10^{-4}$. This is approximately two orders of magnitude lower than that reported by Webb and Dingwell (1989), which we attribute to the high crystal content as crystals also decrease this critical Deborah number (Cordonnier et al., 2012; Wadsworth et al., 2017), see also the reply to reviewer's comments on lines 535-560 of the manuscript.

For the porosity range of material tested herein, we expect the critical Deborah number to follow a linear trend as shown in
355 Figure 12c.

**Response to Quane**

This paper is potentially an excellent contribution to the experimental literature on natural, complex, multi-component, volcanic rocks. Initially, the authors provide an excellent treatment of the relevant rheological behavior of high-temperature
360 deformation in glassy samples which is followed by a good characterization of representative, pre-experiment cores. The study comprises a substantial number of experiments both at ambient room (20°C) and magmatic temperatures (900°C). The rheological data is of high quality and the experimental conditions and span a reasonable range of deformation rates and timescales. The authors do an excellent job of presenting the results separately from analysis and a thorough job of explaining the major decisions and assumptions they had to make in the process of running the experiments (i.e., how and why samples
365 were chosen based on porosity, connected porosity, etc., detail on how experimental charges were loaded and the effect that has on the experiment). The rheological analysis is solid and prima facie, the interpretations seem sound and lead to a number of logical conclusions about the behavior of these multicomponent systems under conditions relevant to Unzen eruptions.

-We thank the reviewer, Steve Quane, for his concise summary and descriptive comments which are answered below.
370

However, I see one main oversight in their otherwise detailed and robust analysis; there are no descriptions, photos, or representative images and quantitative measurements (porosity, density) of experimental run products. This is a major issue for several reasons:

a) Deformation in experimental charges cannot be interpreted by using the rheological data alone. For example, seemingly "viscous" behavior and "brittle" behavior were interpreted (starting is section 4.1) based on the "mechanical responses" of the rocks. The authors make assumptions and "attributions" about the actual mechanical behavior of the samples with no empirical evidence. For example, they "attribute to a narrowing of pre-existing cracks" and "hypothesize may reflect a contribution of viscous deformation upon loading". It is possible that these interpretations are correct, however, it would be relatively easy to test the attributions and hypotheses by halting an experiment at the requisite place on the deformation path and doing microstructural analysis.

-We thank the reviewer for bringing this point to our attention. With reference to the high temperature viscous responses we refer the reviewer to work by Cordonnier et al., (2012), referenced in the text, who look, in detail, at the mechanical response curves of deformed samples and label them 'viscous', 'transitional' and 'brittle' according to X-ray microcomputed tomography scans of deformed samples. We particularly refer to Figures 1 and 2 in Cordonnier et al., (2012) – as such we are not defining new regimes but simply categorising our samples according to regimes already defined. With reference to the brittle experiments, we refer the reviewer to publications mentioned in the text (Hoek and Bieniawski, 1965; e.g. Brace et al., 1966; Scholz, 1968; Heap et al., 2014a) who describe, in detail, the four stages of mechanical loading and brittle failure with reference to the mechanical stress-strain curves. Although the authors agree that 'halting an experiment at the requisite place on the deformation path and doing microstructural analysis' would be a very informative study, this work has already previously been tackled, stress-strain curves have been dissected with respect to sample attributes, and for brittle experiments, the regimes of crack-closure, elastic deformation, strain hardening, and failure are well defined and identifiable by the mechanical curves. Therefore, we reassure the reviewer that care has been taken when labelling a response as 'viscous' or 'brittle' by referring to previous work in rock and lava deformation, based on their stress-strain curves.

That said, we understand the need to examine the experimental run products and have taken the reviewers concerns on board. We have illustrated our examination of the experimental products acquired by SEM imaging to create a new Figure 5, along with further descriptions and photographs of samples after deformation to Figure 12 (previously Figure 10). This analysis, based on the reviewers' comments made us revisit our labelling of samples, and we have issued a new comprehensive, visual description of failure, as seen in Figure 12. We also further concluded that the state of 'transitional' can be further sub-divided to clearly express that it is a spectrum leading from viscous behaviour, indicated from a continuous plateau in stress with substantial strain, to brittle behaviour, defined by a sharp drop in stress with little strain beyond an initial elastic loading response. Therefore, we suggest that a sample can either be in the viscous dominated regime while undergoing a transitional behaviour, where the stress plateaus with strain but there are small stress drops along the way, or the brittle dominated regime where the stress-drop is poorly defined and 'curves' before reaching high strain at failure (Figure 5; Figure 11). Although the post deformation photographs and SEM images are a useful guide, as the same strains before experiment termination/sample failure were not met by every sample (we chose our end strains based on characteristics of the stress-strain curves not on a set total strain) the results are not entirely comparable e.g. a viscous sample experiment would be terminated at much shorter strains (than a transitional sample) as its curve was already defining viscous behaviour. Thus, we consider the stress-strain curves a better method for quantitatively of defining the deformation mode of lavas.

b) Post experiment analysis of end products can lead to surprising conclusions about mechanical behavior. In these multicomponent systems, deformation can occur via several mechanisms. Bubble collapse, brittle fracturing, viscous flow of groundmass glass, microcracking, rotation of grains, grain boundary sliding, internal grain deformation. All of these are factors in accommodating strain in the samples. Hence, strain can be accommodated homogenously (throughout the sample evenly)

or it can be localized into bands or disparate parts of the sample. Without post experiment analysis, these important rheological behaviors cannot be determined. The authors are making the most logical conclusions about their "brittle" and "viscous" determinations based on the rheological data, however, without visual analysis of bulk properties and microstructures, the authors cannot confirm behavior. In addition, they are losing a considerable amount of important information about the nature of the deformation.

-We thank the reviewer for his comments and agree, understanding the complex mechanisms that led to failure in volcanic rocks is important, yet, here, this paper is not trying to decipher the deformation mechanism (e.g., viscous, plastic, brittle) but the deformation mode of lavas (i.e, ductile vs brittle) necessary to constrain (and distinguish between) flow and fragmentation processes. [Please note that the distinction between the two is that of scale: a deformation mode refers to the macroscopic character of sample deformation whereas a deformation mechanism refers to microscopic deformation processes. Thus, unfortunately in this field of laboratory testing, brittle may be used when refereeing to both a deformation mode (sample failure) and a deformation mechanism (i.e., a cracking event); see also Rutter in Tectonophysics (1986) and Heap et al. in Bull. Volc. (2015a) for clarity] We have conducted further analysis of the experimental products as described above. We also guide the reviewer to Figure 2 in Lavallee et al., (2007) where post-experiment textures have been viewed and deformation mechanisms discussed, and to Figure 2 in Kendrick et al., (2013), as well as Figure 2 in Kendrick et al., (2017), where textural evolution with strain is depicted and the deformation mechanisms are interpreted. Appreciating the need for a more in-depth explanation of the overarching deformation mechanisms in the deformation mode discussed (e.g. 'brittle' and 'viscous' and offer the reviewer the new Figure 12 (previously Figure 10) with accompanying edits in the manuscript. A detailed study of the exact microstructural deformation mechanisms at play across all samples is beyond the scope of this paper seeking to constrain deformation mode (not mechanism), and as it has already previously been discussed in other studies, we chose to highlight samples representative of each regime and map the textures associated with the different deformation regimes and link these to the stress-strain curve characteristics used to define the remaining samples.

c) Post experiment analysis of physical properties (i.e., density, porosity) can yield important information on the nature of deformation. Certainly, for the cores that were not destroyed during brittle failure, the authors can make density and porosity determinations via the methods they used on the pre-experiment cores. Bulging of cores may cause a little consternation, however, established methods exist in the volcanology literature to measure density and porosity on irregular samples.

-We advise the reviewer that samples that remained completely intact (only those with a completely 'viscous' response, i.e. those carried out at strain rates of $10^{-5}$ s$^{-1}$) were re-measured to constrain changes in connected porosity. However, the results showed no significant change in porosity, nor in the volume of the sample determined by pycnometry (Table 1); hence, we mention this in the text but do not present the data in the study. Due to minor loss of volume from the experimental process (removal of sample from pistons etc.) the pycnometer readings are within error and thus we concluded, cannot be considered.

Table 1. Example of volume measurements made using pycnometery on samples that remained intact after deformation. Fractional change in volume is $< \pm 0.05\%$ of the measured volume

| Sample | Initial porosity | Strain rate tested (s$^{-1}$) | Temperature tested (°C) | Measured volume before (cm$^3$) | Measured volume after (cm$^3$) | Fractional change in measured volume |
|---|---|---|---|---|---|---|
| UNZ-4-16 | 0.12 | 1.00E-05 | 900 | 12.17 | 12.14 | 0.003 |

| UNZ-4-17 | 0.12 | 1.00E-05 | 900 | 11.67 | 11.72 | -0.005 |
| UNZ-8-16 | 0.18 | 1.00E-05 | 900 | 11.38 | 11.32 | 0.005 |

d) Characterizing the amount of strain in the samples is an independent measure of machine strain. Does the sample show the same amount of strain as the machine? This can be determined through post-experiment analysis of density, porosity and core geometry. It is an important check on the experimental apparatus to ensure all strain from the machine is going into the sample. Quane and Russell, 2005 (cited by authors) and Quane et al., 2004 from American Mineralogist go through these procedures in detail.

We refer the reviewer to the first paragraph in section 2.3. of the manuscript: '[Note: all mechanical data have been corrected for the compliance of the setup, quantified via Instron procedures that monitor length changes due to loading of the pistons in contact with one another]'. This compliance method is carried out at all temperatures tested in our laboratory. Following the application of the compliance correction, the total strain referred to in the manuscript is the sample strain and not machine strain. Post-deformation sample geometry (i.e. final sample length, for the in-tact samples) was measured for the samples to confirm final strains were correct. This point has been added to the manuscript, as well as "…at the relevant experimental temperature" in the sentence describing compliance, to clarify that the different behaviour of the machine at temperature is also accounted for. To clarify, the method to quantify strain in deforming porous samples in Quane et al., (2004) and Quane and Russell (2005) may be applied for glass-bead compacts, but unfortunately not for natural multi-phase material.

Without post-experiment characterization (on samples that will allow it-sometimes even brittle deformation samples can be salvaged and epoxyed), the authors cannot speak with authority on the types of deformation occurring. Unfortunately, by not having that authority, the Conclusions they draw come into question. Certainly, the authors can do an analysis of the run products and produce a figure or two (like Figure 3 does for pre-experiment cores) to describe the major mechanisms of deformation and strain accommodation. Without this, this otherwise very strong, methodical and detailed contribution falls incomplete.

-As mentioned above, this study is first and foremost concerned with a description of the macroscopic deformation modes of lava, not the deformation mechanism. Yet, we fully agree that textural information provides insight into the underlying microscopic deformation mechanism. We draw the reviewer's attention to the newly created Figure 5 and Figure 12 a). Due to the fragmental nature of the samples, particularly those marked as having a brittle or brittle-dominated response, it was impracticable to reconstruct the position of each fragment with epoxy, yet we looked at some fragments (new Figure 5) taken from the inner part of the sample. We provide new data in Figure 10 (now Figure 12), containing photographs of the run-products and accompanying comments in the manuscript. With the photographs, the now more detailed explanation of the curves, and the SEM images in Figure 5, we believe we have satisfied the reviewers concerns about sample characterisation.

Technical corrections in this manuscript a very minimum. Found one spelling mistake, but I lost it!

-We thank the reviewer for searching the document for typos, we have found the assaulting spelling mistake mentioned and have track changed it in the manuscript.

**List of relevant changes**

The following manuscript has been edited with tracked changes to indicate what amendments have been made after reviewer's comments. Minor text edits have been made throughout the manuscript to aid with flow and to correct for typing issues, major changes include:

500
- The addition of a translated abstract from English to Japanese
- The addition of references suggested by reviewers
- The extension of our analysis of our data with comparison to the pore-emanating crack model and wing-crack model, this includes text additions to sections 1.3 and 4.1.2 and the addition of a new figure, Figure 11.
- The addition of an accessible MATALB code for the calculation of Young's Modulus, found at:
505 https://doi.org/10.5281/zenodo.1287237
- A more detailed description of the rheological response of the dome lavas, after comments by Steve Quane. This involved carrying out more SEM work, as seen in Figure 6, and the addition of text in section 3.2.2
- An updated description of the effect of pore anisotropy on rock strength, section 4.1.2, after comments by Jamie Farquharson
510
- An updated description of the failure criterion of porous lavas in section 5.2. This included the newly updated Figure 10, which is now labelled Figure 12.
- Edits were made to Figure 3, Figure, 9 (Now Figure 10), and Figure 4 for consistency
- A correction was made to the units of Equation 11 and a new reference added for clarity

515 The manuscript now includes 12 figures, 6 supplementary figures, and ~13,000 words excluding references and figure captions.

[revised manuscript text omitted]